# The sex of organ geometry

Laura Blackie[1,2,3,11], Pedro Gaspar[1,2,3,11], Salem Mosleh[4,10], Oleh Lushchak[1], Lingjin Kong[1,2], Yuhong Jin[1,2], Agata P. Zielinska[1,2], Boxuan Cao[1,2], Alessandro Mineo[1,2,3], Bryon Silva[1,2,3], Tomotsune Ameku[1,2,3], Shu En Lim[5,6], Yanlan Mao[5,6], Lucía Prieto-Godino[3], Todd Schoborg[7], Marta Varela[8], L. Mahadevan[4,9] & Irene Miguel-Aliaga[1,2,3 ✉]

Organs have a distinctive yet often overlooked spatial arrangement in the body[1–5]. We propose that there is a logic to the shape of an organ and its proximity to its neighbours. Here, by using volumetric scans of many *Drosophila melanogaster* flies, we develop methods to quantify three-dimensional features of organ shape, position and interindividual variability. We find that both the shapes of organs and their relative arrangement are consistent yet differ between the sexes, and identify unexpected interorgan adjacencies and left–right organ asymmetries. Focusing on the intestine, which traverses the entire body, we investigate how sex differences in three-dimensional organ geometry arise. The configuration of the adult intestine is only partially determined by physical constraints imposed by adjacent organs; its sex-specific shape is actively maintained by mechanochemical crosstalk between gut muscles and vascular-like trachea. Indeed, sex-biased expression of a muscle-derived fibroblast growth factor-like ligand renders trachea sexually dimorphic. In turn, tracheal branches hold gut loops together into a male or female shape, with physiological consequences. Interorgan geometry represents a previously unrecognized level of biological complexity which might enable or confine communication across organs and could help explain sex or species differences in organ function.

Recognition that internal organs reside in specific positions in the body is arguably as old as the study of anatomy, with anomalies such as mirror image transpositions ('situs inversus') described as far back as Aristotle[2]. Although our mechanistic understanding of how organs acquire their characteristic sizes and shapes is advancing at a remarkable pace[6–8], the factors determining their higher-level spatial arrangement have received less attention. This is partly because multi-organ relationships are less amenable to the molecular, genetic and imaging approaches which have enabled progress at the organ level, but also because it is easy to dismiss the consistency of such spatial arrangements as a developmental accident.

In principle, a relatively small animal such as *Drosophila melanogaster*, with specialized organs and well-established mechanisms of endocrine interorgan communication, provides an opportunity to investigate the spatial arrangement of internal organs; its sophisticated experimental tools enable functional interrogation of the underlying genetic mechanisms. In practice, its chitin exoskeleton has rendered previous attempts to visualize its internal organs comprehensively either not three-dimensional (3D) (for example, dissections) or relatively low-throughput (for example, tissue clearing).

We now circumvent this issue by scaling up micro-computed tomography (microCT)[9–11], previously applied to smaller groups of flies[12,13], to acquire 3D volumetric scans of larger numbers of intact adult *Drosophila* (Fig. 1a,b, Extended Data Fig. 1a,b, Supplementary Video 1 and Methods). This allows us to quantitatively describe and functionally interrogate the shape, position and relative arrangement of most organs.

## Gut shape is stereotypical and sexually dimorphic

We first focus on the intestinal tract, repurposing a neurite tracing tool to extract the centreline of the gut from our scans and convert it into a 3D shape (Fig. 1c,d and Methods). Geometric morphometric shape analysis reveals several recurrent features; as well as a previously described hindgut clockwise coil[14–16], we observe a sharp bend in the central R3 midgut region surrounded by two loops with antiparallel turning angles. As a result of this configuration, two regions of the main digestive portion of the intestine (midgut R2 and R4 regions) which might be assumed to be apart are, in fact, stacked together (Fig. 1c,d). More unexpectedly, we find that both the shape of these midgut loops, the hindgut and that of the intestine as a whole are sexually dimorphic (Fig. 1d–f, Extended Data Fig. 1c, Supplementary Tables 1–6 and Supplementary Video 1). Female guts are longer and thicker, as might be expected from their overall larger size[17] (Extended Data Fig. 1d–f) but this is not allometric: the sex difference in gut length is accounted for by the length of the central midgut loops, whereas the length of the anterior midgut and hindgut are comparable between the sexes (and therefore relatively shorter in females) (Fig. 1g and Extended Data

[1]MRC Laboratory of Medical Sciences, London, UK. [2]Institute of Clinical Sciences, Faculty of Medicine, Imperial College London, London, UK. [3]The Francis Crick Institute, London, UK. [4]School of Engineering and Applied Sciences, Harvard University, Cambridge, MA, USA. [5]MRC Laboratory for Molecular Cell Biology, University College London, London, UK. [6]Institute for the Physics of Living Systems, University College London, London, UK. [7]Department of Molecular Biology, University of Wyoming, Laramie, WY, USA. [8]Faculty of Medicine, National Heart & Lung Institute, Imperial College London, London, UK. [9]Departments of Physics and Organismic and Evolutionary Biology, Harvard University, Cambridge, MA, USA. [10]Present address: Department of Natural Sciences, University of Maryland Eastern Shore, Princess Anne, MD, USA. [11]These authors contributed equally: Laura Blackie, Pedro Gaspar. ✉e-mail: irene.miguelaliaga@crick.ac.uk

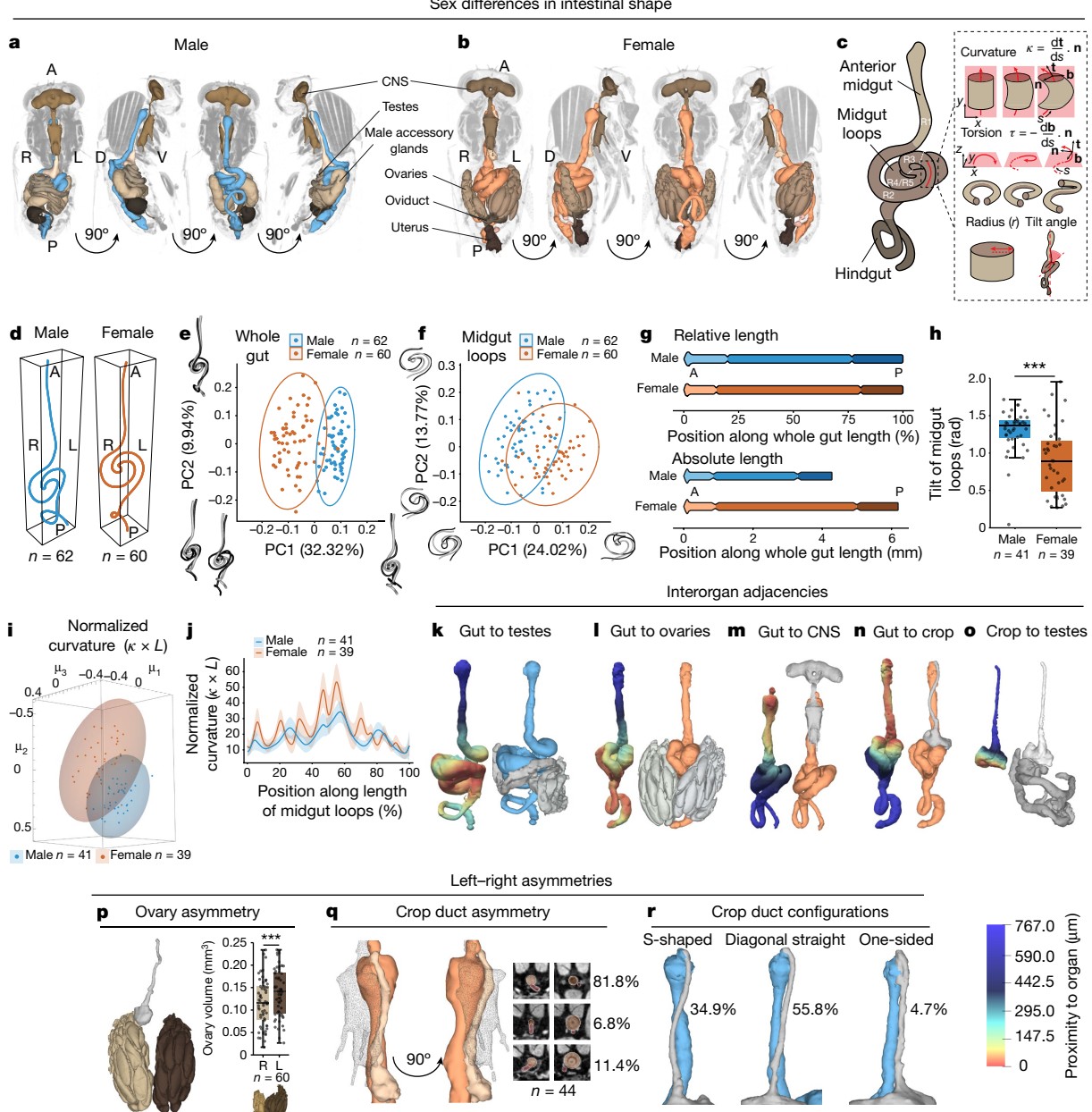

**Fig. 1 | Sex differences in organ shape and interorgan adjacencies.**
**a,b,** Anteroposterior microCT slices overlaid with 3D organ reconstructions for male (**a**) and female (**b**) fruit flies. **c,** Gut regions (R1–R5) and shape descriptors: curvature ($\kappa$); torsion ($\tau$); radius ($r$) of gut tube and tilt angle of midgut loops relative to main gut axis; **t,** tangent vector; **n,** normal vector; **b,** binormal vector; and $s$, arclength. **d,** Average gut centrelines. **e,f,** Gut shape variability PCA plot for whole gut (**e**) and midgut loops (**f**) with extremes of variation along each PC depicted (Methods). **g,** Relative and absolute lengths of anterior midgut, midgut loops and hindgut represented by colour shades. **h,** Male midgut loops are on average more tilted (horizontal) than females. **i,** Multidimensional scaling plot showing significant difference between male and female normalized gut curvature. **j,** Females have higher average normalized gut curvature for most of the midgut loop region. **k**–**o,** 3D segmentation heatmaps showing gut proximity to testes (**k**), ovaries (**l**), CNS (**m**) and crop (**n**) and crop proximity to testes (**o**). **p**–**r**, 3D segmentations showing crop contacting right ovary (**p**), crop duct proximity with gut and CNS (**q**) and crop duct configurations (**r**). **p,** Left ovary (dark brown) is on average larger than right ovary (light brown). **q,** Dorsal–ventral cross-sections through top and centre of proventriculus showing crop duct position asymmetry (red) relative to gut (orange) (right). Line graph: mean and standard deviation. Boxplot: line, median; box, first and third quartiles; whiskers, minimum and maximum. **e,f,i,** Ellipses represent 95% confidence spaces. $n$ denotes number of biologically independent samples. Statistical significance was assessed using two-sided two-sample $t$-test (**h**) and two-sided paired $t$-test (**p**). \*\*\*$P < 0.001$. See Supplementary Information for organ contact frequencies, exact $P$ values, statistical tests and sample sizes. Blue, males; orange, females. CNS, central nervous system comprising brain and ventral nerve cord; R, right; L, left; A, anterior; P, posterior; D, dorsal; V, ventral.

Fig. 1d,e). Sex differences in gut shape were also apparent in two other independent genetic backgrounds: wild-type *CantonS* flies, as well as flies harbouring a mutation in the *white* gene ($w^{1118}$, commonly used in experiments involving transgenic flies) (Extended Data Fig. 1i–n and Supplementary Tables 7–10).

To understand what specific 3D features differ between males and females, we developed methods to quantify the local geometry of the gut along its entire length, parametrized by local curvature and torsion of the gut centreline, the radius of the gut tube along its length and the tilt of the midgut loops relative to the gut longitudinal axis (Fig. 1c gives

definitions and Methods describe algorithms used). Whilst the average torsion of the gut is comparable between males and females (Extended Data Fig. 1g), they differ in several shape features: female guts have a larger radius and their midgut loops have higher curvature and are less tilted than those of males (Fig. 1h–j and Extended Data Fig. 1f,h). Sex differences in gut shape and curvature are, to some extent, independent of sex differences in gut length: they are still apparent when curvature is normalized by length (Fig. 1i,j) and persist in female flies in which the length of the female gut has been genetically or environmentally shortened to be more comparable to that of males (*ovo*[D1] mutant flies[18] and flies starved for 48 h[19], respectively; Extended Data Fig. 1o–v and Supplementary Tables 11–13).

Together, these data show that the intestinal tract of adult *Drosophila* has a consistent yet sexually dimorphic 3D shape.

## Organ adjacencies are stereotypical and sexually dimorphic

As well as enabling investigation of the geometry of individual organs, our volumetric scans provide the opportunity to explore organ adjacencies in 3D. To this end, we segmented all visible organs and repurposed methods normally used in computer graphics for comparing surface mesh reconstructions to describe and quantify organ proximity (Methods). This revealed that organs are tightly packed in the body cavity (Fig. 1a,b), with some interorgan distances lying in the less than 10 μm range (Supplementary Table 14). Our analysis confirmed proximity between the testes and posterior midgut in males[20] and further revealed that this same intestinal region is adjacent to the left ovary in females (Fig. 1k,l, Extended Data Fig. 2a–d and Supplementary Table 14). We also observed unexpected adjacencies and previously undescribed left–right asymmetries. Indeed, the anterior portion of the small intestine abuts the abdominal ganglion of the central nervous system (Fig. 1m). The stomach-like crop is adjacent to the central loops of the small intestine, to the right ovary in females and to the testes in males (Fig. 1n–p, Extended Data Fig. 2e,f and Supplementary Table 14). Both the crop position and the ovary volume (and testes position[21,22] but not the testes volume) are asymmetric. Indeed, the crop duct typically (but not always) emanates from the left side of the foregut–midgut junction then turns towards the right side of the midgut and (possibly consistent with ref. 23) the left ovary is on average larger and less likely to make contact with the crop than is the right ovary (Fig. 1p–r and Extended Data Fig. 2g–i).

In many animals, including flies and humans, organ size and volume remain plastic in adult life; it is conceivable that such plasticity impacts the stereotypical organ distances and adjacencies we have observed. To begin to explore this idea, we sought to reduce organ size in adult females in two independent ways: nutritionally (by starving the flies for 48 h) and genetically (in female flies harbouring an *ovo*[D1] mutation which reduces ovary size and gut length[18,24]) (Extended Data Figs. 1r,v and 2k,l,n,o). Notably, organ shrinkage does not invariably result in increased interorgan distances: although the distance between gut and ovary was increased in *ovo*[D1] mutant females as expected, it was preserved in starved females (Extended Data Fig. 2j,m and Supplementary Tables 15 and 16).

Together, these data show that organ adjacencies are stereotypical and sexually dimorphic, and can be spared or modulated depending on context.

## Extrinsic control of gut shape by a vascular-like organ

Our analysis of *ovo*[D1] mutant female flies with reduced organ volumes (Extended Data Figs. 1o–r and 2j–l and Supplementary Tables 11 and 15) suggests that the tight packing of organs in the body cavity can act as a significant geometrical constraint on organ shape. Consistent with this idea, statistical analysis of how gut shape covaries with the volume of

adjacent organs (Extended Data Fig. 2q,r) indicates that sex differences in gut shape partly ensue from sex differences in physical constraints provided by the tight packing of neighbouring organs in the confined space of the abdomen (Supplementary Tables 17–19).

However, this covariation analysis revealed residual, unaccounted for variability, even after considering the contributions of gut length, the volume of other organs and potential batch effects (Extended Data Fig. 2p–r and Supplementary Tables 17–19): an idea further supported by the finding that sex differences in gut shape are still apparent in starved flies or *ovo*[D1] flies with greatly reduced ovaries (Extended Data Fig. 1p,q,t,u and Supplementary Tables 11–13). We had observed that intestinal regions with high curvature (for example, the central midgut loops) are more profusely populated by terminal branches of the tracheal system: a vascular-like system of interconnected tubes which delivers oxygen to insect organs (Fig. 2a,b and Extended Data Fig. 3a–c). Tracheal branches in this region often span adjacent loops, suggestive of a role for trachea in mechanically holding together apposed gut regions (Fig. 2c). To begin to test a potential role for the tracheal system in extrinsically controlling intestinal shape, we acutely ablated terminal tracheal branches in adult flies. This was achieved by expressing the pro-apoptotic BCL2-associated X (Bax) protein specifically in adult terminal tracheal cells. Effective and selective ablation was confirmed with markers for both the terminal tracheal cell nuclei and branches (*Drosophila* serum response factor (DSRF) staining and *trh(GMR14D03)-GAL4*-driven CD8::GFP expression, respectively)[25] (Fig. 2d and Extended Data Fig. 3d). Adult-specific loss of the terminal tracheal branches which populate the intestine resulted in a discernible effect on gut shape in both male and female flies (Fig. 2e, Extended Data Fig. 3d and Supplementary Tables 20 and 21). This effect was particularly prominent in the densely tracheated central midgut loops, which had become more relaxed and less tightly packed (Fig. 2f,g and Extended Data Fig. 3d).

Although these experiments suggested that tracheal branches hold gut loops together, they also ablated the terminal tracheal branches of other organs and reduced gut length (Extended Data Fig. 3d and data not shown). We therefore sought to more specifically target intestinal tracheal branches and considered gut-borne signals that might sustain intestinal tracheation. The fibroblast growth factor (FGF) ligand Branchless (Bnl) is expressed in the intestine and has been shown to promote tracheal branching[26–30]. Using a previously generated reporter of endogenous *bnl* expression (*bnl-LexA*)[31], we revealed a regional and temporally regulated pattern of *bnl* expression: transiently in intestinal muscles soon after adult emergence and, subsequently, in intestinal enterocytes (Fig. 3a). We validated this intestinal expression pattern by generating a new endogenous *bnl* reporter which does not delete any *bnl* exons (Extended Data Fig. 4a,c,d). In normal homoeostatic conditions and in contrast to regenerative contexts[28,30], no *bnl* expression was observed in trachea themselves or other intestinal cell types using either reporter (Fig. 3a and Extended Data Fig. 4a,c,d and data not shown). However, *bnl* expression is higher in gut regions which will become more highly and densely tracheated (Extended Data Fig. 4b): might gut-derived Bnl control the tracheation of the intestine? Temporal and cell-type-specific downregulation experiments using *Hand-Gal4* or *vm(GMR13B09)-Gal4* (two independent muscle drivers; see Extended Data Fig. 4e,f for their expression in gut muscles and other tissues) indicated that depletion of the Bnl ligand pool made by intestinal muscles during pupation/early adult life (but not that made by enterocytes later on in adult life; Extended Data Fig. 5a,b,e,f and Supplementary Tables 22 and 23) resulted in complete and specific loss of intestinal terminal tracheal cells, as revealed by markers for both the terminal tracheal cell nuclei and branches (DSRF staining and *QF6*-driven mtdTomato, respectively) (Fig. 2h and Extended Data Fig. 3e–h and data not shown). Akin to the effects observed on acute tracheal ablation, loss of intestinal terminal tracheal cells impacted several features of gut shape, including a reduction in curvature and

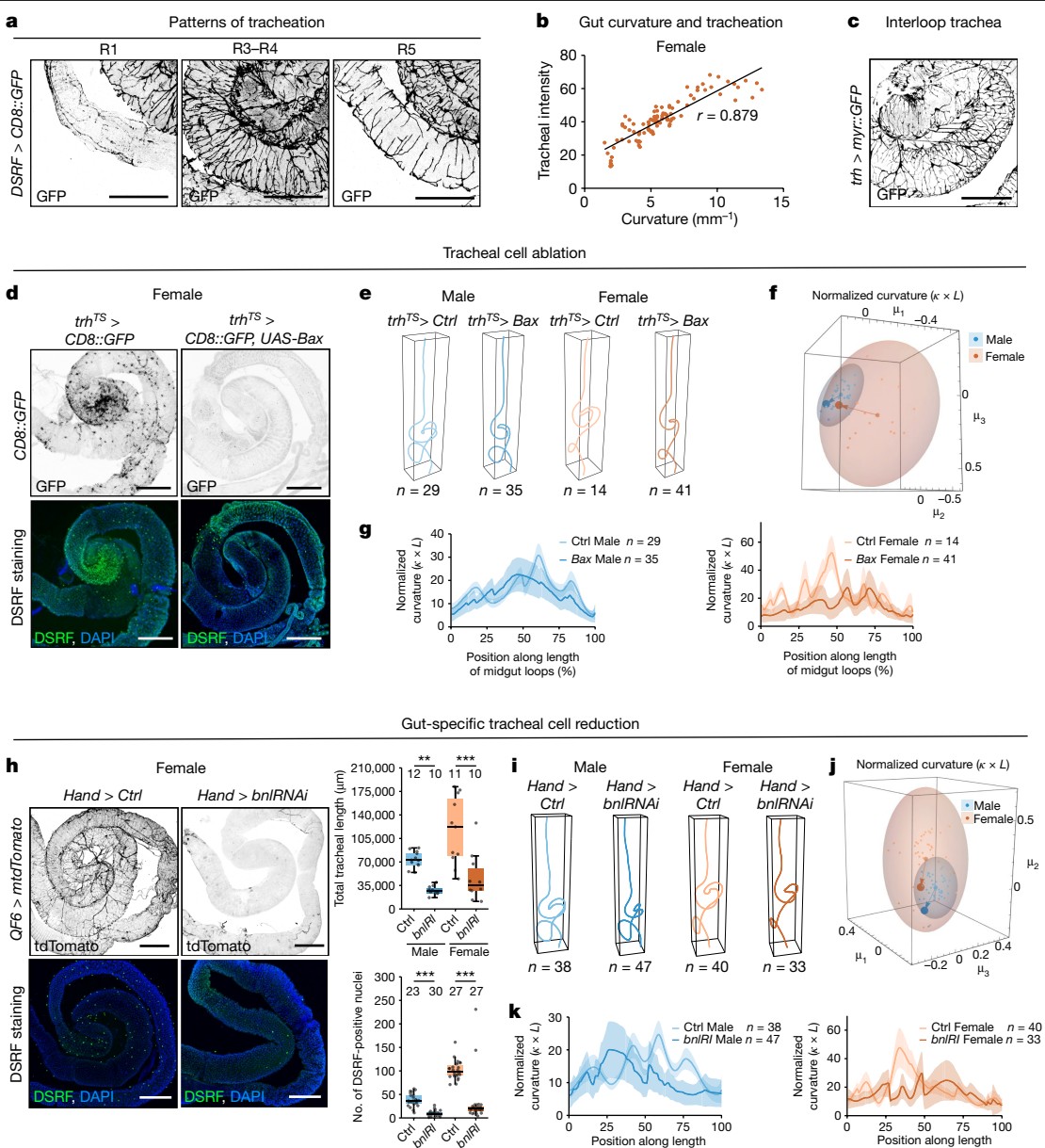

**Fig. 2 | Tracheal branches hold gut loops together. a**, Tracheal branches visualized in different gut regions: sparse and parallel to length of R1; dense and perpendicular to length of R3–R4; sparse and perpendicular to length of R5. **b**, Correlation between average intensity of tracheal signal from *btl>myr::GFP* females and average gut curvature of wild-type *OregonR* females at the same relative midgut positions. **c**, Tracheal branches span across gut loops. **d**, The *trh^TS>Bax* expression reduces tracheal terminal branches (top) and numbers of DSRF-positive nuclei (bottom) in female midguts. **e**, Average *trh^TS>Bax* gut centrelines show differences in gut shape relative to controls. **f**, Multidimensional scaling plot showing change in gut curvature normalized by gut length in *trh^TS>Bax* compared to controls. **g**, The *trh^TS>Bax* guts show reduced average curvature normalized by gut length relative to controls. **h**, *Hand>bnlRNAi* expression reduces tracheal branches (left top, quantified in right top) and number of DSRF-positive nuclei (left bottom, quantified in right bottom) in female midguts. *n* values are shown. **i**, Average *Hand>bnlRNAi*

gut centrelines showing differences in gut shape relative to controls. **j**, Multidimensional scaling plot showing change in gut curvature normalized by gut length between controls versus *Hand>bnlRNAi*. **k**, *Hand>bnlRNAi* guts show reduced average curvature normalized by gut length relative to controls. Line graph: mean and standard deviation. Boxplots: line, median; box, first quartile and third quartile; whiskers, minimum and maximum. Multidimensional scaling plot: ellipsoids represent 95% confidence space for each group, arrows represent shift in mean from control to experimental manipulation. *n* denotes number of biologically independent samples. Statistical significance was assessed using one-way ANOVA followed by Tukey post hoc tests (**h**). **$P < 0.01$; *** $P < 0.001$. Supplementary Information gives exact *P* values, statistical tests and sample sizes. Males, blue; females, orange; controls, lighter matching colours. Ctrl, control group (see genotypes in Supplementary Information). Scale bars, 200 µm.

relaxation of the midgut loops (Fig. 2i–k, Extended Data Fig. 3e,i–k and Supplementary Tables 24–27). We ruled out the possibility that *bnl* affects gut shape by acting on the gut itself by downregulating *breathless* (*btl*, coding for the receptor for Bnl) in gut epithelial cells (*mex1, esg>btlRNAi*). In contrast to the tracheal manipulations, this had

no effect on gut tracheation or shape (Extended Data Fig. 5c,d,g,h and Supplementary Tables 28 and 29).

Finally, we also conducted laser ablations of the tracheal branches which span adjacent midgut regions in dissected guts ex vivo. We observed ablation-induced tracheal recoil in 22 out of 25 guts

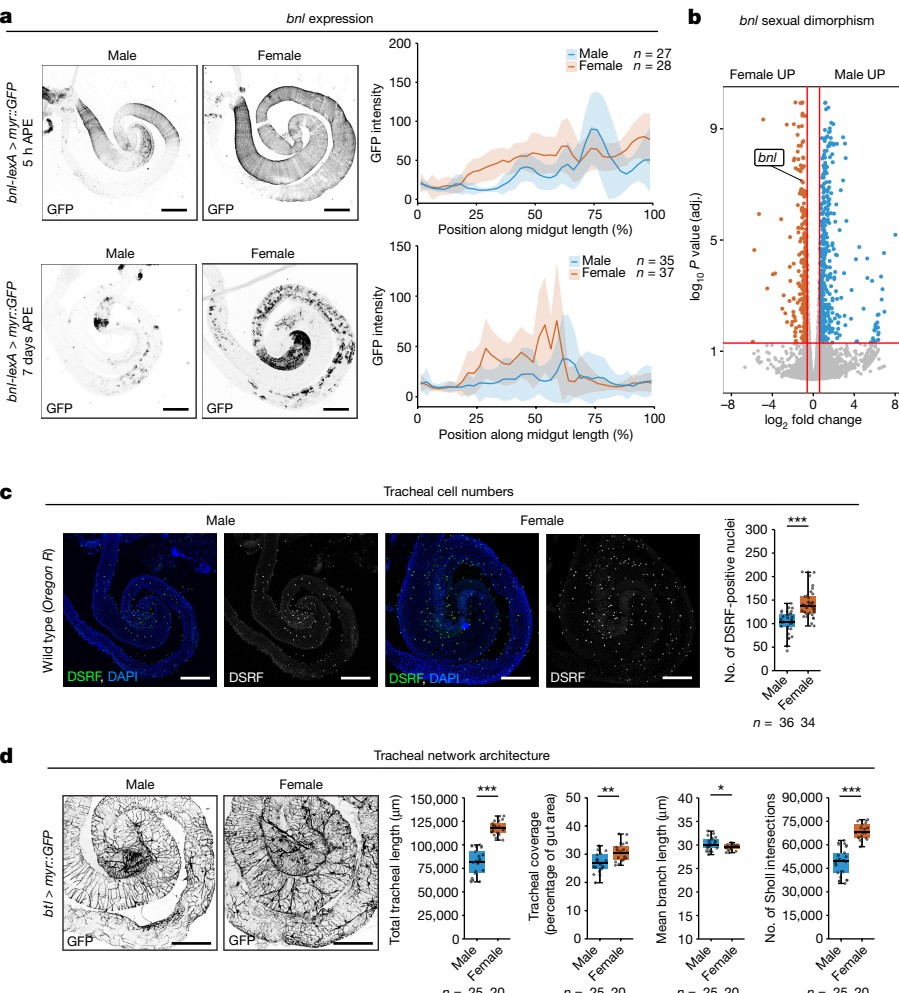

**Fig. 3 | Sex differences in tracheal branching and gut muscle-derived *bnl* expression. a**, The *bnl* expression in guts at 5 h (top) and 7 days (bottom) after pupal eclosion (APE), visualized by *bnl-LexA>myr::GFP* expression (left) and quantified along midgut length (right). **b**, RNA-seq profiling of male and female dissected midguts, visualized as the log₂-transformed fold change in expression between groups plotted against adjusted (adj.) *P* value (using raw transcriptomics data from ref. 17). Genes significantly upregulated (*P* < 0.05) in males and females are coloured in blue and orange, respectively. The *bnl* expression is significantly upregulated in females. **c**, Labelling of tracheal terminal cells by DSRF staining in *OregonR* flies shows difference in tracheal terminal cell number between male and female guts (quantified in right).

**d**, Trachea labelled with *btl>myr::GFP*, showing differences between males and females. Quantifications show higher total tracheal length, tracheal coverage by gut area and tracheal branching (number of Sholl intersections) in females compared to males. Males have longer mean tracheal branch lengths than females. Line graphs: mean and standard deviation. Boxplots: line, median; box, first quartile and third quartile; whiskers, minimum and maximum. *n* denotes number of biologically independent samples. Statistical significance was assessed using two-sided two-sample *t*-tests (**c**,**d**). *P* < 0.05; **P* < 0.01; ***P* < 0.001. See Supplementary Information for exact *P* values, statistical tests and sample sizes. Males, blue; females, orange. Scale bars, 200 μm.

(Supplementary Video 2 and data not shown), confirming that trachea hold tension, at least ex vivo.

Together, these experiments indicate that gut shape is actively maintained beyond development by an extrinsic tubular network which mechanically holds the loops together.

## Muscle–vessel crosstalk renders gut shape sexually dimorphic

The tracheal ablation experiments revealed a role for trachea in maintaining gut shape but a question remained as to how gut shape becomes sex-biased. Closer inspection of the intestinal tracheal system indicated that it is sexually dimorphic at several levels. Females have more terminal tracheal cells on their midgut, as shown by quantifications of two independent nuclear markers (DSRF staining and nuclear GFP expressed from *btl-GAL4*) (Fig. 3c and Extended Data Fig. 6a). Four independent membrane labels (membrane reporters expressed from

*btl-GAL4, QF6, DSRF-GAL4* and *trh-GAL4*; Fig. 3d and Extended Data Fig. 6b) further indicate that terminal tracheal cells are also more highly branched in females. The scarcer branches of males are slightly longer than those of females, potentially suggestive of a tiling/contact inhibition mechanism (Fig. 3d and Extended Data Fig. 6b) but female guts have increased tracheal coverage, even after accounting for their larger size (Fig. 3d). (Of note, we have observed that gut tracheation is reduced in the *white* genetic background, commonly used as a 'wild-type' background, relative to truly wild-type flies. The sexual dimorphism in all these features was nonetheless apparent in both genetic backgrounds (Fig. 3c and Extended Data Fig. 6a) but we routinely ensure that all experimental flies are matched to at least one control with regard to the presence/absence of the *white* gene for this reason).

Might sex differences in tracheation extrinsically impart sex differences to gut shape? We first proposed that the sex of the trachea might be cell-intrinsically controlled by the sex-determination pathway; most sex differences in *Drosophila* result from sex-chromosome-dependent,

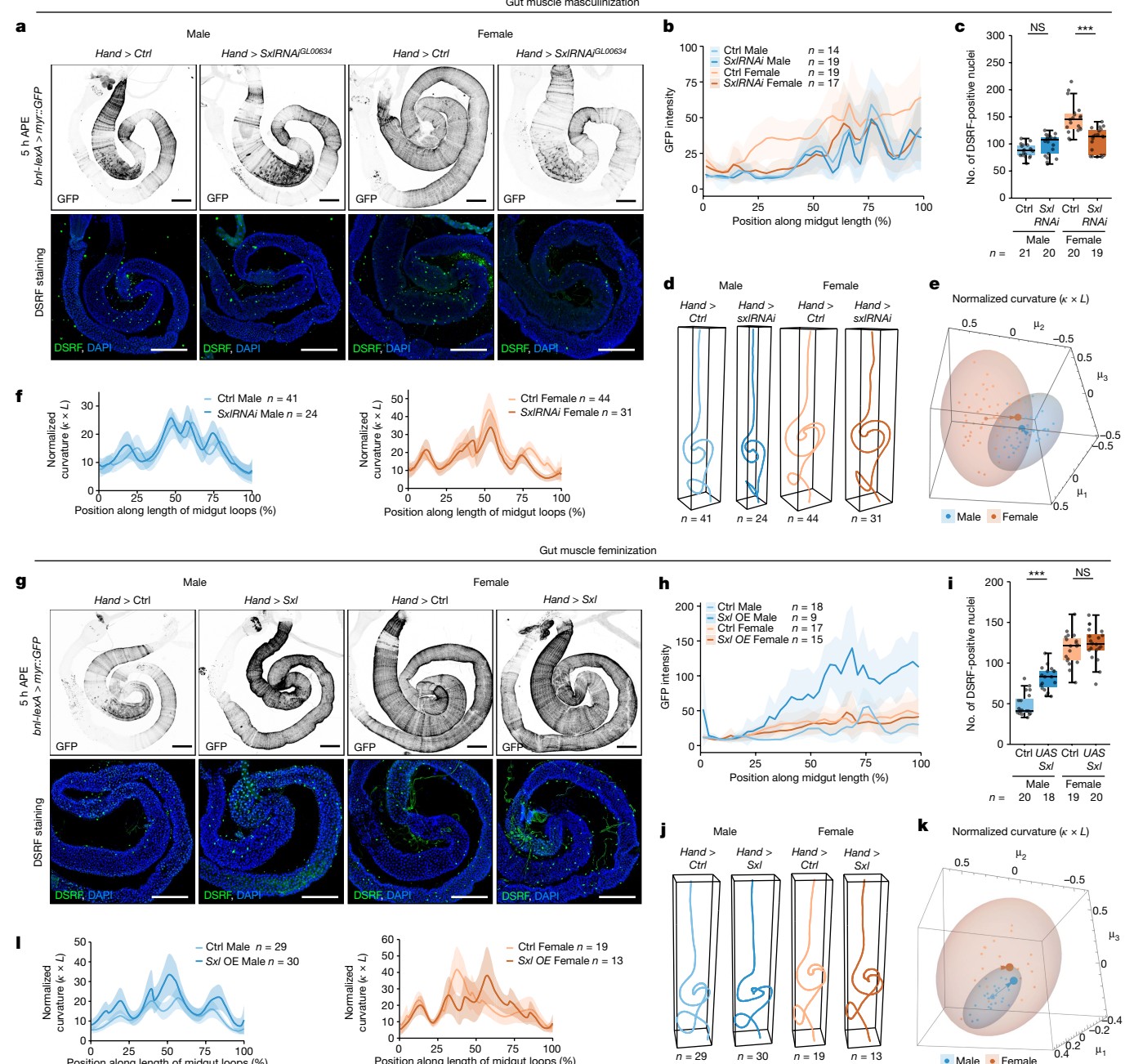

**Fig. 4 | Sex reversals of intestinal muscles impact tracheal branching and gut shape. a–c**, *Hand>SxlRNAi^{GLO034}* masculinizes *bnl* expression at 5 h APE, seen by *bnl-lexA>myr::GFP* (top of **a**, quantified in **b**) and reduces the number of DSRF-positive tracheal terminal cells in females (bottom of **a**, quantified in **c**). **d**, Average centrelines of *Hand>SxlRNAi* guts, showing altered shape in central midgut region of females relative to controls. **e**, Multidimensional scaling plot showing change in gut curvature normalized by gut length for *Hand>SxlRNAi* females compared to controls and no change in males. **f**, Female *Hand>SxlRNAi* show reduced average gut curvature normalized by gut length relative to controls. **g–i**, *Hand>Sxl* feminizes *bnl* expression at 5 h APE, seen by *bnl-lexA>myr::GFP* (top of **g**, quantified in **h**) and increases the number of DSRF-positive tracheal terminal cells in males (bottom of **g**, quantified in **i**). **j**, Average centrelines of *Hand>Sxl* guts showing altered shape in central

midgut region of males relative to controls. **k**, Multidimensional scaling plot showing change in gut curvature normalized by gut length for *Hand>Sxl* males to controls and no change in females. **l**, Male *Hand>Sxl* show increased average gut curvature normalized by gut length. Line graphs: mean and standard deviation. Boxplots: line, median; box, first quartile and third quartile; whiskers, minimum and maximum. Multidimensional scaling plots: ellipsoids represent 95% confidence space for each group, arrows represent shift in the mean from control to experimental manipulation. *n* = number of biologically independent samples. Statistical significance in **c** and **i** was assessed using one-way ANOVA followed by Tukey post hoc tests. NS, not significant ($P > 0.05$). \*\*\*$P < 0.001$. See Supplementary Information for exact *P* values, statistical tests and sample sizes. Males, blue; females, orange; controls, lighter matching colours (see genotypes in Supplementary Information). Scale bars, 200 μm.

female-specific expression of the Sex lethal (Sxl) RNA-binding protein. Sxl induces female-specific alternative splicing of the *transformer* (*tra*) gene, leading to a functional, female fate-determining Tra protein (TraF) only in females[32–39]. Having detected TraF expression in female

tracheal cells (data not shown), we sought to masculinize tracheal cells in females by downregulating *tra*. However, and in contrast to its masculinizing effects on other cell types such as gut stem cells[17], tracheal-specific downregulation of *tra* using two independent RNAi

lines failed to masculinize terminal tracheal cell number and we observed no or inconsistent masculinization of tracheal branching or gut shape (Extended Data Fig. 7a–g and Supplementary Tables 30 and 31).

Our previous transcriptomics experiments[17] had suggested sex-biased expression of the gut-derived Bnl ligand which promotes tracheal growth (Fig. 3b). We confirmed and extended this observation using our *bnl* reporters; *bnl* expression is strongly female-biased, first in gut muscles after pupation and in enterocytes thereafter (Fig. 3a and Extended Data Fig. 4a,c). Because of the effects on tracheal survival observed following complete depletion of gut muscle-derived Bnl (Fig. 2h and Extended Data Fig. 3e), we wondered whether the sex differences in gut-derived Bnl may extrinsically sculpt sex differences in the tracheal network. To test this, we sought to masculinize—rather than totally deplete as in our previous experiment—*bnl* expression in gut muscles. Gut muscle-specific downregulation of *tra* or its upstream regulator *Sex lethal* (*Sxl*) had no effects in male flies but effectively masculinized *bnl* expression levels in the gut muscles of female flies (Fig. 4a,b and Extended Data Fig. 8a,b). Female flies with masculinized *bnl* expression had a masculinized (reduced) gut tracheal network, both at the level of terminal tracheal cell number and tracheal branching (Fig. 4a,c and Extended Data Fig. 8a,c). As expected, this manipulation had no effect on the tracheal network of male flies (Fig. 4a,c and Extended Data Fig. 8a,c). Concurrent with these tracheal masculinizations, gut shape was also specifically affected in female flies with *SxlRNAi*- or *traRNAi*-driven masculinized *bnl* expression: reduced curvature was apparent in the midgut loops of female but not male flies (Fig. 4d–f, Extended Data Fig. 8d–i and Supplementary Tables 32–35). Conversely, ectopic expression of the female determinant Sxl in gut muscles had no effect on female flies but resulted in feminization (increase) of all these three features in male flies: *bnl* expression in the gut muscles, terminal tracheal cell number and tracheal branching (Fig. 4g–i). Accordingly, gut shape was specifically affected in male flies (Fig. 4j–l, Extended Data Fig. 8j,k and Supplementary Tables 36 and 37).

Together, these experiments show sex differences in the genetic and mechanical crosstalk between the gut and its trachea: gut muscles render trachea sexually dimorphic through their sex differences in Bnl expression levels. In turn, the tracheal sexual dimorphism maintains gut shape in a male or female 3D configuration.

## Gut–trachea coupling is physiologically significant

Finally, we explored the physiological significance of the coupling between gut and trachea. We first analysed flies with specific loss of intestinal trachea. This was achieved by gut muscle-specific downregulation of the FGF ligand Bnl, *Hand>bnlRNAi*: the genetic manipulation which more specifically affects gut trachea without affecting the gut itself (gut length is unaffected in males and only modestly affected in females) or other trachea in the fly (as would be the case for the tracheal ablation, *trh^TS>Bax*). Notably, even in the presence of trachea in control animals, expression of hypoxia reporters is higher in the midgut relative to other organs (Extended Data Fig. 9a). Gut tracheal ablation failed to further upregulate expression of these hypoxia reporters (Fig. 5a,b and Extended Data Fig. 9b), arguing against oxygen delivery being the main role of intestinal trachea in normal, homoeostatic conditions. Consistent with this idea, other intestinal features such as transit, excretion and intestinal stem cell proliferation were largely unaffected in flies lacking gut trachea during normal homeostasis (Extended Data Fig. 10a,e,f). Absence of gut trachea does, however, impact whole-body physiology, particularly in females and in response to challenges. Specifically, it differentially impacts two hyperproliferative responses: on the one hand, it increases age-induced hyperproliferation[40] in the intestinal epithelium (Fig. 5c). By contrast, it reduces damage-induced intestinal proliferation in flies fed a detergent (dextran sulfate sodium

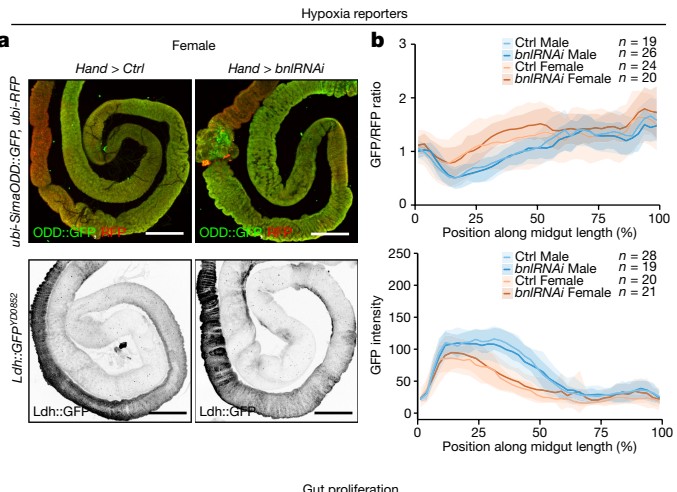

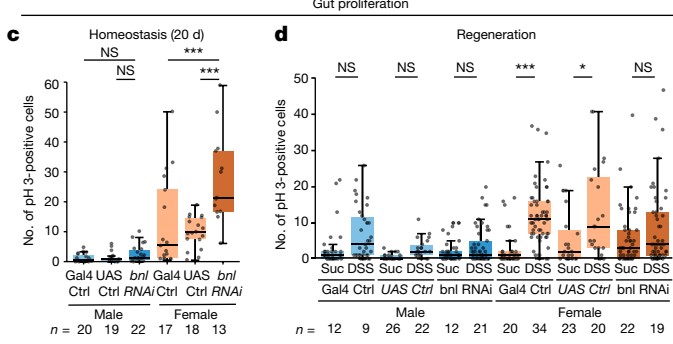

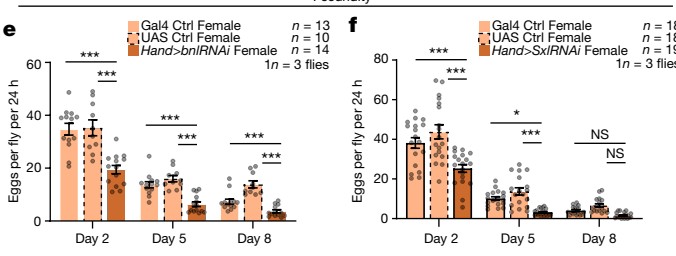

**Fig. 5 | Physiological importance of gut trachea. a,b**, Expression of *bnlRNAi* from gut muscle (*Hand>bnlRNAi*) does not change amounts of SimaODD::GFP (top of **a**, quantified in top of **b**) nor Ldh::GFP (bottom of **a**, quantified in bottom of **b**). **c**, Expression of *bnlRNAi* from gut muscles (*Hand>bnlRNAi*) increases number of mitoses in 20-day-old flies, seen by number of pH 3-positive cells. **d**, *Hand>bnlRNAi* expression in DSS-treated guts reduces mitotic indices in midgut of 7-day-old females relative to DSS-treated controls, seen by number of pH 3-positive cells. **e,f**, Expression of *Hand>bnlRNAi* (**e**) or *Hand>SxlRNAi* (**f**) reduces fecundity in females relative to controls, as measured by the number of laid eggs per fly on days 2, 5 and 8 after mating. GFP intensity and GFP/RFP ratio graphs show mean and standard deviation. Boxplots: line, median; box, first quartile and third quartile; whiskers, minimum and maximum. *n* denotes number of biologically independent samples. Statistical significance was assessed using one-way ANOVA followed by Tukey post hoc tests (**c**–**f**). \**P* < 0.05; \*\**P* < 0.01; \*\*\**P* < 0.001. See Supplementary Information for exact *P* values, statistical tests and sample sizes. Males, blue; females, orange; controls, lighter matching colours (see genotypes in Supplementary Information). Scale bars, 200 μm.

(DSS))/sucrose mixture[41] relative to the baseline intestinal proliferation observed in sucrose-fed flies (Fig. 5d). And although it does not affect food intake (Extended Data Fig. 10b–d), it leads to reduced ability to withstand starvation in both sexes (Extended Data Fig. 10g,h) and blunts reproductive output in females (Fig. 5e): a phenotype which was also apparent in female flies with masculinized gut muscles (*Hand>SxlRNAi*) (Fig. 5f).

## Discussion

We have described and genetically interrogated a previously unrecognized level of organization: the stereotypical yet sexually dimorphic spatial arrangement of internal organs. Our findings provide an example of bidirectional muscle–vessel mechanochemical communication which renders organ shape different between the sexes. Organ networks, such as the insect trachea or the vertebrate vasculature, reach virtually all other tissues in the body. Hence, sex differences in their anatomy and/or activity could result in sex differences in the anatomy and function of many—perhaps all—of their target cells and organs. In this regard, human somatic organs show anatomical sexual dimorphisms which cannot always be explained by the male–female difference in body size[42]. Hence, alongside intrinsic sex chromosome and extrinsic sex hormone effects, possible contributions of 'vascular sex' to the sex of other somatic organs deserves further investigation.

Tracheal branches are known to be remodelled by nutrient scarcity, infection and tumourigenesis[28,30,43]. In light of the increased intestinal stem cell proliferation we have observed in aged flies lacking gut trachea, possible remodelling and contributions of trachea to age-related intestinal dysplasia deserve further investigation, particularly given its known female bias[44–47]. Extrinsic, trachea-like mechanisms also instruct gut looping during embryogenesis in both flies and vertebrates[48–52]. A case in point—and one particularly reminiscent of gut–tracheal contacts—is the physical mechanism enabled by the attachment of the gut tube to the dorsal mesentery in vertebrates and driven by the differential growth of these two tissues[48,50,52]. Our findings raise the possibility that the shape and function of vertebrate guts remain sensitive to the plasticity of tethering systems beyond development, providing one reason why gut shape may need to be extrinsically controlled. Differences in these tethering systems (trachea, vasculature and mesentery) could also account for differences in organ shape between related species.

Why do trachea make the gut loop in a male or female way? Although trachea are the respiratory system of insects, our results argue against an oxygen-delivering role for gut trachea in homoeostatic conditions. Instead, in the absence of gut trachea, we find that the gut changes its shape and female fecundity is reduced. We suggest that trachea maintain (or, when required, change) the 3D configuration of organs to enable or constrain paracrine and/or contact-based exchange of peptide 'hormones', metabolites and/or mechanical cues in or between them. If secreted, signals could be confined spatially by the fact that adult haemolymph is very viscous and, in some areas of the body cavity, must pass through very narrow spaces between organs[53–55], potentially rendering insect circulation less 'open' than once thought, at least in adults. Proximity-enabled, spatially confined communication could help explain paradoxes which have emerged from the study of 'systemic' signals. Molecules such as cytokines, amino acids or insulin-like peptides are reported to relay different information across different organs[20,43,56–61]; how do target organs know where these 'promiscuous' signals come from and what response they are meant to elicit? We propose that the same signal can be used to convey a different message between organs A and B and between organs A and C because C is never in reach of A.

This logic might be relevant beyond animals with an open circulation: the position of specific organs or organ portions relative to the direction of circulation and/or innervation could, in some cases, restrict their ability to communicate. A paradigm in which to explore this idea is provided by the physiological connections between the mammalian intestine and neighbouring organs, such as the pancreas; these involve both secreted signals and direct innervation across organs that bypasses the central nervous system[62–65]. Local interactions might also be significant in disease or therapy: might some of the benefits of certain types of bariatric surgery, in which the connection of specific gastrointestinal tract regions is surgically altered, result

from altered gastrointestinal geometry? Similarly, disorders such as inflammatory bowel disease or colorectal cancer have regional and/or sex-biased incidence for reasons that are not fully understood[66–69]. By considering organ and interorgan geometry in 3D, we hope that specific features of organ shape and/or position can help diagnose gastrointestinal disorders or even predict them ahead of their clinical manifestation.

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

## Methods

### Fly husbandry

Flies were raised on a standard cornmeal/agar diet (6.65% cornmeal, 7.15% dextrose, 5% yeast, 0.66% agar supplemented with 2.2% nipagin and 3.4 ml l$^{-1}$ of propionic acid). All experiments were conducted at 25 °C, 65% humidity and on a 12 h light/dark cycle, unless otherwise stated. Flies were virgin and aged to 5 h or 7 days after eclosion for experiments, unless otherwise stated. Experimental and control flies were raised in identical conditions and processed at the same time. For example, for dissections, experimental and control flies, males and females were dissected and processed at the same time on the same slide. For microCT, they were fixed at the same time, mounted in the same tube and scanned at the same time in each batch.

### Temperature-sensitive experiments

For expression of *UAS-Bax* to trigger apoptosis, flies were raised at 18 °C for 7 days after eclosion and then transferred to 29 °C for 3 days for transgene induction.

For expression of *UAS-bnlRNAi* in pupal stages and early adults, larvae were raised at 18 °C and shifted to 29 °C within the first 20 h of pupal formation and until 7 days after pupal eclosion.

For expression of *UAS-btlRNAi* in the gut epithelium or *UAS-traRNAi* in trachea, larvae were raised at 18 °C and shifted to 29 °C within the first 20 h of pupal formation and until 7 days after pupal eclosion.

### Fly stocks

The following fly stocks were used in this study: *Hand-Gal4[MIO4106-TG4.0]* (BDSC 66795), *mex1-Gal4* (ref. 70), *esg-Gal4* (ref. 71; NP7397), *btl-Gal4* (ref. 72; DGGR 109128), *trh-Gal4* (ref. 25; GMR14D03, BDSC 47463), *vm-Gal4* (ref. 25; GMR13B09, BDSC 48547), *DSRF-Gal4* (ref. 73; BDSC 25753), *bnl-Gal4[MIO0874-TG4.1]* (this study, see below for details), *bnl[lexA]* (a gift from S. Roy; ref. 31), *QF6* (a gift from J. Cordero; ref. 74), *UAS-traRNAi.TRiPJF03132* (BDSC 28512), *UAS-traRNAi.GD764* (VDRC 2560), *UAS-SxlRNAi.TRiPGL00634* (BDSC 38195), *UAS-bnlRNAi.GD3070* (VDRC 5730), *UAS-btlRNAi.KK100331* (VDRC 110277), *UAS-Bax* (a gift from J. Cordero; ref. 75), *UAS-myr(src)::GFP M7E* (BDSC 5432), *UAS-StingerGFP* (ref. 76; BDSC 84278), *UAS-Flybow.1.1B* (used as *10xUAS-CD8::GFP;*ref. 77; BDSC 56803), *QUAS-mtdTomato-3xHA* (ref. 74; BDSC 30005), *13xlexAop2-IVS-myr::GFP* (ref. 78; BDSC 32209), *OregonR* (ref. 79), *w$^{1118}$* (GD control; VDRC 60000), *UAS-mCherryRNAi.Valium10* (TRiP control; BDSC 35787), *ovo$^{D1}$* (BDSC 1309), *UAS-Dcr-2* (BDSC 24646 and 24650), *UAS-Gal80$^{TS}$* (ref. 80; BDSC 7108), *UASp-Sxl.alt5-C8* (used as *UAS-Sxl*; ref. 81; BDSC 58484), *Ubi-EGFP.ODD, Ubi-mRFP.nls* (ref. 82; BDSC 86536) and *Ldh::GFP$^{YD0852}$* (a gift from U. Banerjee; ref. 83).

The *bnl-Gal4* line was generated through integration of a promoter-less *T2A-Gal4* transgene into the MiMIC insertion *bnl[MIO0874]* through recombination-mediated cassette exchange, as described in the Trojan-MiMIC technique[84]. Like *bnl[lexA]* (ref. 31), this reporter results in a *bnl* mutation and is homozygous lethal; the *bnl* coding sequence is fused to *T2A-Gal4* after the first *bnl* exon resulting in a truncated protein after translation (schematic in Extended Data Fig. 4a). Unlike *bnl[lexA]*, however, our construct does not eliminate any endogenous genomic regions and the inserted *T2A-Gal4* is under the control of the endogenous *bnl* promoters/enhancers.

### Immunohistochemistry and tissue stainings

Adult guts were dissected in PBS and then transferred to PBS in a well drawn onto a poly-L-lysine-coated slide (Sigma, P1524) using hydrophobic silicone (Intek Adhesives, Flowsil). Guts were fixed at room temperature for 20 min with 4% formaldehyde in PBS. All washes were done with PBS-T (PBS, 0.2% Triton X-100) following standard protocols. Primary antibodies were incubated overnight at 4 °C and secondary antibodies were incubated at 4 °C for 2–3 h. The following primary antibodies were used: mouse anti-DSRF 1:1,000 (Active Motif, 39093),

goat anti-GFP 1:1,000 (Abcam, ab5450), rabbit anti-mCherry 1:1,000 (Abcam, ab167453), mouse anti-Prospero 1:1,000 (DSHB, MR1A) and anti-horseradish peroxidase (HRP) rhodamine (TRITC)-conjugated 1:500 (Jackson ImmunoResearch, 123-025-021). The following fluorescent secondary antibodies were used: anti-rabbit FITC-conjugated (Jackson ImmunoResearch, 711-97-003), anti-mouse Cy3-conjugated (Jackson ImmunoResearch, 715-166-150), anti-mouse Cy5-conjugated (Jackson ImmunoResearch, 715-175-151) and anti-goat FITC-conjugated (Jackson ImmunoResearch, 112-095-044) and were diluted 1:500. Guts were mounted in Vectashield with DAPI (Vector laboratories).

### Confocal microscopy

Fluorescent images were taken on a Leica SP5 confocal microscope (1.5152 μm pixel size, 8-bit, 1,024 × 1,024 pixels) or a Leica SP8 DLS confocal microscope (1.4127 μm pixel size, 8-bit, 1,024 × 1,024 pixels) with a ×10 objective and using standard PMT detectors. *Z*-stacks were acquired with *z*-step size of 5 μm.

### MicroCT scans

Adult flies were prepared for microCT using a modified version of a previously described protocol[12]. Flies were anaesthetized with $CO_2$ and transferred to an Eppendorf tube with PBS-T (PBS, 0.5% Triton X-100) for 5 min or until all the flies had sunk to the bottom of the tube. Flies were then fixed in Bouin's fixative (Sigma, HT10132) for 16–24 h before being washed in PBS for a day with several solution changes. Flies were then stained in 1:1 Lugol's solution (Sigma, 62650):water for 4 days. Flies were washed once in water and then mounted in p10 pipette tips as follows: two p10 pipette tips were filled with water and the small opening sealed with parafilm. About ten flies were placed end to end inside each tip and the tips were stacked by inserting the tip of one into the open end of the other and then sealing with parafilm. The relative homogeneity and symmetry of *Drosophila* samples allowed us to mount two such tip stacks side by side to double imaging throughput and still retain sufficient contrast to resolve organ structures. This allowed us to mount and scan around 40 flies per scanning session, with each p10 tip containing about 10 flies (Extended Data Fig. 1a). Flies were imaged on the following scanners with the following settings. Zeiss Xradia Versa 510: 40 kV, 75 μA, 3 W, 2.95 pixel size, 0.45 rotation step (801 projection images), LE1 filter, ×4 objective. Bruker Skyscan 1272: 40 kV, 110 μA, 4 W, CMOS camera scanning at a 2.95 μm pixel size, 0.3–0.35 rotation step, 30 μm random movement and four frame averaging. Bruker SkyScan 1172 with a 11 MP CCD detector: 40 kV, 250 μA, 10 W, 2.49 μm pixel size, 0.4 rotation step (479 projection images), 10 μm random movement and four frame averaging. For most experiments, all flies in an experiment were scanned with the same scanner. When several scanners were used, a batch factor was applied in the analysis to control for any potential differences. Images were reconstructed using the Zeiss Reconstructor software v.11 or the Bruker NRecon software, then background was subtracted and images were Gaussian smoothed in FIJI v.2.0.0-rc-69/1.52p.

### RNA-seq data

RNA-seq data were generated as previously described[17]. RNA was extracted from three samples of 30 pooled dissected guts from each sex from wild-type flies: *w; Su(H)GBE-LacZ/w; esg-Gal4 NP7397, UAS-GFP, Tub-Gal80TS/+*. Data visualization was produced with R (v.4.2.1)[85] using a standard volcano plot script.

### Tracheal ablations

Sets of three to five guts were dissected from *btl>myr::GFP*-expressing virgin female flies and lightly attached onto a poly-L-lysine-coated glass bottom FluoroDish (WPI, FD35-100), containing haemolymph-like HL3 saline[86]. Guts were mounted unstretched to preserved their naturally coiled shape and avoid manual rupture of trachea spanning across gut

loops. Laser ablations were focused on trachea spanning across R2, R3 and R4 regions of the midgut. Imaging and ultraviolet-laser ablation of individual tracheal branches was done with a Nikon CSU-W1 SoRa spinning disk microscope, using the NIS-Elements software. Time-lapse recordings lasted between 30 s and 5 min after tracheal ablation.

## Starvation experiments

For the microCT experiments, 7-day-old adult flies raised on the standard cornmeal/agar diet were placed in vials containing 1% agar jelly and starved for 48 h, before processing for imaging.

To assess resistance to starvation, groups of 30–35 virgin flies were transferred to vials containing 1% agar jelly and death events were recorded three or four times a day from 08:00 to 20:00, until all flies had died. Flies were transferred to fresh vials containing the same medium every 3 days during this process. Survival curves were obtained using the Kaplan–Meir estimate and the difference between curves was assessed using the log-rank Mantel–Cox test, using the GraphPad Prism (v.9.4.1) software.

## Fecundity

To assess fecundity, virgin female flies were placed with males for about 20 h for mating. Groups of three mated female flies were placed in vials containing dark media for contrast during egg counting, consisting of 5% of sucrose, 10% autolysed yeast and 1% agar. Flies were allowed to lay eggs for 24 h. Eggs were counted at days 2, 5 and 8 after mating under microscope.

## Quantifications

**Tracheal cell numbers.** DSRF-positive and StingerGFP-positive nuclei were counted in FIJI on maximum intensity projections of confocal stacks manually with the help of Cell Counter for keeping track of counted nuclei. Malpighian tubules and hindgut regions were excluded from these quantifications.

**Tracheal filament length and branching.** Tracheal filament 3D reconstruction and quantification was performed using Imaris x64 v.9.9.0 (RRID:SCR_007370) using the Filament Tracer and Batch packages (RRID:SCR_007366). Using the surfaces tool, a mask was applied to the tracheal signal channel to reduce signal background before segmentation. The filaments tool was applied using the autopath algorithm to segment all filaments between 2 and 30 μm in diameter. The batch package was used to apply the same settings to a set of images acquired at the same time and from the same microscope slide. The 'sum of filament lengths' was taken as the total tracheal length, 'dendrite mean length' was taken as the mean tracheal branch length and 'filament number of Sholl intersections' was taken as a proxy measurement of tracheal branching. Sholl analysis measures the average number of filament intersections on concentric spheres spaced at 1 μm diameters. Tracheal coverage was measured in FIJI, by segmenting trachea area using autothreshold from the *btl>myr::GFP* signal and representing this as a percentage of gut area. Gut area was measured in FIJI, using manual gut outlines obtained with the magnetic lasso tool in Adobe Photoshop v.25.3.1.

**Measurement of intensity along gut length.** The intensity of myr::GFP or Stinger::GFP driven by *bnl-lexA*, *bnl-Gal4* or *btl-Gal4* or the intensity of *ODD::GFP*, *nls::RFP* and *ldh::GFP*, was measured along the midgut length in FIJI from *z*-stacks projected using maximum intensity. Measurements were taken along a 30 pixel-wide line drawn manually using the freehand line tool through the centre of the gut tube along gut length. A landmark was manually placed in the centre of the R3 region and its *x,y* coordinate extracted. Gut length was adjusted relative to the position of the R3 landmark to give percentage position along gut length with R3 aligned between sexes at 50% gut length. In R v.3.6.0, intensity values for several flies were binned into 40 bins along

gut length and the mean and standard deviation found for each bin. Code is available on GitHub through Zenodo (https://doi.org/10.5281/zenodo.10905446)[87].

**Food intake.** To assay the amount of food ingested, we used the standard cornmeal/agar diet supplemented with 1% FCF Blue (Sigma, 80717). For analysis of feeding ad libitum, flies were transferred from the standard diet to the 1% FCF Blue-supplemented diet and allowed to feed for 4 h. For analysis of feeding after starvation, flies were starved for 16 h in vials containing 1% agar jelly and then transferred to the 1% FCF Blue-supplemented diet and allowed to feed for 15 min. Fed flies were frozen in liquid nitrogen and transferred in groups of three to 2 ml round bottom microtubes with 0.5 ml of water and a 5 mm stainless-steel metal bead (QIAGEN, 69989). Fly tissues were homogenized using a QIAGEN TissueLyser II for 90 s at 30 Hz and the homogenates were cleared by centrifugation at 10,000*g* for 10 min. From each microtube, 0.2 ml of clear supernatant was transferred into a 96-well, flat-bottom, optically clear plate (Thermo Fisher Sterilin, 611F96). A BMG Labtech FLUOstar Omega plate reader was used to measure dye content by reading the absorbance at 629 nm.

FlyPAD assays were performed as previously described[20,88]. Half the electrode wells of a given flyPAD arena were filled with a pellet disc of cornmeal/agar diet, punched with a 1 ml pipette tip to the exact diameter of the inner electrode circle. The remaining electrode wells were left empty to record non-feeding baseline interactions. Flies were allowed to feed in the flyPAD arenas for 1 h, at 25 °C and 65% humidity. The Bonsai software was used to register capacitance and a MATLAB R2023b custom script was used to extract the total number of sips per fly during 1 h (ref. 89). All flyPAD experiments were performed at the same time of day between 10:00 and 13:00. Data for experimental and control genotypes used for comparison were always acquired in the same flyPAD assay.

**Intestinal transit.** To assess intestinal transit, groups of 30 virgin flies raised on standard cornmeal/agar diet were starved for 16 h in vials containing 1% agar jelly and then allowed to feed for 15 min in vials containing standard cornmeal/agar diet supplemented with 0.5% bromophenol blue (BPB) sodium salt (Sigma, B5525). Fed flies were quickly frozen in liquid nitrogen. Presence of dyed food in the whole gut versus stereotypically demarcated portions of the gut (midgut, hindgut or ampulla) was visually scored from dissected guts and these guts were mounted stretched on sticky poly-L-lysine-coated slides and lined side-to-side from left to right with reference to the order of dissection. Guts were imaged with a Leica MZ16 FA stereomicroscope and a Nikon DS-Fi3 camera. Gut length was measured using the freehand line tool in FIJI, drawn through the centre of each gut. To compare size-matched guts, we excluded guts from the test group that had length smaller than the mean − 1 s.d. of the control group. The effect of sex and genotype on the presence of food in the whole versus portions of the gut was statistically analysed by a logistic regression using the glm function in the VGAM package v.1.1 in R v.4.2.1.

**Intestinal excretion.** To assess intestinal excretion, groups of six virgin flies raised on standard cornmeal/agar diet were transferred to 5 mm clear plastic dishes, each containing a wedge of 0.5% BPB-supplemented food and allowed to feed and excrete for 60 h (refs. 90,91). Deposits left on the lids of the assay dishes were imaged using a transparency scanner (Epson Perfection V700) and quantification of the total amount of deposits was done using the T.U.R.D software[91].

**Gut proliferation.** Mitotic indices were quantified by manually counting phospho-histone H3-positive cells using a Nikon50i fluorescent microscope. These were quantified in young virgin flies at 7 days after pupal eclosion or in aged virgin flies at 20 days after pupal eclosion. For damage-induced regeneration assays, virgin flies were transferred

to an empty vial containing a piece of 3.75 × 2.5 cm$^2$ paper imbibed with 500 ml of 5% sucrose solution (control) or 5% sucrose plus 3% DSS solution. Flies were transferred to a new vial with fresh feeding paper every day for 3 days before gut dissection and quantification of mitotic indices.

**Segmentations.** ITK-snap (v.3.8.0)[92] was used to manually segment each of the organs. For ovaries and testes, the adaptive paintbrush tool was used. For the crop, the polygon tool was first used to segment the organ perimeter in every 20–30 slices in the axial plane, followed by use of the morphological interpolation tool to fill the spaces in between these presegmented slices[93]. We expanded this presegmented scaffold using an active contour model. For the gut, centreline traces (see below) were increased to a 5 pixel-wide line in FIJI and were imported into ITK-snap as seeds for the active contour model. For the crop and gut, the active contour model was run using the edge attraction mode with a smoothing factor of 2.5 and expansion (balloon) force, smoothing force (curvature) and edge attraction force (advection) were all set to maximums during the evolution of the model. Further manual corrections were performed using the adaptive paintbrush tool.

For visualization purposes, segmentations were converted into triangular meshes using the marching cubes algorithm run in FIJI with the Wavefront obj package. Using Meshlab (v.2020.07)[94], meshes were simplified using quadric edge collapse decimation to reduce the number of faces to 10% and smoothened using HC Laplacian smoothing[95].

Organ volumes were measured from segmentations in FIJI. The area of the segmented region was measured on each image slice, summed and multiplied by the slice depth to calculate the volume.

**Centreline tracing.** Centrelines of the gut tube were traced using the simple neurite tracer plugin (v.3.1.6)[96] in FIJI. Images were first inverted in intensity to make the centre of the gut of highest intensity for the simple neurite tracer algorithm to follow.

**Landmarks for defining midgut loops.** Landmarks were manually marked on the microCT stacks using the FIJI multipoint tool to extract their $x,y,z$ coordinates. Two landmarks were used—the distinction between the apical midgut and the midgut loops was defined as the first main inflection of the midgut, which generally correlated with the point where the midgut transitions from the thorax to the abdomen. The distinction between the midgut loops and the hindgut was defined as the transition between the midgut and the hindgut, easily recognizable morphologically in the microCT image stacks by a reduction in gut diameter and in X-ray absorbance. The $x,y,z$ coordinates of these landmarks were used to subset the centrelines to the midgut loop region before further processing.

**Geometric morphometrics and PCA analysis.** We performed morphometric analysis in R (v.3.6.0) using the geomorph package (v.3.2.1)[97–99]. Centreline data from simple neurite tracer were imported into R (v.3.6.0) using the nat package (v.1.8.18)[100] and divided into 1,000 (for whole gut centrelines) or 500 (for midgut loop region centrelines) equally spaced pseudolandmarks using the geomorph package. Landmark coordinates were then aligned using a generalized procrustes analysis (GPA) to standardize for size and orientation. For visualization of the average centrelines of a group of flies, corresponding GPA aligned landmark coordinates were averaged and then plotted in 3D.

Variation in gut shape was analysed using a principal component analysis (PCA) of the GPA aligned centreline coordinates. A Procrustes type III analysis of variance (ANOVA) with random residual permutation procedures (RRPP; RRPP package v.0.5.2)[98,99] was run to test whether variation in gut shape was significantly associated with variation in other factors using the procD.lm function. The 3D Procrustes aligned shape coordinates were set as the response variable and crop volume,

genital volume, gut length, imaging batch and sex were set as the predictor variables (shape ~ sex + gonad volume + crop volume + gut length + batch; Supplementary Tables 17–19). Batch was included to control for groups of flies scanned on different scanners or the same scanner at different times. For testing the differences between the male versus female and control versus experimental groups, a model with interaction terms was used: shape ~ genotype * sex + batch * genotype + batch * sex or shape ~ genotype * sex, when only one batch was present (Supplementary Tables 12, 13 and 20–37). Post hoc pairwise comparisons of Procrustes distances between least squares means and variances of the groups was then conducted using the pairwise function with RRPP[101] with shape ~ batch or shape ~ 1 as the null model where appropriate. Code is available on GitHub through Zenodo (https://doi.org/10.5281/zenodo.10905446)[87].

For all PCA displays, the diagrams at both ends of each principle component (PC) axis represent the extremes of variation along each PC: average shape in grey, theoretical maximum or minimum shape along each PC in black, as previously described[102]. For all displays in this study, the average coordinates of each Procrustes landmark along the gut were used to generate the average gut centreline in grey. The black lines represent the coordinates of the landmark points from the hypothetical extremes of variation of each PC.

**Measurements of gut length.** Gut length measurements were taken from centreline length measured using the nat package v.1.8.18 in R (v.3.6.0)[85]. Anterior midgut, midgut loops and hindgut measurements were taken from centrelines subsetted by landmarks as described above.

**Measurements of radius.** Gut segmentations were converted into triangular meshes using the marching cubes algorithm run in FIJI with the Wavefront obj package. Using Meshlab (v.2020.07)[94], meshes were simplified using quadric edge collapse decimation to reduce the number of faces to 10%. The radius is then estimated by finding the minimum distance between a given point on the centreline and the unsmoothed gut mesh of the segmentation. Repeating this for all points on the centreline gives the radius as a function of arclength. To smooth the radius as a function of arclength, the function LowpassFilter was implemented in Mathematica (v.13.1)[103] with a cutoff parameter of 0.3.

**Extraction of curvature and torsion along gut length.** To approximate the curvature and torsion along the centreline obtained from simple neurite tracer, the centrelines were first parameterized using an arclength coordinate, $s$, calculated for each point on the curve by summing the lengths of the line segments leading up to it, starting from the anterior. Value $s$ varies between 0 and the length of the centreline ($L$). The centreline in the vicinity of each point was approximated using a third-degree Taylor expansion of the curve position. For this, a neighbourhood of size $\delta = 0.05 \times L$ was chosen around a given point on the curve, which consists of points whose distance from the point of interest is less than $\delta$ and then it was fitted it to a cubic polynomial using the function polyfit implemented in the Python package NUMPY[104]:

$$\mathbf{x}(s) = \mathbf{x}^* + \mathbf{x}'(s^*)(s - s^*) + \frac{1}{2}\mathbf{x}''(s^*)(s - s^*)^2 + \frac{1}{6}\mathbf{x}'''(s^*)(s - s^*)^3$$

Here, $\mathbf{x}'(s^*), \mathbf{x}''(s^*), \mathbf{x}'''(s^*)$ are the first, second and third derivatives of the position with respect to $s$, evaluated at the point whose arclength is $s^*$. Local curvature $\kappa(s^*)$ and torsion $\tau(s^*)$ were computed, using the Frenet–Serret formulae adapted to our parameterization and assuming the curvature is locally uniform:

$$\kappa(s^*) = |\mathbf{x}''(s^*)|, \ \tau(s^*) = \frac{(\mathbf{x}'(s^*) \times \mathbf{x}''(s^*)) \cdot \mathbf{x}'''(s^*)}{\kappa^2(s^*)}$$

This was repeated for all points on the curve to obtain local approximations of the curvature and torsion. To further smooth the curvature and torsion as a function of arclength, the function LowpassFilter with a cutoff parameter of 0.3 was implemented in Mathematica. Curvature and torsion have units of inverse length and, therefore, are not invariant with respect to scale—for example, if we double the size of the centreline without changing its shape, the curvature and torsion will decrease by a factor of two. To produce scale invariant quantities, that depend on the shape of the centreline but not its size, the normalized curvature and torsion is defined by multiplying them by the total length of the centreline. Code is available on GitHub through Zenodo (https://doi.org/10.5281/zenodo.10905446)[87].

**Comparison of curvature using multidimensional scaling.** To compare the curvature of two different centrelines, they were first registered to know which point on the first centreline corresponded to a given point on the second[105]. For this, elastic distortion was minimized on the basis of the Fisher–Rao metric as described in refs. 106,107, which is implemented in the Python package scikit-fda[108].

Once this was known, the two centrelines were compared on a regional basis, by computing the total Euclidean distance between the local morphometric biomarker (such as normalized curvature ($\tilde{\kappa} \equiv L \times \kappa$)), at corresponding points and averaged over the entire centrelines. For example, if the normalized curvatures of the first and second centrelines are $\tilde{\kappa}_1(s_1)$ and $\tilde{\kappa}_2(s_2)$, where $s_1$ and $s_2$ are the respective arclength parameters, the registration is given as the function $s_2 = \gamma(s_1)$ and the distance between two centrelines can be computed using the formula:

$$\text{dist}(\tilde{\kappa}_1, \tilde{\kappa}_2) = \sqrt{\frac{1}{L} \int [\tilde{\kappa}_1(s_1) - \tilde{\kappa}_2(\gamma(s_1))]^2 ds_1}.$$

To discretize the curves, with equally spaced points in the coordinate $s_1$ (denoted as $s_n$, where $n = 1, 2, ...., 200$), the integral was replaced with a sum,

$$\text{dist}(\tilde{\kappa}_1, \tilde{\kappa}_2) = \sqrt{\frac{1}{200} \sum_{n=1}^{200} [\tilde{\kappa}_1(s_n) - \tilde{\kappa}_2(\gamma(s_n))]^2}$$

To calculate the relative distances between pairs of centrelines, the distances were divided by the maximum distance (across all pairs of centrelines in each analysis). Once the distance was computed for each pair of centrelines, a multidimensional scaling (MDS) algorithm was used, which converted the distances between the guts into a 3D coordinate for each gut ($\mu_1, \mu_2, \mu_1$) such that the distance between each pair of points in the MDS space is as close as possible to the original computed matrices. We used the following reference to convert our distances into MDS coordinates[109]. Once the coordinates were obtained for each group, the region occupied in the MDS space was estimated by fitting a normal distribution to the points and drawing the 95% confidence intervals.

Lastly, to test the location change of the region centrelines with experimental manipulations, the LocationTest function in Mathematica was used to compute the $P$ value from several applicable tests ($t$-test, paired sample $t$-test, $Z$-test, paired sample $Z$-test, Mann–Whitney $U$-test, Sign test, Wilcoxon signed-rank test) and returned the $P$ value from the most powerful test (one with the highest probability of rejecting the null hypothesis) that applies to the data. Code is available on GitHub through Zenodo (https://doi.org/10.5281/zenodo.10905446)[87].

**Correlation of curvature and tracheal intensity.** Average tracheal intensity along gut length was measured as described above, for the midgut not including the hindgut of *btl>myrGFP* male and female flies and correlated with the average curvature along length for *OregonR* male and female flies of the equivalent gut region. Pearson's product moment was calculated.

**Correlation of curvature and *bnl* intensity.** Males and females were analysed separately. Curvature and *bnl* intensity were normalized so that their range is [−1, 1] for each gut to allow for comparison across them. For example, if $I_{bnl}(s)$ is the measured intensity as a function of arclength for a single fly, the corresponding normalized intensity will be

$$\tilde{I} \equiv \frac{I_{bnl} - \min(I_{bnl})}{\max(I_{bnl}) - \min(I_{bnl})}$$

with a similar expression for the centreline curvature. Each centreline curvature was paired with a *bnl* intensity curve from a fly of the same sex and an elastic registration was performed between them as mentioned above and then the Pearson correlation coefficient was computed. For the females, there are 39 measured centreline curvatures and 28 *bnl* intensity curves, leading to 39 × 28 = 1,092 correlation coefficients whose values are given in the orange histogram in Extended Data Fig. 4b. Similarly, for the males there are 41 measured centreline curvature and 27 *bnl* intensity curves, leading to 41 × 27 = 1,107 correlation coefficients whose values are given in the blue histogram in Extended Data Fig. 4b.

To obtain a control for the measured histogram, *bnl* intensity curves were simulated by fitting the actual *bnl* intensity curves to an autoregressive stochastic process (using the command ARProcess in Mathematica). Repeating the analysis above leads to the histograms shown in grey in Extended Data Fig. 4b. Code is available on GitHub through Zenodo (https://doi.org/10.5281/zenodo.10905446)[87].

**Tilt.** To estimate the tilt of the midgut loop region relative to the entire gut centreline, the main axis of each gut was calculated as the largest eigendirection of the covariance matrix. The covariance matrix $C$ for a given set of points in 3D, where each point is labelled by the index $i$ and position vector $\mathbf{x}_i$, is given by

$$C = \frac{1}{n-1} \sum_{i=1}^{n} (\mathbf{x}_i - \bar{\mathbf{x}})^T (\mathbf{x}_i - \bar{\mathbf{x}})$$

where $n$ is the total number of points and $\bar{\mathbf{x}}$ is the average position of all the points. The main axis of the midgut loop region is denoted by $\mathbf{V}_m$ and the entire gut by $\mathbf{V}_g$; the angle between them was determined as $\varphi = \cos^{-1}(\mathbf{V}_g \cdot \mathbf{V}_m)/(|V_g||V_m|)$.

**Measurements of proximity.** The proximity between meshes was measured in R v.3.6.0. All surface meshes from the organ segmentations were simplified using quadric edge collapse decimation in Meshlab (v.2020.07)[94] to reduce the number of faces to 1%, other than the testes apical tip meshes which contained few faces so were instead reduced to 10%. Mesh vertex coordinates were then read into R and the minimal distance between each vertex on organX and all the vertices on organY was calculated using a nearest neighbour algorithm using the RANN package (v.2.6.1)[110]. For reference to the midgut loops of the gut or for plotting along gut length, the centreline coordinates were replotted as 100 equally spaced points using the nat package (v.1.8.18)[100]. The centreline is then related to the gut mesh by finding the nearest 20 vertices on the mesh for every centreline point. The minimum distance of these 20 vertices to organY is then assigned to the centreline point for averaging and for plotting. For restricting to midgut loops, the landmarks were used to cut the centreline. Code is available on GitHub through Zenodo (https://doi.org/10.5281/zenodo.10905446)[87].

For visualization of proximities as shown in figures, the Hausdorff Distance function in Meshlab[94,111] was used which samples each vertex of meshX and finds the closest point on meshY to generate a minimal distance value between meshes for each vertex. These minimal distances were then displayed on the mesh as a heatmap in the Paraview software (v.5.10.0)[112].

**Crop duct quantifications.** The directions of the crop duct leaving the proventriculus and travelling through the thorax to enter the crop were manually recorded from viewing the microCT scans from several planes in ITK-snap. Four different configurations were recorded: passing from left to right in an s-shaped pattern, passing from left to right in a straight line, staying on one side of gut and inverted passing from right to left.

Position of the crop and contact it made with the ovaries was manually scored from viewing the microCT scans in several planes in ITK-snap.

### Experimental design and statistical analyses
For each experiment, a minimum of nine samples per group were examined per genotype or condition. Fly numbers are not limiting so no power calculations were used to predetermine sample size. Oversampling was mitigated by choosing sample sizes on the basis of previous knowledge of phenotypic variability in controls and other mutants. Similar sample sizes for different animal groups (for example, downregulations versus controls) were tested in the same experimental design. Exact sample sizes are provided in the Supplementary Information. Experimental and control flies were bred in identical conditions and were randomized whenever possible (for example, with regard to housing and position in tray). Control and experimental samples were processed at the same time and mounted on the same slides for confocal imaging or the same tips for microCT scanning. All replicates were biological rather than technical and all measurements were taken from distinct samples. Experiments were typically repeated two to three times and only those experiments for which repeats resulted in comparable outcomes are included in the manuscript. Experiments were controlled for sex, mating status, genotype, age and physiological state (for example, starved or ad libitum-fed). Details are provided elsewhere in the Methods and Supplementary Information. No data points or outliers were excluded from our experiments and blinding was performed for a subset of experiments. Quantification of DSRF stainings, filament tracing of fluorescently labelled trachea and quantifications of *bnl* expression along gut length was done on data blinded for genotype. Blinding for sex was not possible as this is visually obvious by differences in the length and diameter of the *Drosophila* gut. Similarly, blinding for sex was not possible for microCT scans as ovaries and testes were visible in the images.

All statistical analyses were carried out using R including use of 'dplyr' package (v.1.0.10). For multiple comparisons between groups, data were analysed using one-way ANOVA followed by a post-hoc TukeyHSD test. For single pairwise comparisons, we used Student's *t*-tests. Boxplots and line graphs were plotted in R using the 'ggplot2' package (v.3.4.0). For boxplots, the minimum, maximum, median, first quartile and third quartile are indicated with all data points shown as dots. In all figures, *n* denotes the number of biologically independent samples and $P$ values are indicated as asterisks highlighting the significance of comparisons (non-significant (NS): $P > 0.05$; *$P < 0.05$; **$P < 0.01$; ***$P < 0.001$). For Procrustes ANOVA, $P$ values are capped at a minimum of $P = 0.001$ as the RRPP procedure uses 1,000 iterations. Further information about sample size, $P$ values and statistical tests used for each experiment can be found in the Supplementary Information.

### Reporting summary
Further information on research design is available in the Nature Portfolio Reporting Summary linked to this article.

### Data availability
All reconstructed microCT scans, gut centreline files and organ segmentation files are available through Figshare (https://doi.org/10.25418/crick.25598859)[113]. All remaining data generated or analysed during this study are included in the Article (and its Extended Data and Supplementary Information). Further information can be

requested from the corresponding author. Source data are provided with this paper.

### Code availability
Code generated and used in this study is available on GitHub through Zenodo (https://doi.org/10.5281/zenodo.10905446)[87].

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

**Acknowledgements** We thank U. Banerjee, J. Casanova, J. Cordero and S. Roy for providing reagents. We are grateful to D. Dormann, C. Whilding and D. Matharu for imaging quantification advice, B. Lenhard for statistical advice and C. Amourda, H. Beckwith and L. Gartner for discussions or comments on the manuscript. N. Cakir, E. Elveren, I. Geraldes and B. Mizrak provided experimental assistance and A. Weston and L. Collinson supported us with our initial microCT scans. This work was funded by an ERC Advanced Grant (ERCAdG 787470 'IntraGutSex', a UKRI Frontier Research Guarantee grant EP/Y036298/1), MRC intramural funding and the Francis Crick Institute, which receives its core funding from Cancer Research UK (CC2258), the UK Medical Research Council (CC2258) and the Wellcome Trust (CC2258). T.S. was supported by grants from the National Heart, Lung and Blood Institute of the National Institutes Health (1K22HL137902-01) and an Institutional Development Award from the National Institute of General Medical Sciences of the National Institutes of Health under grant no. 2P20GM103432. A.M. was funded by an EMBO LTF (63-2017). S.M. and L.M. were partially funded by a DARPA MURI grant MINT and by the Simons Foundation. S.E.L. was supported by a Wellcome Trust grant no. 225439/Z/22/Z. Y.M. was supported by the MRC award MR/W027437/1, a Lister Institute Research Prize and the EMBO Young Investigator Programme.

**Author contributions** L.B. and P.G. performed most of the experiments including microCT scanning and reconstruction, quantification of tracheal cell numbers and branches and organ segmentations. L.B. also performed centreline and landmark extractions, wrote code for and performed geometric morphometric analysis, wrote code for and performed organ proximity analysis, wrote code for finding intensity along gut length, performed crop duct scoring, correlation of curvature with trachea intensity and measurement of organ volumes. P.G. also performed stem cell proliferation analyses, feeding and food transit assays and assisted with FlyPad feeding experiments, lifespan and fecundity analysis experiments. S.M. performed the mathematical analysis of curvature, torsion, radius, tilt and correlation of curvature with *bnl* intensity. O.L., L.K., Y.J., A.P.Z. and B.C. assisted with microCT scanning and organ segmentations. O.L. also performed starvation and fecundity experiments. A.M. performed stem cell proliferation analyses and contributed muscle driver stainings. B.S. conducted FlyPad assays. T.A. assessed the sex determination pathway in tracheal cells. S.E.L. and Y.M. assisted with the ex vivo tracheal ablations. L.P.-G. and T.S. supported and conducted our pilot microCT scans, respectively. M.V. assisted with organ proximity analysis. L.M. supervised the mathematical analysis of gut shape features. I.M.-A. conceived, coordinated and supervised the project. L.B., P.G. and I.M.-A. wrote the paper, with input from all the authors.

**Funding** Open Access funding provided by The Francis Crick Institute.

**Competing interests** The authors declare no competing interests.

**Additional information**
**Correspondence and requests for materials** should be addressed to Irene Miguel-Aliaga.

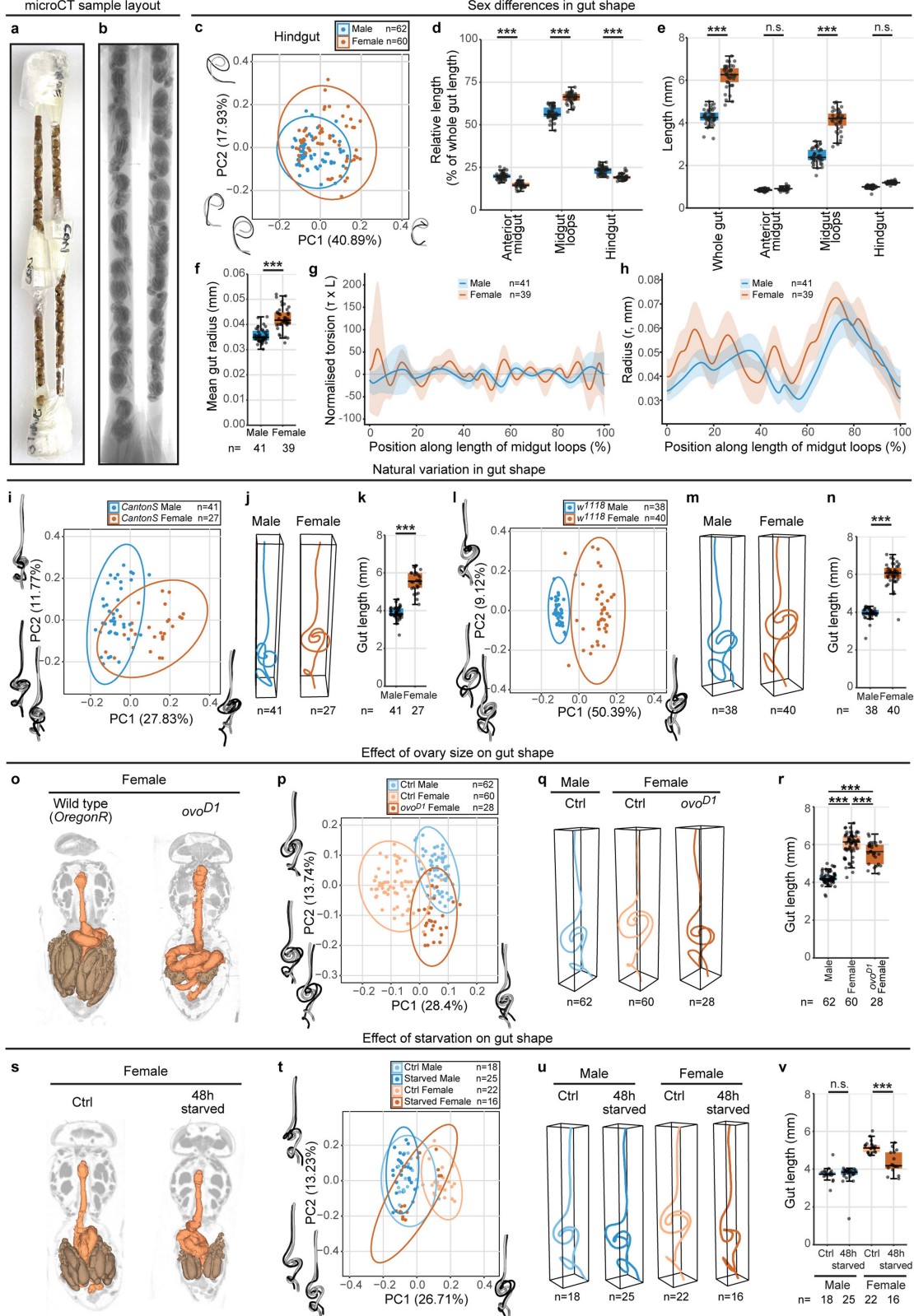

**Extended Data Fig. 1** | See next page for caption.

**Extended Data Fig. 1 | Sex differences, variation and plasticity of gut shape.**
**a**,**b**, Sample layout for microCT, showing parallel stacking of sample tubes (**a**) and a radiograph exemplifying a batch scan of multiple *Drosophila* specimens (**b**). **c**, PCA plot of shape variability in hindgut centrelines for wild-type *OregonR*. Male and female hindguts are significantly different in shape (Supplementary Table 5-6). **d**-**f**, Relative length (% of whole gut length) (**d**) and absolute length (**e**) of different gut regions and average gut radius (**f**) for *OregonR*. **g**-**h**, Gut torsion normalized by length (**g**) and radius (**h**) along midgut loop region for *OregonR*. **i**, PCA plot of wild-type *CantonS* whole gut shape variability showing male and female guts are significantly different in shape. **j**,**k**, Average *CantonS* gut centrelines (**j**) and gut length (**k**) showing differences between males and females. **l**, PCA plot of $w^{1118}$ gut shape variability showing male and female guts are significantly different in shape. **m**,**n**, Average $w^{1118}$ gut centrelines (**m**) and gut length (**n**) showing differences between males and females. **o**, Anteroposterior slices of female microCT scans overlaid with 3D reconstructions of the gut (orange) and ovaries (brown), showing differences in gut position between *ovoD1* mutant females and controls. **p**, PCA plot of *ovoD1* gut shape variability showing significant difference in shape to controls.

**q**,**r**, Average *ovoD1* mutant female gut centreline (**q**) and gut length (**r**) showing differences relative to controls. **s**, Anteroposterior slices of female microCT scans overlaid with 3D reconstructions of the gut (orange) and ovaries (brown), showing gut and ovary positions in flies starved for 48 h and controls. **t**, PCA plot of 48 hour-starved gut shape variability showing significant difference in shape to controls. **u**,**v**, Average gut centrelines (**u**) and gut length (**v**) of flies starved for 48 h, showing differences relative to controls. Line graphs: mean and standard deviation. Boxplots: line = median, box = first quartile and third quartile, whiskers = minimum and maximum. PCA plots: ellipses represent the 95% confidence space for each group. Diagrams represent the extremes of variation along each PC (see methods). n = number of biologically independent samples. Statistical significance was assessed using one-way ANOVA followed by Tukey post-hoc tests (**d**,**e**,**r**,**v**) or a two-sided two-sample t-test (**f**,**k**,**n**): non-significant (n.s.) = P > 0.05; ***= P < 0.001. See Supplementary Information for exact P-values, statistical tests and sample sizes. Males are in blue, females in orange and controls in lighter matching colours. Ctrl = control group (see genotypes in Supplementary Information).

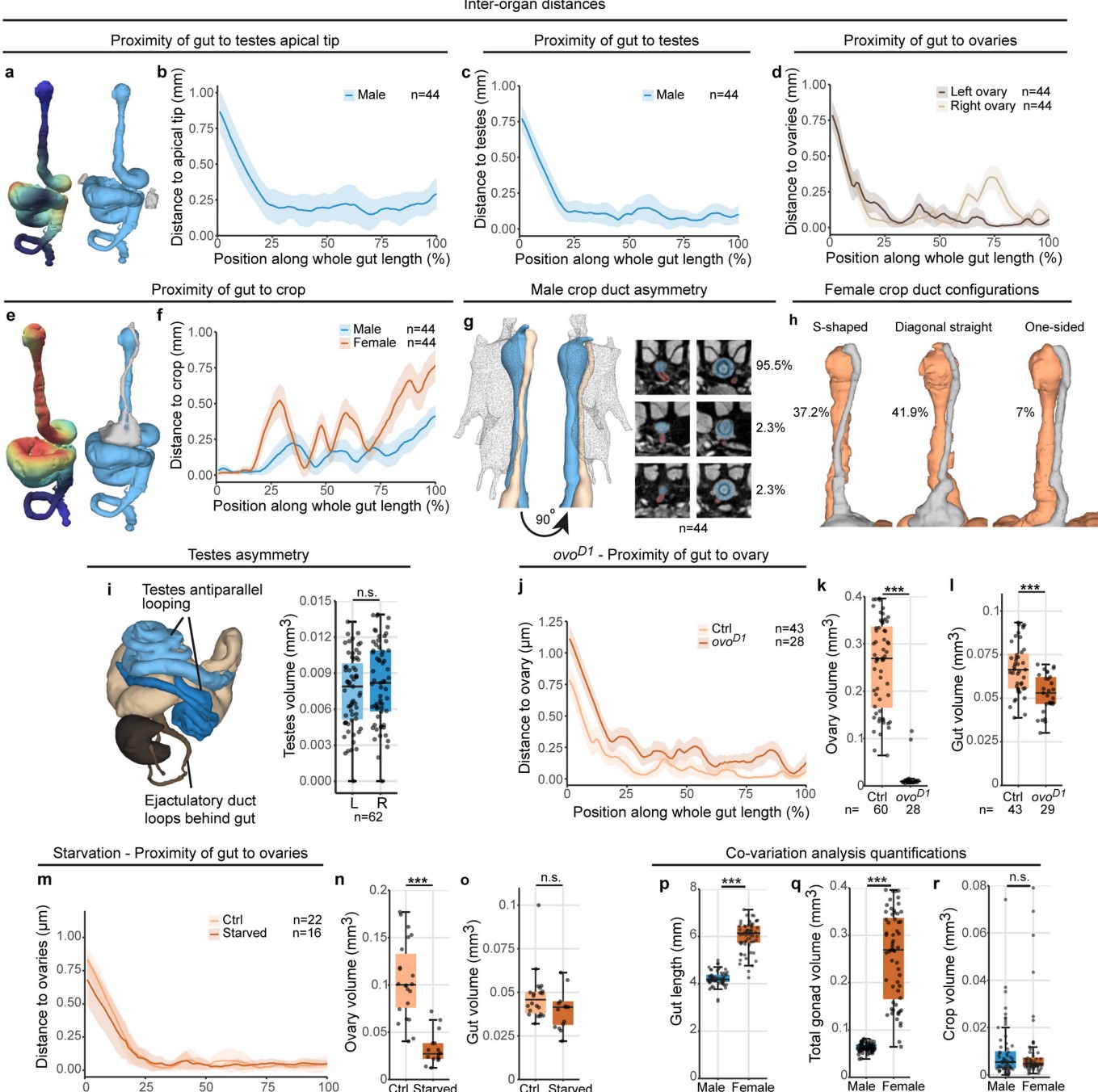

**Extended Data Fig. 2 | Multi-organ analyses. a-f**, Gut proximity to testes apical tip (**a-b**), testes (**c**), ovaries (**d**) and crop (**e-f**) in wild-type *OregonR* flies, shown by heatmap on representative 3D segmentation of the gut (**a**,**e**, Key in Fig. 1) and by plotted distance along the whole gut length (**b**,**c**,**d**,**f**). **g**, 3D volume segmentation showing crop duct proximity with gut and CNS in males (left). Dorsal-ventral cross-sectional views through upper thorax at top and centre of the proventriculus showing asymmetry of crop duct position (red) relative to gut (orange, right). **h**, 3D volume segmentation showing crop duct configurations and anterior midgut of *OregonR* females. **i**, 3D segmentation of male genital tract including testes (blue), accessory glands, ejaculatory duct and ejaculatory bulb (shades of brown). Volume of left and right testes is not significantly different. **j**, Gut proximity to ovaries in *ovoD1* mutant and control females, shown by plotted distances to positions along the whole gut length. **k-l**, Ovary (**k**) and gut (**l**) volumes in *ovoD1* mutant females and controls. **m**, Gut

proximity to ovaries in 48 hour-starved and control females, shown by plotted distances to positions along the whole gut length. **n-o**, Ovary (**n**) and gut (**o**) volumes in 48 hour-starved females and controls. **p-r**, Gut length (**p**), total gonad volume (**q**) and crop volume (**r**) for *OregonR* males and females. (These data were used in Procrustes ANOVA to assess constraint on gut shape, see Supplementary Tables 17–19). Line graphs: mean and standard deviation. Boxplots: line = median, box = first quartile and third quartile, whiskers = minimum and maximum. n = number of biologically independent samples. Statistical significance was assessed using two-sided paired t-test (**i**) or a two-sided two-sample t-test (**k**,**l**,**n-r**): non-significant (n.s.) = P > 0.05; *** = P < 0.001. See Supplementary Information for exact organ contact frequencies, P-values, statistical tests and sample sizes. Males are in blue, females in orange and controls in lighter matching colours. Ctrl = control group (see genotypes in Supplementary Information).

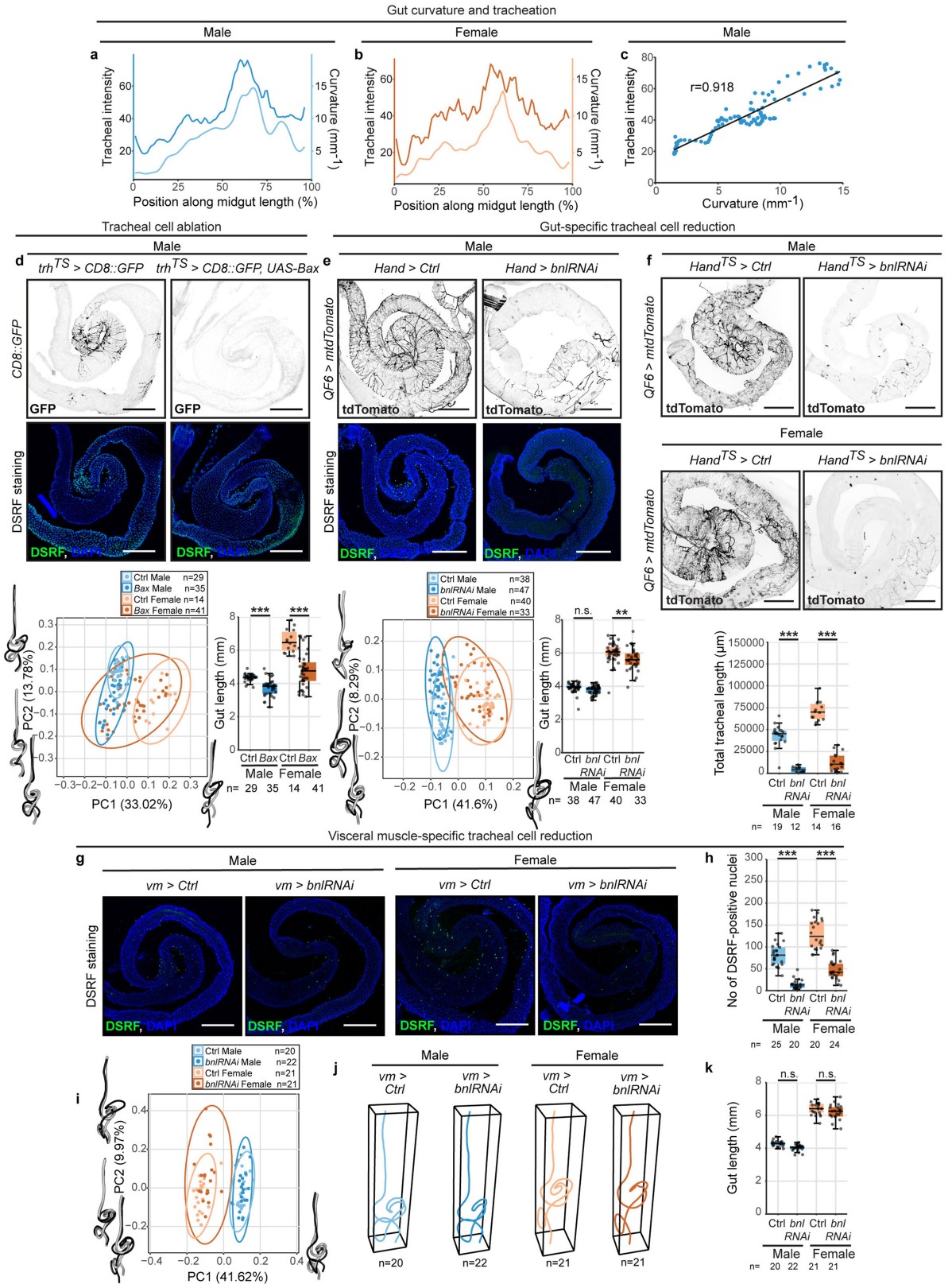

**Extended Data Fig. 3** | See next page for caption.

**Extended Data Fig. 3 | Genetic targeting of intestinal trachea. a-c**, Similar pattern of gut tracheation and gut curvature, shown by average intensity of tracheal signal from *btl>myr::GFP* and average gut curvature of *OregonR* (lighter matching colours) along the midgut for males (**a**) and females (**b**) and correlated at same relative midgut positions for males (**c**). **d**, *trh^{TS}>Bax* reduces tracheal terminal branches (top) and number of DSRF-positive nuclei (middle) in male midguts. PCA plot of *trh^{TS}>Bax* gut shape variability (bottom left). Groups are significantly different in shape (Supplementary Tables 20,21). *trh^{TS}>Bax* have differences in gut length relative to controls (bottom right). **e**, *Hand>bnlRNAi* reduces trachea in male midguts, shown by decrease in trachea branches (top), or in number of DSRF-positive nuclei (middle). PCA plot of *Hand>bnlRNAi* gut shape variability (bottom left). Groups are significantly different in shape (Supplementary Tables 24-25). *Hand>bnlRNAi* show marginal differences in female gut length relative to controls (bottom right). **f**, *Hand^{TS}>bnlRNAi* reduces trachea in the midgut, shown by decreased trachea branches (top) and quantified by the total tracheal branch length (bottom). **g-h**, *vm>bnlRNAi* decreases number of DSRF-positive nuclei (**g**, quantified in **h**). **i**, PCA plot of *vm>bnlRNAi* gut shape variability. Groups are significantly different in shape (Supplementary Tables 26-27). **j,k** Average *vm>bnlRNAi* gut centrelines (**j**) and gut length (**k**) showing altered shape but no differences in gut length relative to controls. Line graphs show mean. Boxplots: line = median, box = first quartile and third quartile, whiskers = minimum and maximum. PCA plots: ellipses represent the 95% confidence space for each group. Diagrams represent the extremes of variation along each PC (see methods). n = number of biologically independent samples. Statistical significance was assessed using one-way ANOVA followed by Tukey post-hoc tests (**d-f,h,k**): non-significant (n.s.) = $P > 0.05$; ** = $0.01 > P > 0.001$; *** = $P < 0.001$. See Supplementary Information for exact P-values, statistical tests and sample sizes. Males are in blue, females in orange and controls in lighter matching colours. Ctrl = control group (see genotypes in Supplementary Information). Scale bars: 200 μm.

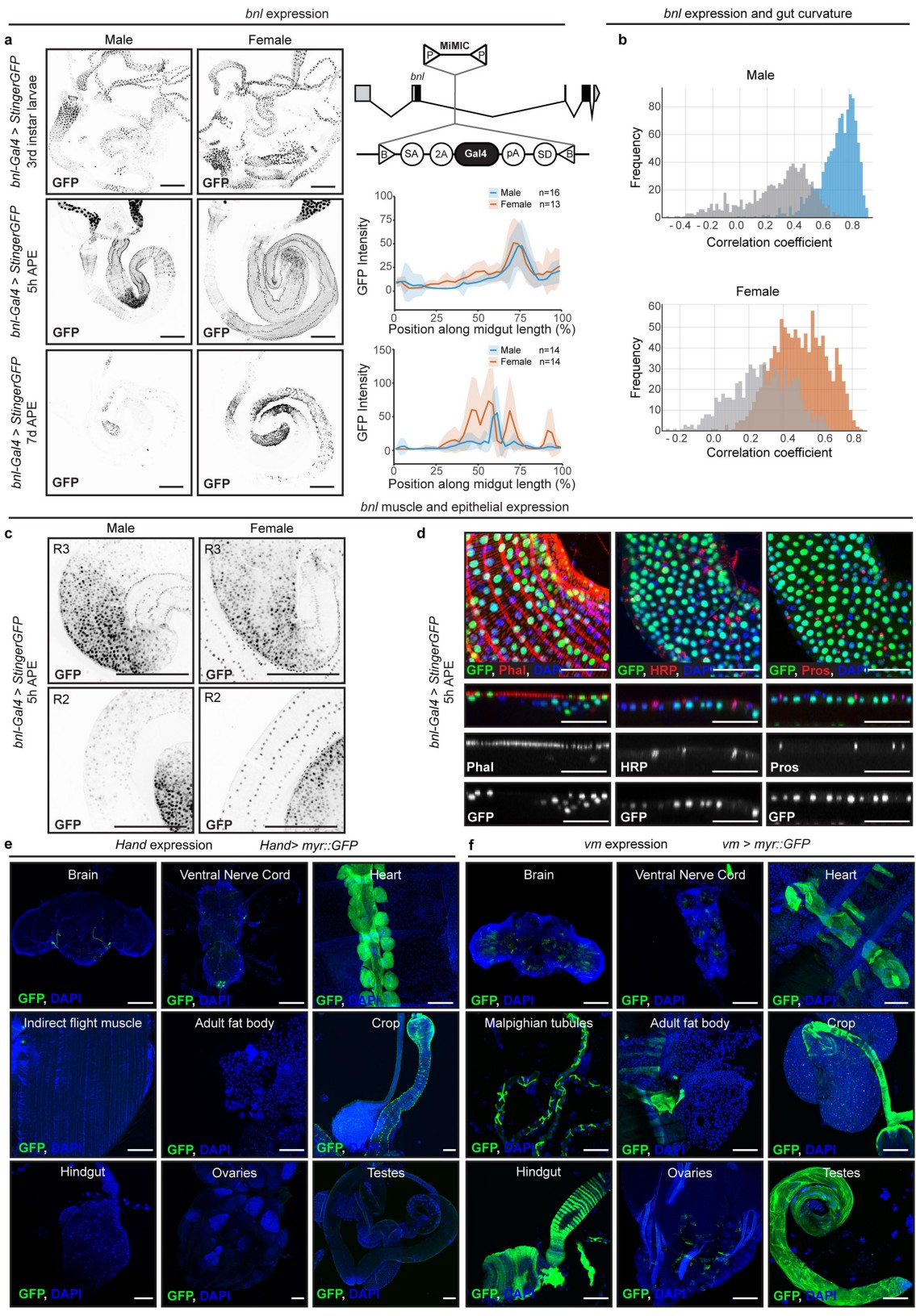

**Extended Data Fig. 4** | See next page for caption.

**Extended Data Fig. 4 | Expression of *bnl* and gut muscle drivers. a**, *bnl* expression (*bnl-Gal4>StingerGFP*) in the gut epithelium and muscles at different stages of development: in larval enterocytes (top left), in adult gut muscle at 5 h after pupal eclosion (APE) (middle left) and in adult enterocytes at 7 days APE (bottom left). Schematic shows *bnl* locus and the position of *T2A-Gal4* insertion (top right). GFP intensity quantified at 5 h APE (middle right) and 7 days APE (bottom right). **b**, Histogram of Pearson correlation coefficients between gut curvature and *bnl* intensity along midgut. Grey bars represent correlations with a "control" *bnl* intensity generated by fitting an autoregressive process to the true data (see Methods). **c**, *bnl* expression (*bnl-Gal4>StingerGFP*) at 5 h APE in gut R3 region showing enterocyte expression (top) and R2 region showing muscle expression (bottom). **d**, *bnl* expression (*bnl-Gal4 > StingerGFP*) overlaps F-actin-rich muscle layer, shown by Phalloidin staining (Phal), but not intestinal stem cells or enteroblasts, shown by horseradish peroxidase staining (HRP), nor enteroendocrine cells, shown by Prospero staining (Pros). **e**, *Hand* expression (*Hand-Gal4>myr::GFP*) in: adult brain, ventral nerve cord, heart, indirect flight muscle, fat body, crop, hindgut, ovaries and testes. **f**, *vm* expression (*vm-Gal4>myr::GFP*) in: adult brain, ventral nerve cord, heart, Malpighian tubules, fat body, crop, hindgut, ovaries and testes. Line graphs: mean and standard deviation. n = number of biologically independent samples. See Supplementary Information for exact P-values, statistical tests and sample sizes. Males are in blue and females in orange. Scale bars: (**a**,**c**) 200 μm, (**d**) 50 μm, (**e**,**f**) 100 μm.

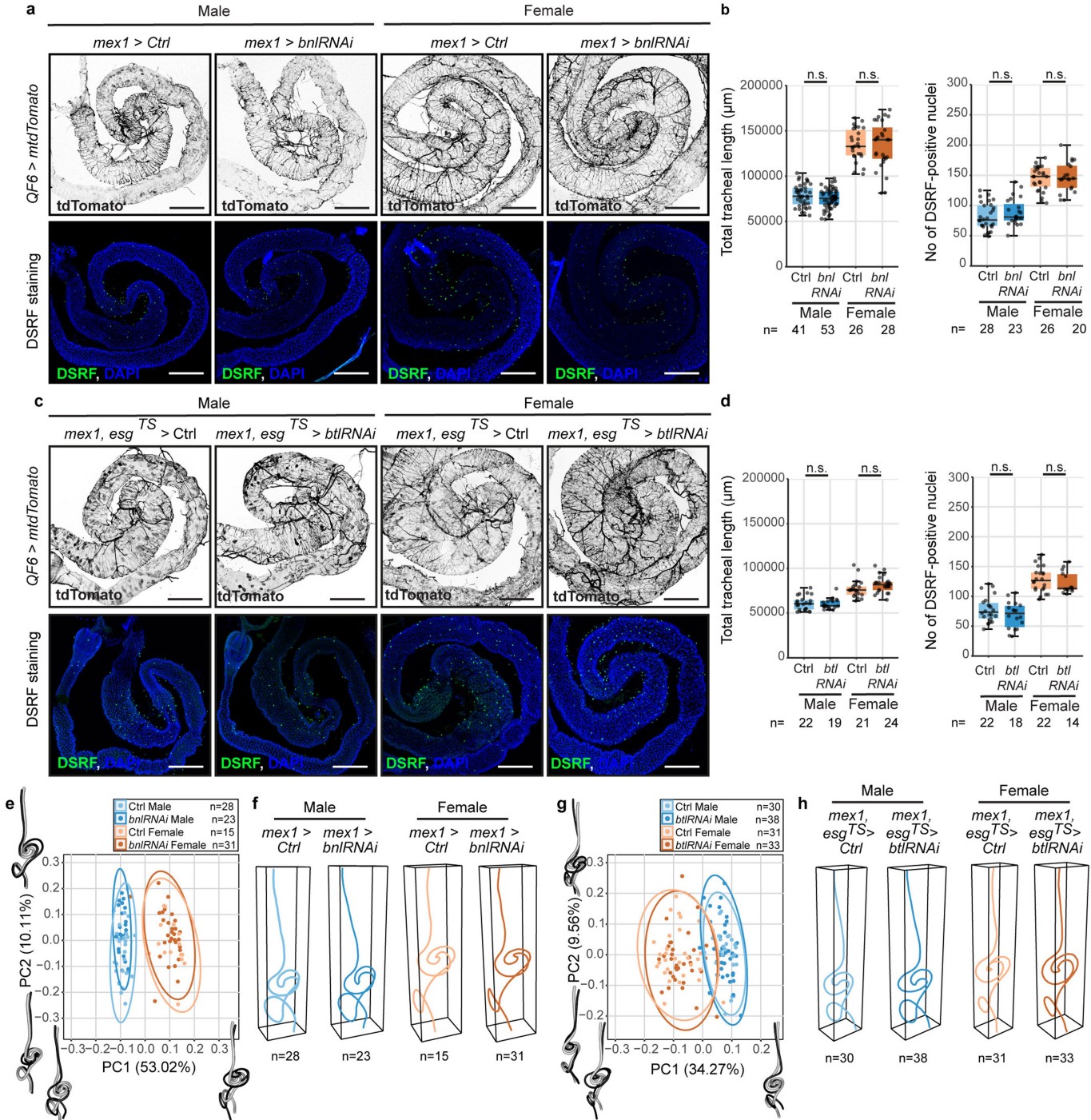

**Extended Data Fig. 5 | Lack of epithelial contribution to gut shape phenotypes. a,b**, *bnl* downregulation from gut enterocytes (*mex1>bnlRNAi*) does not reduce tracheal branches (*QF6>mtdTomato*) (top (**a**), quantified in left (**b**)), or number of DSRF-positive nuclei (bottom (**a**), quantified in right (**b**)). **c-d**, *btl* downregulation from gut epithelium for 3 days prior to pupal eclosion (*mex1,esg^TS^>btlRNAi*) does not reduce tracheal branches (*QF6>mtdTomato*) (top (**c**), quantified in left (**d**)), or number of DSRF-positive nuclei (bottom (**c**), quantified in right (**d**)). **e**, PCA plot of *mex1>bnlRNAi* gut shape variability showing no significant difference in shape to controls. **f**, Average *mex1>bnlRNAi* gut centrelines show no differences in shape relative to controls. **g**, PCA plot of *mex1,esg^TS^>btlRNAi* gut shape variability showing no significant difference in

shape to controls. **h**, Average *mex1,esg^TS^>btlRNAi* gut centrelines show no significant differences in shape relative to controls. Boxplots: line = median, box = first quartile and third quartile, whiskers = minimum and maximum. PCA plots: ellipses represent 95% confidence space for each group. Diagrams represent the extremes of variation along each PC (see methods). n = number of biologically independent samples. Statistical significance was assessed using one-way ANOVA followed by Tukey post-hoc tests (**b**,**d**): non-significant (n.s.) = P > 0.05. See Supplementary Information for exact P-values, statistical tests and sample sizes. Males are in blue, females in orange and controls in lighter matching colours. Ctrl = control group (see genotypes in Supplementary Information). Scale bars: 200 μm.

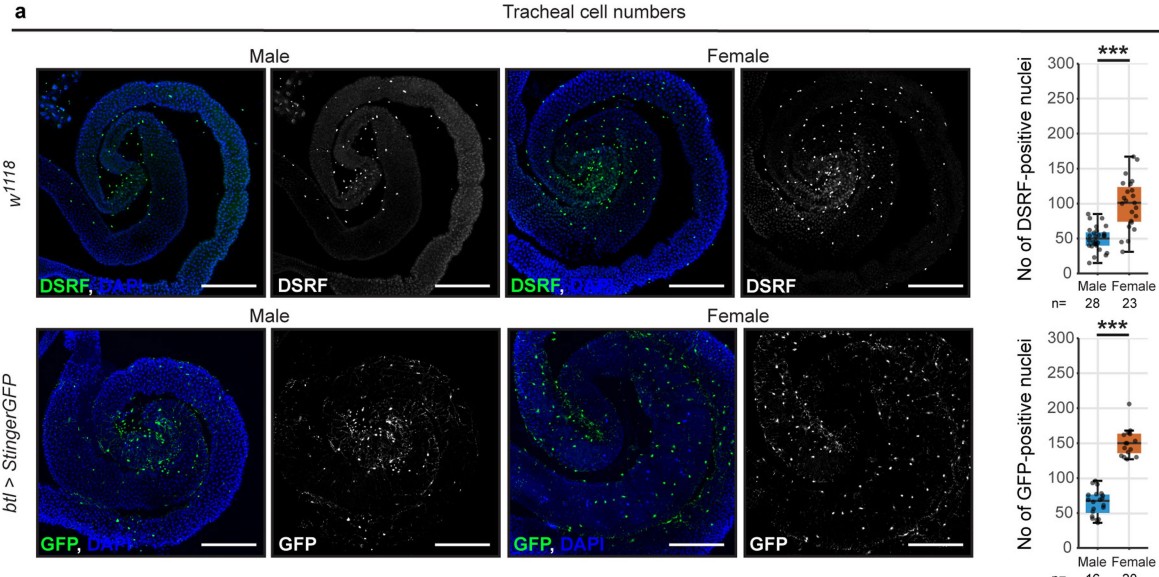

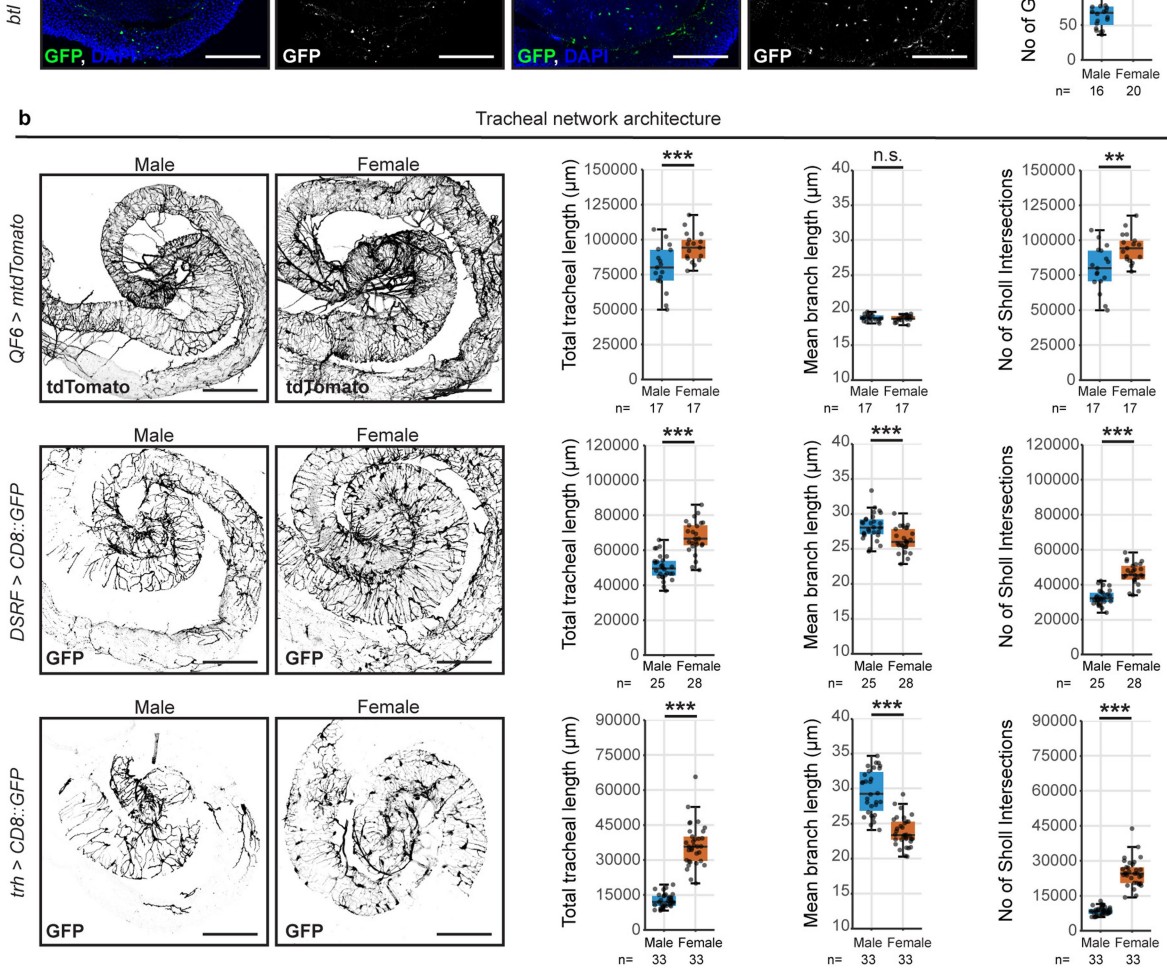

**Extended Data Fig. 6 | Sex differences in the intestinal tracheal network.**
**a**, Trachea labelled by DSRF staining in *w^1118^* (top) and *btl>StingerGFP* (bottom)
show difference in tracheal terminal cell number between male and female
guts (quantifications in right). **b**, Trachea labelled by *QF6>mtdTomato* (top),
*DSRF > CD8::GFP* (middle) or *trh > CD8::GFP* (bottom), showing higher total
tracheal length and tracheal branching (no. sholl intersections) in females
compared to males and shorter mean tracheal branch lengths in females

compared to males (except in *QF6>mtdTomato*) (quantifications shown in
right). Boxplots: line = median, box = first quartile and third quartile,
whiskers = minimum and maximum. n = number of biologically independent
samples. Statistical significance was assessed using two-sided two-sample
t-tests (**a**,**b**): non-significant (n.s.) = p > 0.05; ** = 0.01>p > 0.001; *** = p < 0.001.
See Supplementary Information for exact p-values, statistical tests and sample
sizes. Males are in blue, females in orange. Scale bars: 200 μm.

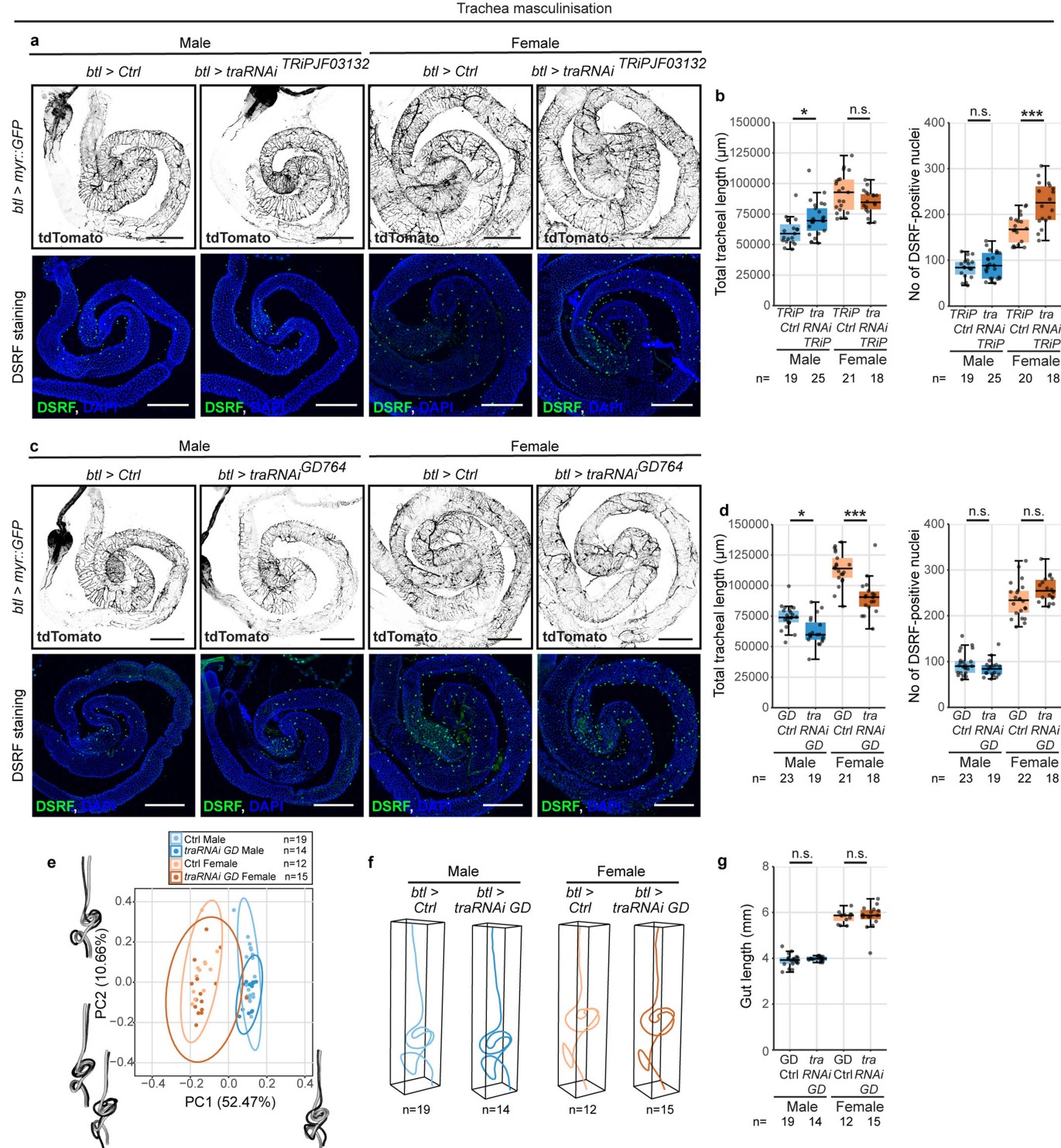

Trachea masculinisation

**Extended Data Fig. 7 | *tra*-mediated tracheal masculinizations.**
a–d, Expression of two *traRNAi* lines in tracheal cells, by means of *btl > TRiPJF03132*
(**a**,**b**) and *btl > GD764* (**c**,**d**), does not effectively masculinize midgut trachea,
as seen by trachea labelled by *btl>myr::GFP* (top (**a**,**c**), quantified in left (**b**,**d**)),
or by DSRF-positive tracheal terminal cell nuclei (bottom (**a**,**c**), quantified in
right (**b**,**d**)). **e**, PCA plot of *btl>traRNAi.GD764* gut shape variability showing no
significant difference in shape to controls. **f**,**g**, Average *btl>traRNAi.GD764* gut
centrelines (**f**) and gut length (**g**) show no differences relative to controls. We
note that we did not use a *UAS-Dcr-2* transgene to enhance *tra* downregulation
as its expression in trachea results in deleterious phenotypes in the absence of
other transgenes. Boxplots: line = median, box = first quartile and third
quartile, whiskers = minimum and maximum. PCA plots: ellipses represent
the 95% confidence space for each group. Diagrams represent the extremes
of variation along each PC (see methods). n = number of biologically
independent samples. Statistical significance was assessed using one-way
ANOVA followed by Tukey post-hoc tests (**b**,**d**,**g**): non-significant (n.s.) =
p > 0.05; * = 0.05>p > 0.01; *** = p < 0.001. See Supplementary Information for
exact p-values, statistical tests and sample sizes. Males are in blue, females
in orange and controls in lighter matching colours. Ctrl = control group
(see genotypes in Supplementary Information). Scale bars: 200 μm.

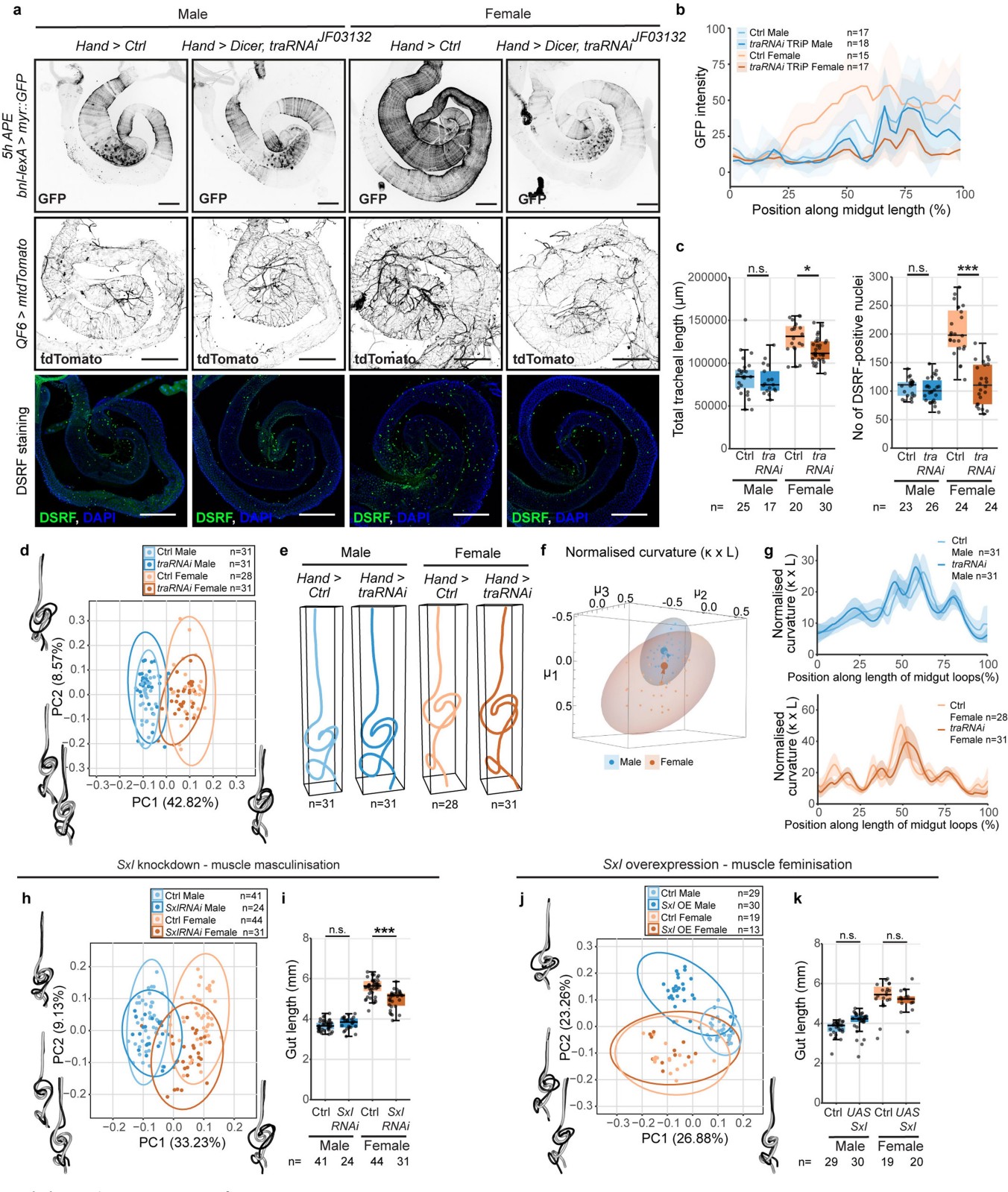

**Extended Data Fig. 8** | See next page for caption.

**Extended Data Fig. 8 | Genetic sex reversals of intestinal muscles.**
**a-c**, *Hand>traRNAi.JF03123* masculinizes *bnl* expression at 5 h after pupal
eclosion (APE), seen by *bnl-lexA>myr::GFP* (top (**a**), quantified in (**b**)) and
reduces tracheal branches (*QF6>mtdTomato*) (middle (**a**), quantified in left (**c**))
and the number of DSRF-positive nuclei (bottom (**a**), quantified in right (**c**)).
**d**, PCA plot of *Hand>traRNAi* gut shape variability. Females are significantly
different in shape to controls (Supplementary Tables 32-33). **e**, Average
*Hand>traRNAi* gut centrelines show altered shape in the central midgut region
in females relative to controls. **f**, Multidimensional scaling plot representing
similarity in gut curvature normalized by gut length between *Hand>traRNAi*
and controls. Arrows represent shift in the mean from control to *Hand>traRNAi*.
**g**, *Hand>traRNAi* females have reduced average gut curvature normalized
by gut length along midgut loop region relative to controls. **h**, PCA plot of
*Hand>SxlRNAi* gut shape variability. Groups are significantly different in shape

(Supplementary Tables 34-35). **i**, *Hand>SxlRNAi* guts show differences in length
in females relative to controls. **j**, PCA plot of *Hand>Sxl* gut shape variability.
Males are significantly different in shape to controls (Supplementary Tables
36-37). **k**, *Hand>Sxl* guts show no differences in length relative to controls. Line
graphs: mean and standard deviation. Boxplots: line = median, box = first
quartile and third quartile, whiskers = minimum and maximum. **d,f,h,j**, Ellipses
represent 95% confidence space for each group. Diagrams represent the
extremes of variation along each PC (see methods). n = number of biologically
independent samples. Statistical significance was assessed using one-way
ANOVA followed by Tukey post-hoc tests (**c,i,k**): non-significant (n.s.) = p > 0.05;
* = 0.05>p > 0.01; *** = p < 0.001. See Supplementary Information for exact
p-values, statistical tests and sample sizes. Males are in blue, females in orange
and controls in lighter matching colours. Ctrl = control group (see genotypes
in Supplementary Information). Scale bars: 200 μm.

**Extended Data Fig. 9 | Hypoxia reporters in the presence and absence of trachea. a**, Expression of *SimaODD::GFP* across several tissues (brain, midgut and gonads), showing higher GFP levels in the adult midgut of both males and females. **b**, Expression of *bnlRNAi* from gut muscle (*Hand>bnlRNAi*) in males does not change levels of SimaODD::GFP (top) nor Ldh::GFP (bottom) compared to controls. Ctrl = control group (see genotypes in Supplementary Information). Scale bars: 200 µm.

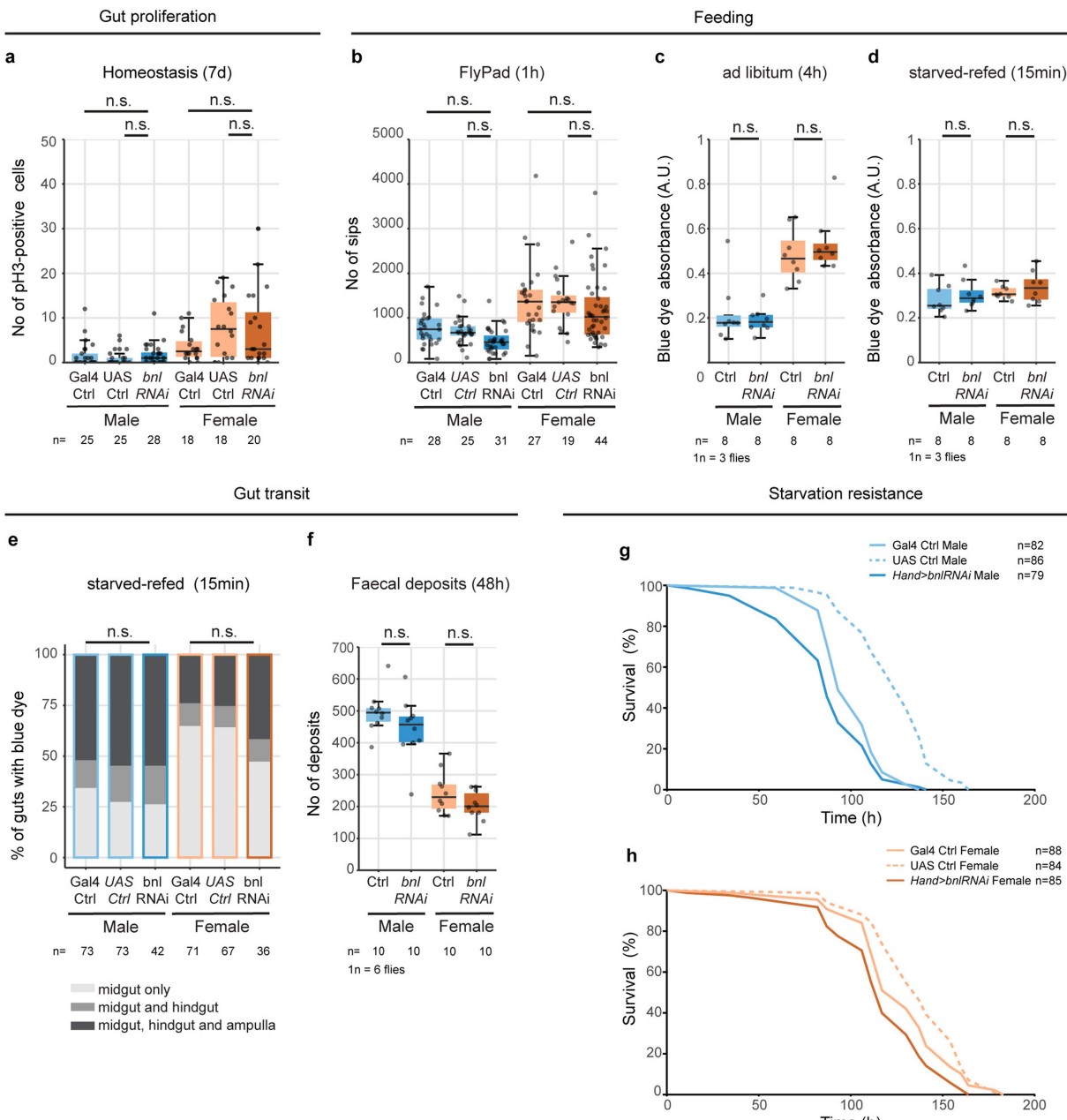

**Extended Data Fig. 10 | Functional interrogation of flies with reduced gut tracheation. a**, Expression of *bnlRNAi* from gut muscles (*Hand>bnlRNAi*) does not change mitotic indices in midgut of 7-day-old flies seen by number of pH3-positive cells. **b**, *Hand>bnlRNAi* expression marginally reduces total number of sips during 1 h of feeding, as recorded in a FlyPad device (see Methods). **c-d**, *Hand>bnlRNAi* expression does not alter FCF blue-dyed food contents in gut of adult flies fed *ad libitum* (**c**) or starved for 16 h and re-fed for 15 min (**d**), as measured by blue dye absorbance at 634 nm from cleared homogenates of groups of 3 flies. **e**, *Hand>bnlRNAi* expression does not alter food transit time in gut, measured by percentage of flies with dyed food present in portions of the gut (midgut, hindgut and ampulla). Only size-matched guts are shown to rule out effects of gut size (see Methods). **f**, *Hand>bnlRNAi* expression does not alter the total number of faecal deposits, quantified by

bromophenol blue-dyed deposits after 60 h of feeding (see Methods). **g-h**, *Hand>bnlRNAi* expression reduces survival in male (**g**) and female (**h**) flies starved in 1% agar relative to respective controls, shown by Kaplan-Meier survival curves. Boxplots: line = median, box = first quartile and third quartile, whiskers = minimum and maximum. n = number of biologically independent samples. Statistical significance was assessed using one-way ANOVA followed by Tukey post-hoc tests (a–d,**f**), logistic regression with Chi2 post-hoc tests (**e**) and log-rank test (**g**,**h**): non-significant (n.s.) = p > 0.05; * = 0.05>p > 0.01; ** = 0.01>p > 0.001; *** = p < 0.001. See Supplementary Information for exact p-values, statistical tests and sample sizes. Males are in blue, females in orange and controls in lighter matching colours. Ctrl = control group (see genotypes in Supplementary Information).

# Reporting Summary

## Statistics

For all statistical analyses, confirm that the following items are present in the figure legend, table legend, main text, or Methods section.

| n/a | Confirmed | |
|---|---|---|
| ☐ | ☒ | The exact sample size (*n*) for each experimental group/condition, given as a discrete number and unit of measurement |
| ☐ | ☒ | A statement on whether measurements were taken from distinct samples or whether the same sample was measured repeatedly |
| ☐ | ☒ | The statistical test(s) used AND whether they are one- or two-sided<br>*Only common tests should be described solely by name; describe more complex techniques in the Methods section.* |
| ☐ | ☒ | A description of all covariates tested |
| ☐ | ☒ | A description of any assumptions or corrections, such as tests of normality and adjustment for multiple comparisons |
| ☐ | ☒ | A full description of the statistical parameters including central tendency (e.g. means) or other basic estimates (e.g. regression coefficient) AND variation (e.g. standard deviation) or associated estimates of uncertainty (e.g. confidence intervals) |
| ☐ | ☒ | For null hypothesis testing, the test statistic (e.g. *F*, *t*, *r*) with confidence intervals, effect sizes, degrees of freedom and *P* value noted<br>*Give P values as exact values whenever suitable.* |
| ☒ | ☐ | For Bayesian analysis, information on the choice of priors and Markov chain Monte Carlo settings |
| ☒ | ☐ | For hierarchical and complex designs, identification of the appropriate level for tests and full reporting of outcomes |
| ☐ | ☒ | Estimates of effect sizes (e.g. Cohen's *d*, Pearson's *r*), indicating how they were calculated |

*Our web collection on statistics for biologists contains articles on many of the points above.*

## Software and code

Policy information about availability of computer code

| Data collection | Zeiss Xradia Versa 510, Bruker Skyscan 1272 and Bruker Skyscan 1172 for microCT image acquistion; Zeiss Reconstructor v11 and Bruker Nrecon v2 for microCT image reconstruction; Leica SP5 and Leica TCS SP8 DLS for confocal image acquisition; Nikon CSU-W1 SoRa spinning disk microscope with NIS-Elements software for laser ablation experiments; FlyPAD device with Bonsai software for feeding assays. |
|---|---|
| Data analysis | Imaris x64 v9.9.0 with 'Filament tracer' and 'Batch' packages for confocal image analysis; FIJI v2.0.0-rc-69/1.52p for confocal image and microCT scan analysis with Simple neurite tracer plugin v3.1.6 for centreline tracing; R v3.6.0 with packages 'geomorph' v3.2.1, 'nat' v1.8.18, 'RRPP' v0.5.2 for geometric morphometric analysis of gut shape, and 'RANN' v1.8.18 for proximity analysis; R v4.2.1 with packages 'ggplot2' v3.4.0, R 'dplyr' v1.0.10 for data visualisation and statistical analysis and package 'VGAM' v1.1 for intestinal transit; Adobe Photoshop v25.3.1 for tracheal coverage; ITK-snap v3.8.0 for microCT segmentation; Meshlab v2020.07, Paraview 5.10.0 for 3D mesh analysis; Mathematica v13.1, Python v3.10.0 with packages 'NUMPY' and 'scikit-fda' for gut shape analysis; MATLAB R2023b for analysis of FlyPAD recordings; T.U.R.D software for intestinal excretion; GraphPad Prism v9.4.1 for starvation resistance. Custom code is available on GitHub via Zenodo: http://doi.org/10.5281/zenodo.10905446 |

For manuscripts utilizing custom algorithms or software that are central to the research but not yet described in published literature, software must be made available to editors and reviewers. We strongly encourage code deposition in a community repository (e.g. GitHub). See the Nature Portfolio guidelines for submitting code & software for further information.

## Data

Policy information about availability of data

All manuscripts must include a data availability statement. This statement should provide the following information, where applicable:
- Accession codes, unique identifiers, or web links for publicly available datasets
- A description of any restrictions on data availability
- For clinical datasets or third party data, please ensure that the statement adheres to our policy

All reconstructed microCT scans, gut centreline files and organ segmentation files are available on Figshare with the identifier: https://doi.org/10.25418/crick.25598859. All remaining data generated or analysed during this study are included in this published article (and its Extended Data/Supplementary Information files) and accompanying source data files. Further information can be requested from the corresponding author.

## Research involving human participants, their data, or biological material

Policy information about studies with human participants or human data. See also policy information about sex, gender (identity/presentation), and sexual orientation and race, ethnicity and racism.

| | |
|---|---|
| Reporting on sex and gender | N/A |
| Reporting on race, ethnicity, or other socially relevant groupings | N/A |
| Population characteristics | N/A |
| Recruitment | N/A |
| Ethics oversight | N/A |

Note that full information on the approval of the study protocol must also be provided in the manuscript.

# Field-specific reporting

Please select the one below that is the best fit for your research. If you are not sure, read the appropriate sections before making your selection.

☒ Life sciences ☐ Behavioural & social sciences ☐ Ecological, evolutionary & environmental sciences

For a reference copy of the document with all sections, see nature.com/documents/nr-reporting-summary-flat.pdf

# Life sciences study design

All studies must disclose on these points even when the disclosure is negative.

| | |
|---|---|
| Sample size | Sample sizes are provided for each experiment in Supplementary Information. Fly numbers are not limiting so no power calculations were used to pre-determine sample size. Oversampling was mitigated by choosing sample sizes based on previous knowledge of phenotypic variability in controls and other mutants. In particular, we used similar sample sizes used previously to determine variability in Drosophila gut length, as reported in Hudry et al 2019 (https://doi.org/10.1038/nature16953), White et al. 2021 (https://doi.org/10.1073/pnas.2018112118) and Bonfini et al. 2021 (https://doi.org/10.7554/eLife.64125). Similar sample sizes for different animal groups (e.g.mutants vs controls) were tested in the same experimental design. |
| Data exclusions | All measured datapoints are displayed in figures and outliers were not excluded from data analysis |
| Replication | Experiments were typically replicated 3 times and only those experiments for which repeats gave comparable outcomes are included in the manuscript. |
| Randomization | Experimental and control flies were bred in identical conditions, and were randomised whenever possible (for example, with regard to housing, position in tray). Control and experimental samples were dissected and processed at the same time and on the same slides or tips for confocal imaging and microCT. Experiments were controlled for sex, mating status, genotype and age. Detailed information is provided in the main text and, more systematically for each figure panel and in the Supplementary Information. |
| Blinding | Blinding was performed for a subset of experiments. Quantification of DSRF stainings, filament tracing of QF6>tomato-labelled trachea and quantifications of bnl expression along gut length was done on data blinded for genotype. Blinding for sex was not possible as this is visually obvious by differences in the length and diameter of the Drosophila gut. Similarly blinding for sex was not possible for microCT scans as ovaries and testes were visible in the images. |

# Reporting for specific materials, systems and methods

We require information from authors about some types of materials, experimental systems and methods used in many studies. Here, indicate whether each material, system or method listed is relevant to your study. If you are not sure if a list item applies to your research, read the appropriate section before selecting a response.

## Materials & experimental systems

| n/a | Involved in the study |
|-----|----------------------|
| ☐ | ☒ Antibodies |
| ☒ | ☐ Eukaryotic cell lines |
| ☒ | ☐ Palaeontology and archaeology |
| ☐ | ☒ Animals and other organisms |
| ☒ | ☐ Clinical data |
| ☒ | ☐ Dual use research of concern |
| ☒ | ☐ Plants |

## Methods

| n/a | Involved in the study |
|-----|----------------------|
| ☒ | ☐ ChIP-seq |
| ☒ | ☐ Flow cytometry |
| ☒ | ☐ MRI-based neuroimaging |

## Antibodies

**Antibodies used**

Primary antibodies: Mouse anti-DSRF (Active Motif, 39093), goat anti-GFP (Abcam, ab5450), rabbit anti-mCherry (Abcam, ab167453), mouse anti-prospero (DSHB, MR1A) and anti-horseradish peroxidase (HRP) Rhodamine (TRITC)-conjugated (Jackson ImmunoResearch, 123-025-021). Secondary antibodies: anti-rabbit FITC-conjugated (Jackson ImmunoResearch, 711-97-003), anti-mouse Cy3-conjugated (Jackson ImmunoResearch715-166-150), anti-mouse Cy5-conjugated (Jackson ImmunoResearch, 715-175-151) and anti-goat FITC-conjugated (Jackson ImmunoResearch, 112-095-044).

**Validation**

All antibodies used were previously published and tested in Drosophila melanogaster tissues. Working concentrations were based on previously published assessments: Mouse anti-DSRF (Active Motif, 39093) was tested and reported to label terminal tracheal cells in the Drosophila gut in Linneweber et al. 2014 (https://doi.org/10.1016/j.cell.2013.12.008); Mouse anti-prospero (DSHB, MR1A) was shown to label Drosophila gut enteroendocrine cells in Michelli and Perrimon 2006 (https://doi.org/10.1038/nature04371) and in Ohlstein and Spradling 2006 (https://doi.org/10.1038/nature04333). Anti-horseradish peroxidase (HRP) Rhodamine (TRITC)-conjugated (Jackson ImmunoResearch, 123-025-021) was show to label Drosophila gut stem cells and enteroblasts in O'Brien et al. 2011 (https://doi.org/10.1016/j.cell.2011.08.048). Goat anti-GFP (Abcam, ab5450) and rabbit anti-mCherry (Abcam, ab167453) were used as tested by the manufacturer.

## Animals and other research organisms

Policy information about studies involving animals; ARRIVE guidelines recommended for reporting animal research, and Sex and Gender in Research

**Laboratory animals**

Adult Drosophila melanogaster, males and females, upto 2-5 weeks old. Strains used: Hand-Gal4[MI04106-TG4.0] (BDSC: 66795), mex1-Gal4, esg-Gal4 (NP7397), btl-Gal4 (DGGR: 109128), trh-Gal4 (GMR14D03, BDSC: 47463), vm-Gal4 (GMR13B09, BDSC: 48547), DSRF-Gal4 (BDSC: 25753), bnl-Gal4[MI00874-TG4.1] (this study), bnl[lexA] (a gift from Sougata Roy), QF6 (a gift from Julia Cordero), UAS-traRNAi.TRiPJF03132 (BDSC: 28512), UAS-traRNAi.GD764 (VDRC: 2560), UAS-SxlRNAi.TRiPGL00634 (BDSC: 38195), UAS-bnlRNAi.GD3070 (VDRC: 5730), UAS-btlRNAi.KK100331 (VDRC: 110277), UAS-Bax (a gift from Julia Cordero), UAS-myr(src)::GFP M7E (BDSC: 5432), UAS-StingerGFP (BDSC:84278), UAS-Flybow.1.1B (used as 10xUAS-CD8::GFP, BDSC: 56803), QUAS-mtdTomato-3xHA (BDSC: 30005), 13xlexAop2-IVS-myr::GFP (BDSC: 32209), OregonR, w1118 (GD control, VDRC: 60000), UAS-mCherryRNAi.Valium10 (TRiP control, BDSC: 35787), ovoD1 (BDSC: 1309), UAS-Dcr-2 (BDSC: 24646, 24650), UAS-Gal80TS (ref. 73, BDSC: 7108), UASp-Sxl.alt5-C8 (used as UAS-Sxl, BDSC: 58484), Ubi-EGFP.ODD, Ubi-mRFP.nls (BDSC:86536), Ldh::GFPYD0852 (a gift from U. Banerjee).

**Wild animals**

This study did not involve wild animals.

**Reporting on sex**

Drosophila virgin male and virgin female samples were used and sex is clearly stated in main text and figures. All data display and analysis is shown seperately for each sex

**Field-collected samples**

The study did not involve samples collected in the field.

**Ethics oversight**

No ethical approval was required. The use of Drosophila melanogaster does not require ethical approval or guidance.

Note that full information on the approval of the study protocol must also be provided in the manuscript.

