## [Peer Review File · Nature]

Manuscript Title: The sex of organ geometry

Reviewer Comments & Author Rebuttals

Reviewer Reports on the Initial Version:

Referees' comments:

Referee #1 (Remarks to the Author):

Reviewer Report: Nature MS "The Sex of Organ Geometry" By Blackie et al.

In this manuscript, the authors present an outstanding piece of research that sheds light on the often-overlooked relationship between organ spatial arrangements and their functionality and communication. This work constitutes a masterful contribution of the highest quality, employing innovative techniques to quantify 3D features of organ shape, position, and inter-individual variability in *Drosophila melanogaster* flies. The hypothesis that there exists a logic to the shape and proximity of organs is intriguing, and the use of high-content volumetric scans to analyze large numbers of *Drosophila* flies offers an impressive and comprehensive dataset to address this question.

One of the most striking findings of this study is the revelation of stereotypical yet sexually dimorphic shapes of organs and their relative arrangements. This groundbreaking discovery challenges our current understanding of organ development and raises important questions about the underlying mechanisms responsible for these differences. The authors demonstrate an exemplary focus on the intestine, a highly relevant and complex organ whose sexually dimorphic shape is actively maintained by the tracheal system. This finding represents a significant advancement in our knowledge of the factors influencing organ morphology and how they may contribute to signaling paradoxes and sex differences in organ function. Moreover, the identification of a previously unrecognized bidirectional mechanism of sex differentiation in the tracheal system, rendering it sexually dimorphic, adds a new layer of biological complexity that has not been previously recognized. This discovery opens up exciting avenues of research, both in understanding sex-specific differences in organ geometry and in unraveling the broader implications of the tracheal system's role in shaping organs.

The conclusions drawn from this work are undoubtedly novel and original, significantly expanding our understanding of the relationship between organ geometry and sexual dimorphism. It has the potential to prompt a paradigm shift in how we perceive and study organ development and function. The outlook of this work is highly impressive as it holds great promise in advancing research in health and disease. The newfound knowledge of how organs' spatial arrangements impact communication and functionality is invaluable and will likely have far-reaching implications in medicine and biology.

In summary, the study "The Sex of Organ Geometry" is a remarkable achievement of the highest quality. Its innovative methods, insightful findings, and groundbreaking conclusions make it a milestone in the scientific community. The implications of this research are profound, and it will undoubtedly inspire and fuel further investigations in this exciting area of study. I recommend its publication without hesitation.

I feel the story would be even more complete and impactful if the authors discussed or even better included additional experimental work addressing the following points:

- Can an intestinal functional output (Transit time? Fecal deposits appearance?) be assayed in this work to probe the extent to which sex and the genetic manipulation impacting gut geometry used in the work are influencing intestinal biology?
- In the final paragraph of the manuscript's page 5, the authors focus on *bnl* expression in gut muscles. By downregulating *tra* expression in female gut muscles, they manage to antagonize *bnl* expression to "masculinize" it (i.e., reduce levels vs. female levels). While I agree that there is a clear effect on *bnl* expression, I believe such masculinization by manipulating *tra* levels will be pleiotropic; hence, I feel it would be good to provide a rescue experiment by selectively restoring *bnl* expression in a masculinized female gut and check the extent of gut geometry change (and physiological outputs if possible, see the point above).
- In general in the work, the dynamics of ISCs proliferation at steady state between males and females and/or during the genetic manipulation impacting gut geometry are not discussed nor investigated. This parameter is likely to impact gut shape, especially if this is region-restricted/specific, right? I feel it is probably important to integrate this parameter in the discussion/investigation?

Referee #2 (Remarks to the Author):

The manuscript by Blackie et al reports on organ shape sexual dimorphism. The researchers use micro-CT of adult *Drosophila* to discover size and shape differences of the gut and trachea between males and females. The topic is extremely interesting and the data presented clearly documents dimorphic organ features, and clearly shows a relationship between the shape of one organ and another. Where the manuscript appears to fall short relates to the authors' claim that "these results confirm a role for tracheal cells in extrinsically controlling sex differences in gut shape." The data, in my view, does not show this *per se*. Additionally, while the differences are well documented, it remains unclear if these sexually dimorphic differences are important for organ/animal function.

Specific comments

Technical achievement

The authors have nicely implemented micro-CT for their analysis and mention "scaling up" the approach. This is an important obstacle for the field to overcome, but the authors don't state how this was done. Were more than one adult fly scanned simultaneously, was the process sped up in some way, or do they mean to say they scanned more flies?

Documenting 3D position and sex difference

The section on pages 2 and 3 nicely describe difference between the male and female gut and the relationship with other organs – testes, ovaries and crop. The work here is nicely described and quantified. I have a few comments regarding the language used: 1) I was not sure why the title stated that "organs assumed to be distant are geometrically adjacent" was used as I am not sure who assumed this (anyone working with flies would likely not). 2) Similarly with the phrase "initially

anticipated” it was not clear who anticipated larger distances. Is this a common and published view in the field of organ geometry? 3) Finally, when referring to “novel methods”, what exactly is meant? Are the authors referring to their measurements?

One interesting observation that was not highlighted is that the ovoD1 resulted in shorter loop length and shorter gut length. To me this was counter intuitive as I would have thought that the ovary would spatially restrict gut growth, while in fact the ovary promotes gut growth.

Co-variation

The authors nicely document the covariation of the trachea density and gut curvature. Key to my comments below, however, is that the correlation holds true for both male and female (Figure 2b and supp 3a). The authors then reasonably hypothesize that there is a causal relationship between trachea and gut. In support of this the authors manipulate the trachea and see an effect on the gut. One experiment used to show this was to downregulate the FGF ligand “branchless” (Bnl) secreted by the gut, which results in a reduction in the terminal tracheal cells and also a change in gut shape.

The authors conclude from this experiment that the gut shape is influenced by the trachea (first sentence, last paragraph, page 4). This is a reasonable hypothesis, but the data do not show this per se. To place trachea upstream of gut the authors would have to show that the loss of the FGF ligand has no effect on the gut directly, which would require experimental conditions that would maintain WT trachea in a mex-Gal4 x bnlRNAi background. Thus the conclusion through page 4 is that there is a correlation between gut shape and trachea abundance/shape.

The relationship between gut and trachea sexual dimorphisms

Now the authors get to the crux of the study, which is well phrased in their question “Might sex differences in tracheation extrinsically impart sex difference to gut shape?” and “we wondered whether the sex differences in gut derived Bnl may extrinsically sculpt sex differences in the tracheal network.”

Unfortunately the attempt to masculinize the trachea did not work. However, the authors were successful in “masculinizing bnl expression” in female and male guts, showing a reduction in bnl in females but not males. This is consistent with the higher female expression of gut bnl and their previous experiment of bnl knockdown showing a change in only female gut curvature (torsion, radius and tilt are not measured).

Here is where the interpretation gets tricky. The authors conclusion that “these results confirm a role for tracheal cells in extrinsically controlling sex difference in gut shape”. I would agree that there is a relationship between the trachea and gut in the female, although it is not clear if it is gut \diamond trachea vs trachea \diamond gut, but that’s not too important. The more important thing is that the authors have not shown that changes in the gut (or trachea) do NOT lead to changes in the trachea (or gut) in the male. Without this, the data from the female really only shows that the gut and trachea shape are related, not that the relationship is female specific. If the authors didn’t mean to say this (or I have misunderstood), then why say “sex difference” instead of just saying “differences” in gut shape. Regardless of sex, gut curvature and trachea amount are related (Figure 2b and supp 3a). If the authors could reduce tracheation in the male by 50%, what would happen? Their natural variation (Figure 2b and supp 3a) within each sex shows that they are not different.

To summarize, this study shows that the gut is sexually dimorphic, the tracheal is sexually dimorphic, and their shapes are related to one another regardless of sex. They have not shown that the co-dependency of the trachea and gut is sexually dimorphic.

The importance of organ differences to the animal

While the authors present possible roles for the organ dimorphism, there is no test for a possible role. Have the authors identified an advantage for a sex specific organ shape? Have they attempted competition or behavioral assays in cases where they manipulated female trachea and gut? How does female organ shape convey an advantage (to the female) over a male organ shape? Is feeding, animal or egg size, age, fecundity, or anything else different?

Identifying such an advantage would constitute a major finding.

Minor

- Page 2: change “relatively shorter in females” to “accounts for a smaller percentage of gut length in females”.

- I found that the drawings on the axes of many graphs such as 1e and 1f were uninterpretable.

Referee #3 (Remarks to the Author):

In this manuscript the authors provide the first detailed study of the three-dimensional organization of the *Drosophila* gut in both male and female adult flies. They use micro-CT and rigorous mathematical approaches to quantify their findings. The manuscript has a lot of data and will be a valuable resource for workers in this field. The authors draw several conclusions from their analysis:

(1) The guts of male and female flies are different in many ways - their length, their thickness, their curvature and torsion.

(2) The male and female guts are (not unexpectedly) adjacent to different organs most notably the ovaries and testes. Interestingly, the left and right ovaries seem to contact different portions of the gut.

(3) The tracheal system seems to hold non-adjacent portions of the gut together

(4) The gut seems to play an instructive role in determining the branching of the tracheal system which in turn seems to stabilize gut conformations.

The authors suggest that the proximity of different portions of the gut to each other or of parts of the gut to other organs might facilitate communication between the juxtaposed elements. They also suggest that the internal organization (tight packing) might be a way of confining signals to local regions of the abdomen. While the data presented in the paper are beautiful, I was left wondering how much evidence they provide for their conjectures.

The main weaknesses of the manuscript are:

1) The lack of any functional data with respect to the importance of the three-dimensional organization that they report. For example, why does it matter that the left ovary is close to the

posterior midgut and the right ovary is next to the crop. Is there any evidence that the left and right ovary do different things.

2) The authors present some numbers without any context which makes the reader wonder why this is important or unexpected. They say that most organs are more tightly packed than originally anticipated (most inter-organ distances are 5-500 micrometers apart). Why is this surprising given the size of the insect? Presumably they are even smaller in smaller insects and larger in larger insects.

3) The authors mention that the hemolymph is viscous yet there is no comparison with other systems. Is it much more viscous than plasma in vertebrates? How far would typical hemolymph proteins diffuse over what time frames? Is it possible to address this experimentally by injecting dyes or macromolecules of specific sizes to assess whether long-range endocrine communication is possible? Otherwise there is little support for their conjecture that: "We propose that the same signal can be used to convey a different message between organs A and B and between organs C and D because D is never within reach of A."

4) There is little reference to research in vertebrates except at the very end. Surely physiologists must have explored these questions in mammals where the locations of organs can be experimentally manipulated. Many organs seem to function reasonably well after operations where they are moved around a lot.

For all of these reasons I feel that there is a big gap between all the beautiful data in the paper and their attempts to demonstrate a functional importance for these findings.

Minor point: The authors mention "Of note, we have observed that the white genetic background commonly used as a "wild-type" stock has reduced tracheation compared to truly wild-type flies" The work of Sasaki et al (2021) PMID: 33820991 shows that intestinal stem cells in white mutants are different at least in the context of aging - maybe there are differences that could explain what the authors are seeing as well.

In summary, this is a beautiful descriptive study that will form a solid foundation for exploring the relevance of the proximity of different organs to each other in the *Drosophila* abdomen. It is unclear as yet as to whether these stereotypic and unexpected proximities are of functional relevance.

Referee #4 (Remarks to the Author):

In this study, Blackie et al. present a collaborative effort from Miguel-Aliaga and Mahadevan labs, presenting an intriguing finding about how organs maintain sexually-dimorphic shape. This paper shows how bidirectional signaling between the gut and tracheal system in *Drosophila* maintains the female gut shape. Based on this finding, they propose an idea of ‘vascular sex’ – that insect tracheae/ and perhaps the equivalent vasculature- encode a program that enables organs to maintain a sexually dimorphic shape, akin to a corset. Importantly, they present compelling data to test this complex theory using a combination of cutting-edge quantitative 3D-volumetric imaging and genetics. Specifically, the authors developed a 3D-volumetric scan method for imaging internal organs for large numbers of fruit flies and a quantitative image analysis pipeline; this allowed them to characterize the inter-organ adjacencies and left-right asymmetries in detail. It is to be noted that this information alone if made widely available along with tools needed for such analysis, will be a highly valuable community resource for the *Drosophila* physiology community. As the authors discuss, the information from such a characterization is vast and can be used to test many paradoxes in the organ specificity of such hormonal signals. For this study, the authors have focused on one important aspect- “How do sex differences in 3D-organ geometry arise?” by focusing on the sexual dimorphism of 3D-gut loops. First, they test the obvious hypothesis that these gut loops, which are adjacent to ovaries, maintain their shapes because of adjacent organs, such as ovaries. However, they found that despite genetically reducing the size of ovaries, the dimorphism of the gut was maintained, suggesting the existence of other mechanisms by which the gut maintains shape. In their quest to identify this non-cell-intrinsic sexual dimorphism, they identified a cross-talk between the gut and trachea that enables the trachea to maintain the female-specific gut shape. This reviewer finds the study to be original, thought-provoking, and of wide interest. The manuscript is a pleasure; the statistics were appropriately applied and described. But before publication, I would encourage the authors to consider revising the manuscript in the following ways to increase the functional impact of their study (see below) and provide further information that would make their 3D technology more widely accessible to other researchers.

1.

a. What is the functional impact of masculinizing gut shape in female flies? Have they performed any functional tests such as its effect on food transit, feeding patterns, reproductive status, and recovery of ISCs from insults such as DSS in female flies with masculinized gut shape (e.g., *mex-Gal4>bnl-RNAi*; *Hand>bnl-RNAi*)? Doing so will be important to close the loop on the idea that maintenance of gut shape by trachea has functional physiological relevance. Even if the relevance is not obvious, presenting this “negative” data would be important, as it would allow the reader to understand the importance of 3D shape in function.

b. It is possible that in a baseline physiological state, the 3D shape dimorphism of the gut has minimal impact on physiology – feeding, reproduction, recovery from DSS, etc. - but what happens if the system is stressed? For instance, these could compare non-virgin and virgin females; flies fed an obesogenic regime versus normal diets, etc.,

2. The methods presented for MicroCT scans need to be more detailed- preferably a working protocol- to enable other researchers to use this important method more widely. Similarly, is there an open source website such as GitHub where the code for segmentation, curvature comparison, etc., can be posted for wider access by others?

Minor comments:

1. What is DSRF stand for? A reference for the appropriate paper and further explanation of methods for that this is a standard stain for terminal track nuclei must be elaborated.
2. What stringerGFP is used for nuclei tracking? Please provide the info.

Referee #5 (Remarks to the Author):

In this interesting manuscript, Blackie et al. use micro-computed tomography to demonstrate quantitatively the 3D morphology and spatial arrangement of the internal organs of the adult fruit fly, with a special focus on the intestine, which spans the anterior-posterior axis of the animal body. They use volumetric scans in combination with computer graphics' methods to segment internal organs, visualize previously unnoticed left-right asymmetries, identify sex differences in organ shape and quantify organ-organ proximities. Specifically, by imaging large number of flies, they show that the male and female adult intestine differ in shape especially at the central loop region, with the female gut being longer and more curved. Then, they go ahead to explain gut shape sex differences by assessing the role of the vasculature-like trachea that contacts the intestine throughout its length. They show that the intestinal tracheal branches mechanically hold the gut loops together and contribute to gut shape in a sex-specific manner. They propose that, in response to the sex determination pathway, the FGF ligand Branchless (Bnl) produced by gut cells at different levels/expression patterns in males and females is key to determining male vs female gut shape. The experiments are clearly presented and, for the most part, are well-executed. The work is timely and will be of interest to broad readership, since it describes quantitatively and in detail the 3D arrangement and spatial proximities of an adult animal's internal organs, which might explain organ communication in space.

Please see below specific points that the authors should address before publication.

1. The tomographic work is elegant and nicely shows the adult fly spatial internal organ arrangement. The authors use wild-type OregonR flies to perform their analyses. To exclude the possibility that organ shape is affected by the genetic background, did they also observe the same spatial organ relationships in other wild-type genotypes?
2. Figure 1a-b: although the tomography images are clear, for the non-expert it would be useful to add the V (ventral) and D (dorsal) annotation in the figure or legend.
3. Figure 1c: the authors should correct the regional annotations in the drawing. What they annotate as "anterior midgut" corresponds to R1 and R1 is never found in the gut loops according to published work on regionalization.
4. Page 3: "...most inter-organ distances are in the 5-500 μm range (Fig. 1a,b, Extended Data Table 1)." There is no information in Fig. 1a, b on interorgan distances. The authors should refer to Fig. 1k-m here and relevant information in Ext. Data Fig. 1g-h, l-o.
5. Page 3: The last sentence of the second paragraph is unclear. "... they are meant to elicit? We propose that the same signal can be used to convey a different message between organs A and B and between organs C and D because D is never within reach of A." Maybe the authors mean: We propose that the same signal can be used to convey a different message between organs A and B and between organs A and C because C is never within reach of A?
6. Page 3 and Ext. Data Fig. 2: the ovoD1 females have shorter guts/gut loops. Is their overall body

size smaller than controls? Also, is the intestinal tracheation of ovoD1 females of the same extent as in the controls? It would be interesting to show the internal organ-organ proximities of the ovoD1 fly in comparison to the control.

7. The authors use a *bnl*-LexA reporter to show the *bnl* expression pattern in the midgut. They also generated a *bnl*-Gal4 reporter from the original LexA line, which does not lack any exons (does the latter express full length *bnl*? I did not see the data). They show that this reporter is expressed in the gut enterocytes (ECs) in larvae, and then in adults, as well as in the gut VM transiently in young adults (5 hrs APE); thus, they focus their silencing experiments in the ECs and the VM. Since the Bnl ligand is known to be very dynamic during development, it is possible that the reporters do not recapitulate the full expression pattern of *bnl*. In addition, it has been shown (Perochon et al, 2021; Tamamouna et al, 2021) that tracheal *bnl* is involved in visceral tracheal branching in adults in response to damage and tumors. Have the authors tested other potential *bnl* sources, e.g. the trachea, that may affect gut shape apart from the ECs?

8. Figure 2: The authors use Hand-Gal4 to silence *bnl* (and later *tra*) specifically in the VM, but they do not actually show the specificity of expression of this driver. Maybe I have missed it, but I could not find any information about it either in the manuscript or in Flybase. If this is true, the authors should show the detailed characterization of the Hand-Gal4 driver (or provide a reference for its expression pattern).

9. Does VM-specific *bnl* silencing affect gut length? Also, expression of the *bnl*RNAi under Hand-Gal4 is induced early during development (without Gal80ts), so the lack of tracheal cells can be interpreted as a developmental phenotype. What happens to the tracheal cells? Are they not specified or do they die after they are born? In addition, since VM-specific *bnl* silencing is a very important experiment for their study, the authors should use alternate VM-specific drivers (e.g., *how*-Gal4, which is also expressed in the trachea in larvae Aghajanian et al, Dev. Biol. 2016) to knock down *bnl* expression and assess gut shape etc, and silencing should be ideally restricted to adult stages. Alternatively, have the authors tried to silence *bnl* using the *bnl*-Gal4 they generated in this study? This will be informative too. Finally, is VM-specific overexpression of *bnl* sufficient to change tracheation and the shape/looping of the gut?

10. Figure 2h: it is not clear from the legend or the methods whether the number of DSRF-positive cells and tracheal length are normalized to the gut surface; they should be.

11. Figure 3c-d; Figure 4c: please see comment 10.

12. The authors observe sex differences in tracheal coverage and *bnl* abundance and since they cannot masculinize the female tracheal cells (Ext. Data Fig. 7a-h), they decide to masculinize ligand expression by silencing *tra* in the female VM (Fig. 4 and Ext. Data Fig. 7). Did the authors check whether the genes involved in the sex determination pathway are expressed in the tracheal cells and the VM (as the same lab had done in Hurdy et al, 2016)? The *tra*RNAi is a key experiment and should be validated with additional *tra*RNAi lines, as well as with other genes involved in the sex determination pathway. For example, does silencing of *Sxl* or *tra2* affect gut *bnl* expression? It seems that although Hand-Gal4>*tra*RNAi-mediated female *bnl* masculinization reduces gut tracheal coverage and gut length (note that the tracheal features in Fig. 7c should be normalized to gut surface), the gut shape and curvature pattern does not change much as shown in Fig. 4d-f. Finally, can overexpression of *traF* in males feminize *bnl* expression, tracheation and gut shape?

13. Ext. Data Fig. 4b: in the single channel *bnl*-Gal4>Stinger-GFP images, the same guts should be shown in the top and bottom panels to indicate ECs and VM in the same gut region.

14. Ext. Data Fig. 5, Fig.6 and Fig. 7: all tracheal features measured should be normalized to the gut

surface.

15. All figures (main and supplementary) need titles.

16. The introduction is missing. Although there might be a length limit, the authors should at least introduce their study in one paragraph.

17. Page 4, top paragraph: “intriguingly” is used in 2 consecutive sentences.

18. Figure 3 is described in the text (Page 4, second paragraph) before the second half of Figure 2. All figures should be described in the order they are presented.

Author Rebuttals to Initial Comments:

We are pleased by how our manuscript was received by the reviewers, and would like to thank all five reviewers for the quality of their reviews; we feel that their complementary expertise and collective feedback has greatly improved our manuscript.

We have worked hard to address all their comments and suggestions in full. We have conducted 36 new experiments leading to the incorporation of new figure panels (13 in the main figures, and 43 provided as Extended Data), 1 Supplementary Video and 6 Figures for the Reviewers. Notably, we have incorporated new data in the revised manuscript that:

- 1) strengthen the conclusion that trachea have a novel structural role in “stitching” gut loops together
- 2) strengthen the conclusion that the sexual identity of gut muscles extrinsically controls sexually dimorphic tracheal branching to control gut shape
- 3) provide evidence that the genetic manipulations that impact gut shape are functionally relevant

We have also amended the wording of the main text and made a clearer distinction between results and hypotheses/discussion wherever appropriate.

Details of all changes are provided below in our point-by-point response; figures for reviewers are provided at the end of this document.

Referee #1 (Remarks to the Author):

Reviewer Report: Nature MS "The Sex of Organ Geometry" By Blackie et al.

*In this manuscript, the authors present an outstanding piece of research that sheds light on the often-overlooked relationship between organ spatial arrangements and their functionality and communication. This work constitutes a masterful contribution of the highest quality, employing innovative techniques to quantify 3D features of organ shape, position, and inter-individual variability in *Drosophila melanogaster* flies. The hypothesis that there exists a logic to the shape and proximity of organs is intriguing, and the use of high-content volumetric scans to analyze large numbers of *Drosophila* flies offers an impressive and comprehensive dataset to address this question.*

One of the most striking findings of this study is the revelation of stereotypical yet sexually dimorphic shapes of organs and their relative arrangements. This groundbreaking discovery challenges our current understanding of organ development and raises important questions about the underlying mechanisms responsible for these differences. The authors demonstrate an exemplary focus on the intestine, a highly relevant and complex organ whose sexually dimorphic shape is actively maintained by the tracheal system. This finding represents a significant advancement in our knowledge of the factors influencing organ morphology and how they may contribute to signaling paradoxes and sex differences in organ function. Moreover, the identification of a previously unrecognized bidirectional mechanism of sex differentiation in the tracheal system, rendering it sexually dimorphic, adds a new layer of biological complexity that has not been previously recognized. This discovery opens up exciting

avenues of research, both in understanding sex-specific differences in organ geometry and in unraveling the broader implications of the tracheal system's role in shaping organs.

The conclusions drawn from this work are undoubtedly novel and original, significantly expanding our understanding of the relationship between organ geometry and sexual dimorphism. It has the potential to prompt a paradigm shift in how we perceive and study organ development and function. The outlook of this work is highly impressive as it holds great promise in advancing research in health and disease. The newfound knowledge of how organs' spatial arrangements impact communication and functionality is invaluable and will likely have far-reaching implications in medicine and biology.

In summary, the study "The Sex of Organ Geometry" is a remarkable achievement of the highest quality. Its innovative methods, insightful findings, and groundbreaking conclusions make it a milestone in the scientific community. The implications of this research are profound, and it will undoubtedly inspire and fuel further investigations in this exciting area of study. I recommend its publication without hesitation.

I feel the story would be even more complete and impactful if the authors discussed or even better included additional experimental work addressing the following points:

- *Can an intestinal functional output (Transit time? Fecal deposits appearance?) be assayed in this work to probe the extent to which sex and the genetic manipulation impacting gut geometry used in the work are influencing intestinal biology?*

We have tackled this question by focusing on flies with specific loss of gut trachea (achieved by gut muscle-specific downregulation of the FGF ligand *bnl*, *Hand>bnl-RNAi*). This genetic manipulation leads to loss of gut trachea and reduced gut curvature in both males and females. It is also the genetic manipulation that more specifically affects gut trachea without affecting the gut itself (gut length is unaffected in males and only modestly affected in females), other trachea in the fly (as would be the case for the tracheal ablation, *trh^{TS}>Bax*) or, potentially, other aspects of gut muscle sexual dimorphism (as *Hand>Sxl-RNAi* or *Hand>tra-RNAi* may conceivably impact).

We have assessed how this genetic manipulation affects intestinal features such as transit, excretion, intestinal stem cell proliferation and hypoxia – as one might have expected this pathway to be activated by the lack of trachea.

Interestingly, we observe no or modest effects on any of these intestinal features. In flies which lack gut trachea, excretion is normal and transit is not affected in males (it is a bit faster in females, but this phenotype is no longer apparent when only guts of comparable length are assessed). In normal homeostatic conditions, intestinal stem cell proliferation is also unaffected. Particularly striking are the observations we have made using two independent hypoxia reporters: 1) gut hypoxia appears to be constitutive even in the presence of trachea, and 2) gut hypoxia is not further exacerbated by the absence of gut trachea. Together, these results (which are now shown in Extended Data Figs. 9 and 10) strongly suggest that the main role of trachea in homeostatic conditions is not to deliver oxygen to the gut but, rather, to hold gut loops together. Of note, we have further validated this idea by conducting laser ablation experiments (see response to your next point below).

Loss of gut trachea does, however, have functional consequences at the whole-organism level: although it does not affect food intake, we have observed that it reduces fecundity in females and also leads to reduced ability to withstand starvation in both sexes. These data are now shown in Extended Data Fig. 10d-f, i-k. A future goal will be to establish how gut trachea sustain female fecundity; changes in the adjacency and communication between the ovaries and specific gut portions is an intriguing possibility in this regard. This is now discussed in the revised discussion.

• *In the final paragraph of the manuscript's page 5, the authors focus on *bnl* expression in gut muscles. By downregulating *tra* expression in female gut muscles, they manage to antagonize *bnl* expression to "masculinize" it (i.e., reduce levels vs. female levels). While I agree that there is a clear effect on *bnl* expression, I believe such masculinization by manipulating *tra* levels will be pleiotropic; hence, I feel it would be good to provide a rescue experiment by selectively restoring *bnl* expression in a masculinized female gut and check the extent of gut geometry change (and physiological outputs if possible, see the point above).*

We agree with the reviewer and had, in fact, planned to do this experiment. The difficulty we encountered is that ectopic expression of *bnl* leads to enlarged or malformed guts, which would confound gut shape analysis. Indeed, strong *bnl* over-expression using *UAS-bnl¹* leads to dramatic increase in the number of terminal tracheal cells (as assessed with DSRF expression), far above the numbers seen in wild-type females (see Figure for Reviewer R1). We also attempted milder *bnl* over-expression using *UAS-bnlSAM* (CRISPR-based overexpression from the endogenous *bnl* promoter²) but this also increased gut diameter. Although these manipulations do affect gut shape in both males and females (see Figure for Reviewer R1), we are reluctant to use these *bnl* expression tools to conduct rescue experiments because of their ectopic effects on the gut itself, which we have not observed in other manipulations.

We have, nonetheless, conducted two alternative sets of experiments that strengthen the conclusion that muscle-derived *bnl* controls gut shape via its effect on trachea.

Firstly, we have complemented the gut muscle *tra-RNAi* masculinisation experiment in females by conducting a feminisation experiment in males. We have expressed *UAS-Sxl* in gut muscles and have observed that this feminises muscle *bnl* expression levels and increases the number of tracheal terminal tracheal cells in males, to levels comparable to those in females. As expected, this genetic manipulation has no effects in females. Importantly, and opposite to the previous masculinisation experiment, this manipulation affects gut shape in males, but not in females. These data are now shown in Fig. 4g-l and Extended Data Fig. 8j,k. We have also conducted an additional masculinisation experiment (gut muscle-specific *UAS-Sxl-RNAi*) which has confirmed the initial observations we had made with the gut muscle-specific, *tra*-mediated masculinisation (now shown in Fig. 4a-f, Extended Data Fig. 8h,i and Extended Data Fig. 8a-g, respectively).

In parallel, we have tested the idea that trachea hold gut loops together by conducting laser ablations of the tracheal branches that span adjacent gut regions *ex vivo*. We have observed ablation-induced tracheal recoil in 22 out of 25 guts, confirming that trachea hold tension, at

least *ex vivo*. These data, which is now shown in Supplementary Movie 2, further lends support to the idea that tracheal branches hold gut loops together.

• *In general in the work, the dynamics of ISCs proliferation at steady state between males and females and/or during the genetic manipulation impacting gut geometry are not discussed nor investigated. This parameter is likely to impact gut shape, especially if this is region-restricted/specific, right? I feel it is probably important to integrate this parameter in the discussion/investigation?*

We have quantified stem cell proliferation in the guts of both males and females lacking gut trachea (*Hand>bnl-RNAi* flies). We have observed no effects in normal homeostatic conditions. This result, shown in Extended Data Fig. 10a, rules out changes in intestinal proliferation as the reason for the effect of tracheal ablation on gut shape, and is in line with the idea that the intestinal epithelium does not have a major requirement for tracheal oxygen supply in these conditions.

We have also tested intestinal proliferation in two hyperproliferative contexts: ageing³⁻⁵ and detergent (DSS) treatment⁶. Intriguingly, DSS-induced hyperproliferation is blunted in female flies following gut muscle-specific *bnl* depletion, leading to loss of gut trachea. By contrast, age-related hyperproliferation is further increased in these flies above the levels seen in control flies; again, the effect is particularly prominent in females. These data are now shown in Extended Data 10b,c. Although future work will be required to understand the differential effects of *bnl* and tracheal loss on homeostatic, age-related and damage-induced proliferation, our findings raise the possibility that age-related intestinal stem cell hyperproliferation (which is also female-biased) is at least partly caused by age-related loss in gut trachea. This is now discussed in the revised manuscript.

Referee #2 (Remarks to the Author):

The manuscript by Blackie et al reports on organ shape sexual dimorphism. The researches use micro-CT of adult Drosophila to discover size and shape differences of the gut and trachea between males and females. The topic is extremely interesting and the data presented clearly documents dimorphic organ features, and clearly shows a relationship between the shape of one organ and another. Where the manuscript appears to fall short relates to the authors' claim that "these results confirm a role for tracheal cells in extrinsically controlling sex differences in gut shape." The data, in my view, does not show this per se. Additionally, while the differences are well documented, it remains unclear if these sexually dimorphic differences are important for organ/animal function.

We provide additional data which we believe addresses both points. Please see detailed responses to the points below.

Specific comments

Technical achievement

The authors have nicely implemented micro-CT for their analysis and mention "scaling up" the approach. This is an important obstacle for the field to overcome, but the authors don't state how this

was done. Were more than one adult fly scanned simultaneously, was the process sped up in some way, or do they mean to say they scanned more flies?

Yes, we are now able to scan 40 flies at once, as illustrated in two new figure panels (Extended Data Fig. 1a,b). As well as providing these images, we provide full details of how we scaled up the scans in our Methods.

Documenting 3D position and sex difference

The section on pages 2 and 3 nicely describe difference between the male and female gut and the relationship with other organs – testes, ovaries and crop. The work here is nicely described and quantified. I have a few comments regarding the language used:

- 1) I was not sure why the title stated that “organs assumed to be distant are geometrically adjacent” was used as I am not sure who assumed this (anyone working with flies would likely not).*
- 2) Similarly with the phrase “initially anticipated” it was not clear who anticipated larger distances. Is this a common and published view in the field of organ geometry?*

Apologies that this was unclear, and point taken that we should have made a clearer distinction between results and our interpretations/impressions.

To clarify our statement: before acquiring these 3D data, we and other *Drosophila* researchers would not have necessarily anticipated that the fly stomach (crop) would be adjacent to an ovary, or that the anterior midgut would abut a portion of the central nervous system. With regard to the larger distances: current models of inter-organ communication rely on the principle that most, if not all, signals are endocrine to relay information systemically over large distances. Our observations in adults, in particular the < 10µm distances between some organs, raise the possibility that at least some of these signals may act within a shorter paracrine range between organs to mediate those communications.

We do, nonetheless, acknowledge that these statements came across as gratuitous/unfounded in the previous version of the manuscript. To address this, we have entirely separated facts from fiction (namely, Results from Discussion) in our revised manuscript, and have confined these hypothetical considerations to the revised Discussion.

- 3) Finally, when referring to “novel methods”, what exactly is meant? Are the authors referring to their measurements?*

We are using methods that have not been previously used to analyse and/or quantify gut shape and inter-organ distances, such as a neurite tracer, 3D Procrustes analysis, mesh analysis and multidimensional scaling. We recognise that “novel” was ambiguous and have replaced this in the revised manuscript with a detailed explanation of our approaches in the Methods. We also provide access to all our new code on GitHub (<https://github.com/Lblackie1/OrganGeometry>).

One interesting observation that was not highlighted is that the ovoD1 resulted in shorter loop length and shorter gut length. To me this was counter intuitive as I would have thought that the ovary would spatially restrict gut growth, while in fact the ovary promotes gut growth.

There are two aspects to this. The reduced midgut length observed in *ovoD1* mutants is, to some extent, expected given the reported effect of ecdysone (a hormone at least partly produced by the ovaries) on promoting intestinal stem cell proliferation and gut growth; ecdysone production is impaired in *ovoD1* mutants^{7,8}. This is clarified in the first Results section.

That said, the ovary does spatially restrict gut shape. This is demonstrated both by the analysis of *ovoD1* mutants, in which the gut occupies a position that would be occupied by ovaries in control female flies, as well as the co-variation analysis in control female flies, in which differences in gut shape significantly correlate with ovary volume. These data are now shown in Extended Data Fig. 1o-r, Extended Data Fig. 2p-r and Supplementary Tables 17-19.

We have considered this comment further and have expanded on our initial *ovoD1* analysis to investigate whether, as well as changing the shape and position of the intestine, reduced ovary size further impacts inter-organ distance. This is indeed the case and is now shown in Extended Data Fig. 2j.

Co-variation

The authors nicely document the covariation of the trachea density and gut curvature. Key to my comments below, however, is that the correlation holds true for both male and female (Figure 2b and supp 3a). The authors then reasonably hypothesize that there is a causal relationship between trachea and gut. In support of this the authors manipulate the trachea and see an effect on the gut. One experiment used to show this was to downregulate the FGF ligand “branchless” (Bnl) secreted by the gut, which results in a reduction in the terminal tracheal cells and also a change in gut shape.

*The authors conclude from this experiment that the gut shape is influenced by the trachea (first sentence, last paragraph, page 4). This is a reasonable hypothesis, but the data do not show this per se. To place trachea upstream of gut the authors would have to show that the loss of the FGF ligand has no effect on the gut directly, which would require experimental conditions that would maintain WT trachea in a *mex-Gal4 x bnlRNAi* background. Thus the conclusion through page 4 is that there is a correlation between gut shape and trachea abundance/shape.*

We have conducted a series of experiments to address these points.

Firstly, we note that we have shown that trachea control gut shape not only by downregulating the FGF ligand *bnl* from gut muscle (which results in loss of gut trachea and reduced gut curvature in both males and females) but, importantly, by acutely ablating trachea without affecting any other intestinal cell types (achieved by adult-specific and temporally restricted expression of the pro-apoptotic gene *Bax* from a terminal tracheal cell-specific driver, *trh>Bax^{TS}*). This genetic manipulation leads to loss of terminal tracheal cells on the gut and also reduces gut curvature in both males and females. This is shown in Fig. 2d-g.

We nonetheless acknowledge the reviewer’s point that the loss of *bnl* could affect gut shape by acting on the gut itself rather than its trachea. We have ruled this out by downregulating *Btl* (the receptor for *bnl*) in gut epithelial cells (*mex1, esg^{TS} > btlRNAi*). In contrast to the tracheal manipulations, this had no effect on gut tracheation or shape. These data are now shown in Extended Data 5c,d,g,h.

Furthermore, we have strengthened the model that gut-derived FGF ligand controls tracheal branching which, in turn, controls gut shape by comparing the effect of downregulating the FGF ligand Bnl specifically from gut muscles (which we had already targeted by means of *Hand>bnl-RNAi*) vs new experiments in which we have downregulated Bnl expression specifically in intestinal enterocytes, as this is the only other intestinal cell type in which we have observed expression of Bnl (Fig. 3a and Extended Data Fig. 4a,c,d). Only the muscle-specific (but not the epithelium-specific) *bnl* downregulations affect tracheal branching and gut shape. These data are now shown in Fig. 2h-k and Extended Data Fig. 5a,b,e,f).

Finally, we have directly tested the idea that trachea hold gut loops together by conducting laser ablations of the tracheal branches that span adjacent gut regions *ex vivo*. We have observed ablation-induced tracheal recoil in 22 out of 25 guts, confirming that trachea hold tension, at least *ex vivo*. These data, which is now shown in Supplementary Movie 2, further lend support to the idea that tracheal branches hold gut loops together.

The relationship between gut and trachea sexual dimorphisms

Now the authors get to the crux of the study, which is well phrased in their question “Might sex differences in tracheation extrinsically impart sex difference to gut shape?” and “we wondered whether the sex differences in gut derived Bnl may extrinsically sculpt sex differences in the tracheal network.”

Unfortunately the attempt to masculinize the trachea did not work. However, the authors were successful in “masculinizing bnl expression” in female and male guts, showing a reduction in bnl in females but not males. This is consistent with the higher female expression of gut bnl and their previous experiment of bnl knockdown showing a change in only female gut curvature (torsion, radius and tilt are not measured).

Here is where the interpretation gets tricky. The authors conclusion that “these results confirm a role for tracheal cells in extrinsically controlling sex difference in gut shape”. I would agree that there is a relationship between the trachea and gut in the female, although it is not clear if it is gut \leftrightarrow trachea vs trachea \leftrightarrow gut, but that’s not too important. The more important thing is that the authors have not shown that changes in the gut (or trachea) do NOT lead to changes in the trachea (or gut) in the male.

Without this, the data from the female really only shows that the gut and trachea shape are related, not that the relationship is female specific. If the authors didn’t mean to say this (or I have misunderstood), then why say “sex difference” instead of just saying “differences” in gut shape. Regardless of sex, gut curvature and trachea amount are related (Figure 2b and supp 3a). If the authors could reduce tracheation in the male by 50%, what would happen?

Their natural variation (Figure 2b and supp 3a) within each sex shows that they are not different.

To summarize, this study shows that the gut is sexually dimorphic, the tracheal is sexually dimorphic, and their shapes are related to one another regardless of sex. They have not shown that the co-dependency of the trachea and gut is sexually dimorphic.

These concerns require some clarifications and some additional experiments, which we have now performed.

Firstly, it is important to clarify that both tracheal and gut muscle cells have an intrinsic sexual identity which can be genetically altered. Indeed, they both express the sex determinant TraF only in females but not in males – see Figure for Reviewer R2. The “failed” masculinisation of trachea was not failed in the sense that we are unable to downregulate TraF expression in females using *tra-RNAi*, but rather that the sexual dimorphism in tracheal branching and number is not intrinsically controlled from the tracheal cells themselves. Effectiveness of the *tra-RNAi* transgene is, in fact, demonstrated in this study when we masculinise muscle features (e.g. *bnl* expression) by expressing the same *tra-RNAi* transgene from a muscle rather than tracheal driver (see Extended Data Fig. 8a-g). Additionally, we have downregulated *tra* in trachea with multiple independent *tra-RNAi* transgenes, which results in the same lack of masculinisation of the tracheal network. These data are now shown Extended Data Fig. 7. These transgenes also masculinise other intestinal cell types as we have done and published several times before^{9,10}). The finding (which is in our opinion an important one in the sex determination field) is that, in this particular case, the sexual dimorphism in tracheal cell branching and number is not controlled from the tracheal cells themselves by the intrinsic sex determination pathway (namely, TraF and its upstream regulator Sxl in females, see¹¹ for a review) but, rather, extrinsically by the levels of the gut muscle-derived FGF ligand Bnl.

It is also important to clarify that we agree with the reviewer that the mechanism by which trachea controls gut shape does NOT differ between the sexes; in both cases, and for the reasons stated above in response to the previous point, we believe that trachea do so by structurally holding gut loops together through their branches. The sexual dimorphism stems instead from differential tracheal branching density in males and females, which is ultimately dictated by the sex of the gut muscles; female muscles make more Bnl ligand, which makes more tracheal cells and more branching, increasing curvature. Less Bnl ligand in male muscles leads to less trachea and less curvature. In the complete absence of Bnl, all gut trachea are lost and curvature is strikingly reduced in both males and females.

There were several experiments in the previous version of the manuscript that supported this model:

1. The total lack of gut trachea (achieved either by terminal tracheal cell ablation or by depleting Bnl specifically from gut muscles) reduces curvature in both males and females (Fig. 2d-g). Importantly, in terms of directionality/causality, the terminal tracheal cell ablation targets trachea without targeting any gut cells.
2. Tracheal cell number/branching are higher in females, and the expression levels of the FGF ligand are higher in the gut muscles of females than those of males (Fig. 3).
3. Genetic and cell-intrinsic masculinisation of gut muscles (achieved by gut-muscle specific downregulation of the intrinsic sex determinant *tra* coding for the feminising protein TraF) masculinises the expression levels of the FGF ligand Bnl in females, without affecting expression in males (Extended Data Fig. 8a, b). This manipulation reduces tracheation and gut curvature only in females (Extended Data Fig. 8a-g).

We have now tested this model further in several additional ways:

1. The experiments described in response to the previous point, which show that tracheal branches hold gut loops together (e.g. laser ablation) and that the muscle is the relevant Bnl source.
2. An additional gut muscle masculinisation experiment. We have downregulated expression of the upstream intrinsic female sex determinant *Sxl* from gut muscles by means of *Hand-Gal4*-driven *UAS-Sxl-RNAi* expression. Consistent with the *tra-RNAi* masculinisation, we have observed masculinised Bnl expression in gut muscles, and reduced tracheal cell number, branching and reduced gut curvature in females only. This is now shown in Fig. 4a-f and Extended Data Fig. 8h, i.
3. We have complemented the gut muscle *bnl* RNAi masculinisation experiment by conducting a feminisation experiment. To this end, we have expressed *UAS-Sxl* in the gut muscles (males do not normally express this female sex determinant) and have observed that this feminises (increases) Bnl expression levels in the gut muscles of males, and increases the number and branching of terminal tracheal cells only in males, to levels comparable to those in females. As expected, this genetic manipulation has no effects in females. Importantly, and opposite to the previous masculinisation experiment, this feminisation affects gut shape in males, but not in females. This is now shown in Fig. 4g-l and Extended Data Fig. 8j,k.

As well as including these new experiments in the revised version of the manuscript, we have also sought to clarify these points by changing some of the wording throughout the description of our results.

The importance of organ differences to the animal

While the authors present possible roles for the organ dimorphism, there is no test for a possible role. Have the authors identified an advantage for a sex specific organ shape? Have they attempted competition or behavioral assays in cases where they manipulated female trachea and gut? How does female organ shape convey an advantage (to the female) over a male organ shape? Is feeding, animal or egg size, age, fecundity, or anything else different?

Identifying such an advantage would constitute a major finding.

We have explored the functional consequences of our genetic manipulations.

We initially chose to focus on flies with specific loss of gut trachea (achieved by gut muscle-specific downregulation of the FGF ligand *bnl*, *Hand>bnl-RNAi*). Although this genetic manipulation leads to reduced gut curvature in both males and females, it is also the genetic manipulation that more specifically affects gut trachea without affecting the gut itself (gut length is unaffected in males and only modestly affected in females), other trachea in the fly (as would be the case for the tracheal ablation, *trh^{TS}>Bax*) or, potentially, other aspects of gut muscle sexual dimorphism (as *Hand>Sxl-RNAi* or *Hand>tra-RNAi* may conceivably impact). Hence, it is the manipulation that more specifically affects gut shape.

We observe functional consequences at the whole-organism level: although this genetic manipulation does not affect food intake, it does lead to reduced fecundity in females and

reduced ability to withstand starvation in both sexes. These data are now shown in Extended Data Fig. 10d-f, i-k.

Because we were particularly intrigued by the female-specific effect on fecundity, we conducted an additional experiment in which we feminised the gut muscle (by means of *Hand>Sxl-RNAi*). This too resulted in reduced female fecundity and is now shown in Extended Data Fig. 10l.

A future goal will be to establish the link(s) between gut shape and female fecundity; changes in the adjacency and communication between the ovaries and specific gut portions is an intriguing possibility in this regard. This is now discussed in the revised discussion.

Although not requested by this reviewer, we note that we also assessed functional effects on the gut itself; details are provided in response to the questions raised by Reviewer #1, but overall these experiments are consistent with the idea that the main role of trachea in normal homeostatic conditions is a structural one in maintaining gut shape.

Minor

- Page 2: change “relatively shorter in females” to “accounts for a smaller percentage of gut length in females”.

This has now been amended.

- I found that the drawings on the axes of many graphs such as 1e and 1f were uninterpretable.

The average (grey) gut centreline shape and the extremes of variation (black) shapes along each PC are commonly represented in this way in other publications using Procrustes analysis^{12,13}. Classically, these are often displayed as warped outlines of a structure’s shape, overlaid with its average shape. In our case, we have plotted the average coordinates of each Procrustes landmark along the gut to represent the average gut centreline in grey. The black lines represent the coordinates of the landmark points from the hypothetical extremes of variation of each PC. With these drawings, we aim to provide context for where each point in the PCA plot sits in the range of variation in 3D shape mapped by the PCA space. We feel it is important to retain these drawings as they give a visual impression of the nature and degree of variability in 3D shape. We have, however, improved our explanation of these drawings in the Methods, under “Geomorphic morphometrics and PCA analysis”. We also quantify more concrete features such as curvature along gut length in other panels.

Referee #3 (Remarks to the Author):

In this manuscript the authors provide the first detailed study of the three-dimensional organization of the Drosophila gut in both male and female adult flies. They use micro-CT and rigorous mathematical approaches to quantify their findings. The manuscript has a lot of data and will be a valuable resource for workers in this field. The authors draw several conclusions from their analysis:

(1) The guts of male and female flies are different in many ways - their length, their thickness, their curvature and torsion.

(2) The male and female guts are (not unexpectedly) adjacent to different organs most notably the ovaries and testes. Interestingly, the left and right ovaries seem to contact different portions of the gut.

(3) The tracheal system seems to hold non-adjacent portions of the gut together

(4) The gut seems to play an instructive role in determining the branching of the tracheal system which in turn seems to stabilize gut conformations.

The authors suggest that the proximity of different portions of the gut to each other or of parts of the gut to other organs might facilitate communication between the juxtaposed elements. They also suggest that the internal organization (tight packing) might be a way of confining signals to local regions of the abdomen. While the data presented in the paper are beautiful, I was left wondering how much evidence they provide for their conjectures.

The main weaknesses of the manuscript are:

We are glad that the reviewer feel that our manuscript will be a valuable resource. We believe that we have addressed the weaknesses listed below with our additional experiments.

1) The lack of any functional data with respect to the importance of the three-dimensional organization that they report. For example, why does it matter that the left ovary is close to the posterior midgut and the right ovary is next to the crop. Is there any evidence that the left and right ovary do different things.

We have tackled this question by focusing on flies with specific loss of gut trachea (achieved by gut muscle-specific downregulation of the FGF ligand *bnl*, *Hand>bnl-RNAi*). This genetic manipulation leads to reduced gut curvature in both males and females. It is also the genetic manipulation that more specifically affects gut trachea without affecting the gut itself (gut length is unaffected in males and only modestly affected in females), other trachea in the fly (as would be the case for the tracheal ablation, *trh^{TS}>Bax*) or, potentially, other aspects of gut muscle sexual dimorphism (as *Hand>Sxl-RNAi* or *Hand>tra-RNAi* may conceivably impact).

We have assessed how this genetic manipulation affects intestinal features such as transit, excretion, intestinal stem cell proliferation and hypoxia – as one might have expected this pathway to be activated by the lack of trachea.

Interestingly, we observe no or modest effects on any of these intestinal features; in flies which lack gut trachea, excretion is normal and transit is not affected in males (it is a bit faster in females, but this phenotype is no longer apparent when only guts of comparable length are assessed). In normal homeostatic conditions, intestinal stem cell proliferation is also unaffected. Particularly striking are the observations we have made using two independent hypoxia reporters: 1) gut hypoxia appears to be constitutive even in the presence of trachea, and 2) gut hypoxia is not further exacerbated by the absence of gut trachea. Together, these results (which are now shown in Extended Data Figs. 9 and 10) strongly suggest that the main

role of trachea in homeostatic conditions is not to deliver oxygen to the gut but, rather, to hold gut loops together. Of note, we have further validated this idea by conducting laser ablation experiments of the gut tracheal branches *ex vivo*. This is now shown in Supplementary Movie 2.

Loss of gut trachea does, however, have functional consequences at the whole-organism level: although it does not affect food intake, we have observed that it reduces fecundity in females and also leads to reduced ability to withstand starvation in both sexes. These data are now shown in Extended Data Fig. 10d-f, i-k. A future goal will be to establish how gut trachea sustain female fecundity; given the lack of major effects on the gut itself, we feel that changes in the adjacency and communication between the ovaries and specific gut portions is an intriguing possibility in this regard. This is now discussed in the revised discussion.

Regarding the specific point the reviewer raises about the ovary asymmetry. Although we feel that it is beyond the scope of this manuscript to explore this functional asymmetry (given that the manuscript primarily concerns the intestine) we agree that it is a very interesting observation. From this perspective, we find the data in this genome-wide association study of ovariole number in a natural population of flies¹⁴ very interesting; although not specifically stated or addressed by the study, our interpretation of their quantifications of left vs right ovariole number is that it is always the same ovary that has a higher number of ovarioles regardless of the specific genetic background of the flies – suggesting that the asymmetry is somehow “hardwired”. We have mentioned this reference in the revised version of the manuscript to provide some context.

2) The authors present some numbers without any context which makes the reader wonder why this is important or unexpected. They say that most organs are more tightly packed than originally anticipated (most inter-organ distances are 5-500 micrometers apart). Why is this surprising given the size of the insect? Presumably they are even smaller in smaller insects and larger in larger insects.

We acknowledge that this statement came across as gratuitous/unfounded in the previous version of the manuscript. We had the *Drosophila* larva in mind, the organs of which have been circumstantially referred to by our colleagues as “floating in haemolymph”.

To address the reviewer’s point about adult organs more directly, we have segmented the organs of flies after starving them for 48h and quantified the distance between gut and ovary; although both gut length and ovary volume are reduced, the distance between gut and ovary remains comparable to that of well fed flies. By contrast, the distance between gut and ovary is increased in *ovoD1* female flies, in which ovary size is genetically reduced. These results (which are now shown in Extended Data Fig. 2j-o) indicate that inter-organ distances do not simply scale with organ size and could be spared/modulated to enable/constrain inter-organ signalling (see also response to next point).

Finally, we have also amended the structure of the manuscript to entirely separate facts from fiction (namely, Results from Discussion) in our revised manuscript, and have confined hypothetical considerations to the revised Discussion.

3) The authors mention that the hemolymph is viscous yet there is no comparison with other systems. Is it much more viscous than plasma in vertebrates? How far would typical hemolymph proteins diffuse over what time frames? Is it possible to address this experimentally by injecting dyes or macromolecules of specific sizes to assess whether long-range endocrine communication is possible? Otherwise there is little support for their conjecture that: "We propose that the same signal can be used to convey a different message between organs A and B and between organs C and D because D is never within reach of A."

The viscosity of the larva haemolymph is around 1.34cP, which is slightly higher than 'healthy' human blood plasma, which varies between 1.2-1.3cP^{15,16}. The viscosity of adult haemolymph has not been quantified to our knowledge, but seems higher than that of larvae in our experience.

We are beginning to address the question of whether haemolymph composition is regionalised in adult flies and our initial data is consistent with this hypothesis. Although we are still in the process of troubleshooting and optimising our methodology, analysis of haemolymph proteomics samples derived from three different locations in adult flies (head, abdomen and thorax) show that the different samples (each corresponding to haemolymph pooled from 20-30 flies) separate nicely by body part (Figure for Reviewer R3).

We do nonetheless recognise that we should have made a clearer distinction between results and these more hypothetical statements, and have confined hypothetical considerations to the revised Discussion.

4) There is little reference to research in vertebrates except at the very end. Surely physiologists must have explored these questions in mammals where the locations of organs can be experimentally manipulated. Many organs seem to function reasonably well after operations where they are moved around a lot.

Surprisingly, they have not; it is not trivial to experimentally manipulate the locations of organs without substantial damage. Whilst surgery of specific organs rarely leads to total failure of adjacent organs, there is plenty of both circumstantial and documented evidence of unexplained symptoms and "collateral damage". A case in point, and one particularly intriguing in light of our findings, is the gastrointestinal issues frequently experienced by women who have undergone hysterectomies¹⁷. Another intriguing observation in this regard is the effect of pregnancy on IBD, leading to reduced relapse rates even 3 years after pregnancy¹⁸. Finally, the now outdated use of corsets provides another example of gastrointestinal issues associated with bowel rearrangements^{19,20}.

Minor point: The authors mention "Of note, we have observed that the white genetic background commonly used as a "wild-type" stock has reduced tracheation compared to truly wild-type flies" The work of Sasaki et al (2021) PMID: 33820991 shows that intestinal stem cells in white mutants are different at least in the context of aging - maybe there are differences that could explain what the authors are seeing as well.

To clarify, whilst we have observed reduced levels of tracheation in flies with a mutation in the *white* gene (*w*¹¹¹⁸), both the sexual dimorphism in tracheal cell and number and the sex

differences in gut shape are still apparent in these mutants – this is now shown in Extended Data Fig. 11-n and clarified in the first Results Section. In our genetic manipulations, experimental flies have always been matched to at least one control with regard to the presence/absence of the *white* gene. This is explained in the last Results section.

The findings described in the paper mentioned by this reviewer are nonetheless intriguing; we wonder whether age-dependent changes in tracheation could contribute to the age-dependent, *white* gene-modulated and female-biased increase in intestinal proliferation described in this paper.

We have conducted an experiment that is consistent with this idea. We have assessed intestinal proliferation in flies specifically lacking gut trachea in normal homeostasis as well as in two hyperproliferative contexts: ageing (akin to²¹) and detergent (DSS) treatment⁶. We find that lack of gut trachea does not affect normal homeostatic intestinal stem cell proliferation and reduces DSS-induced hyperproliferation. By contrast, age-induced hyperproliferation is, in fact, further increased in flies lacking gut trachea, above the levels seen in control flies; the effect is particularly prominent in females. It will be interesting to explore how gut tracheation changes in aged flies, whether this differs between the sexes as well as possible links between tracheal changes, the function of the *white* gene and intestinal stem cell proliferation. This is now discussed in the revised manuscript.

These data are now shown in Extended Data Fig. 10a-c and their potential relevance in the context of Sasaki *et al.* is discussed in the revised Discussion of the manuscript.

For all of these reasons I feel that there is a big gap between all the beautiful data in the paper and their attempts to demonstrate a functional importance for these findings.

In summary, this is a beautiful descriptive study that will form a solid foundation for exploring the relevance of the proximity of different organs to each other in the Drosophila abdomen. It is unclear as yet as to whether these stereotypic and unexpected proximities are of functional relevance.

Our new functional data has hopefully helped close this gap. We would also like to point out that the main purpose of this manuscript was not to demonstrate a functional importance for these findings. We sought to convey two findings in the developmental biology/sex determination space, which seem very striking and should not be overlooked in our opinion.

Firstly, we describe a new, bidirectional mechanism of reciprocal sex differentiation between adjacent organs: intestinal muscles make the vascular-like system of the gut sexually dimorphic through female-biased FGF signalling. In turn, tracheal branches mechanically hold the gut loops together into a male or female shape. This contrasts with the current “binary” view of sex determination in which cells acquire their sexual fate through the intrinsic actions of their sex chromosomes and/or their exposure to circulating hormonal signals²²; there is a third way to sculpt sex differences, whereby adjacent organs control each other’s sexual identity.

Secondly, we demonstrate that there is more to insect trachea than oxygen delivery; our experiments indicate that intestinal oxygen availability and responses are unaffected by the

presence or absence of gut trachea, but these trachea structurally hold gut loops together instead, with functional consequences.

We have emphasised these considerations in our revised Discussion.

Referee #4 (Remarks to the Author):

In this study, Blackie et al. present a collaborative effort from Miguel-Aliaga and Mahadevan labs, presenting an intriguing finding about how organs maintain sexually-dimorphic shape. This paper shows how bidirectional signaling between the gut and tracheal system in Drosophila maintains the female gut shape. Based on this finding, they propose an idea of ‘vascular sex’ – that insect tracheae/ and perhaps the equivalent vasculature- encode a program that enables organs to maintain a sexually dimorphic shape, akin to a corset. Importantly, they present compelling data to test this complex theory using a combination of cutting-edge quantitative 3D-volumetric imaging and genetics. Specifically, the authors developed a 3D-volumetric scan method for imaging internal organs for large numbers of fruit flies and a quantitative image analysis pipeline; this allowed them to characterize the inter-organ adjacencies and left-right asymmetries in detail. It is to be noted that this information alone if made widely available along with tools needed for such analysis, will be a highly valuable community resource for the Drosophila physiology community. As the authors discuss, the information from such a characterization is vast and can be used to test many paradoxes in the organ specificity of such hormonal signals. For this study, the authors have focused on one important aspect- “How do sex differences in 3D-organ geometry arise?” by focusing on the sexual dimorphism of 3D-gut loops. First, they test the obvious hypothesis that these gut loops, which are adjacent to ovaries, maintain their shapes because of adjacent organs, such as ovaries. However, they found that despite genetically reducing the size of ovaries, the dimorphism of the gut was maintained, suggesting the existence of other mechanisms by which the gut maintains shape. In their quest to identify this non-cell-intrinsic sexual dimorphism, they identified a cross-talk between the gut and trachea that enables the trachea to maintain the female-specific gut shape. This reviewer finds the study to be original, thought-provoking, and of wide interest. The manuscript is a pleasure; the statistics were appropriately applied and described. But before publication, I would encourage the authors to consider revising the manuscript in the following ways to increase the functional impact of their study (see below) and provide further information that would make their 3D technology more widely accessible to other researchers.

Thank you!

1.

a. What is the functional impact of masculinizing gut shape in female flies? Have they performed any functional tests such as its effect on food transit, feeding patterns, reproductive status, and recovery of ISCs from insults such as DSS in female flies with masculinized gut shape (e.g., mex-Gal4>bnl-RNAi; Hand>bnl-RNAi)? Doing so will be important to close the loop on the idea that maintenance of gut shape by trachea has functional physiological relevance. Even if the relevance is not obvious, presenting this “negative” data would be important, as it would allow the reader to understand the importance of 3D shape in function.

b. It is possible that in a baseline physiological state, the 3D shape dimorphism of the gut has minimal impact on physiology – feeding, reproduction, recovery from DSS, etc. - but what happens if the system

is stressed? For instance, these could compare non-virgin and virgin females; flies fed an obesogenic regime versus normal diets, etc.,

We first tackled this question by focusing on flies with specific loss of gut trachea (achieved by gut muscle-specific downregulation of the FGF ligand *bnl*, *Hand>bnl-RNAi*). This genetic manipulation leads to reduced gut curvature in both males and females. It is also the genetic manipulation that more specifically affects gut trachea without affecting the gut itself (gut length is unaffected in males and only modestly affected in females), other trachea in the fly (as would be the case for the tracheal ablation, *trh^{TS}>Bax*) or, potentially, other aspects of gut muscle sexual dimorphism (as *Hand>Sxl-RNAi* or *Hand>tra-RNAi* may conceivably impact). Hence, it is the manipulation that more specifically affects gut shape.

We have assessed how this genetic manipulation affects intestinal features such as transit, excretion, intestinal stem cell proliferation and hypoxia – as one might have expected this pathway to be activated by the lack of trachea.

Interestingly, we observe no or modest effects on any of these intestinal features; in flies which lack gut trachea, excretion is normal and transit is not affected in males (it is a bit faster in females, but this phenotype is no longer apparent when only guts of comparable length are assessed). In normal homeostatic conditions, their intestinal stem cell proliferation is also unaffected. Particularly striking are the observations we have made using two independent hypoxia reporters: 1) gut hypoxia appears to be constitutive even in the presence of trachea, and 2) gut hypoxia is not further exacerbated by the absence of gut trachea. Together, these results (which are now shown in Extended Data 9 and 10) strongly suggest that the main role of trachea in homeostatic conditions is not to deliver oxygen to the gut but, rather, to hold gut loops together. Of note, we have further validated this idea by conducting laser ablation experiments (now shown in Supplementary Movie S2).

Loss of gut trachea does, however, have functional consequences at the whole-organism level: although it does not affect food intake, we have observed that it reduces fecundity in females and also leads to reduced ability to withstand starvation in both sexes. These data are now shown in Extended Data Fig. 10d-f, i-k.

Because we were particularly intrigued by the female-specific effect on fecundity, we conducted an additional experiment in which we feminised the gut muscles (by means of *Hand>Sxl-RNAi*). This too resulted in reduced female fecundity and is now shown in Extended Data Fig. 10l.

A future goal will be to establish the link(s) between gut shape and female fecundity; changes in the adjacency and communication between the ovaries and specific gut portions is an intriguing possibility in this regard. This is now discussed in the revised discussion.

2. The methods presented for MicroCT scans need to be more detailed- preferably a working protocol- to enable other researchers to use this important method more widely. Similarly, is there an open source website such as GitHub where the code for segmentation, curvature comparison, etc., can be posted for wider access by others?

Yes, this is important. We have expanded the description of our scanning protocol in the Methods so that it is easily followed and implemented. We have also deposited all the code and annotated instructions on how to run it on GitHub. This can now be accessed here: <https://github.com/Lblackie1/OrganGeometry>.

Minor comments:

1. *What is DSRF stand for? A reference for the appropriate paper and further explanation of methods for that this is a standard stain for terminal track nuclei must be elaborated.*

Apologies for this omission. DSRF stands for “*Drosophila* serum response factor”. It has been shown to be a marker of terminal tracheal cells²³ and can be used to quantify their number²⁴. We have inserted this clarification and relevant references in the revised manuscript.

2. *What stringerGFP is used for nuclei tracking? Please provide the info.*

The specific line is stated in the Methods together with references for all the other published reagents used in our study.

Referee #5 (Remarks to the Author):

In this interesting manuscript, Blackie et al. use micro-computed tomography to demonstrate quantitatively the 3D morphology and spatial arrangement of the internal organs of the adult fruit fly, with a special focus on the intestine, which spans the anterior-posterior axis of the animal body. They use volumetric scans in combination with computer graphics' methods to segment internal organs, visualize previously unnoticed left-right asymmetries, identify sex differences in organ shape and quantify organ-organ proximities. Specifically, by imaging large number of flies, they show that the male and female adult intestine differ in shape especially at the central loop region, with the female gut being longer and more curved. Then, they go ahead to explain gut shape sex differences by assessing the role of the vasculature-like trachea that contacts the intestine throughout its length. They show that the intestinal tracheal branches mechanically hold the gut loops together and contribute to gut shape in a sex-specific manner. They propose that, in response to the sex determination pathway, the FGF ligand Branchless (Bnl) produced by gut cells at different levels/expression patterns in males and females is key to determining male vs female gut shape. The experiments are clearly presented and, for the most part, are well-executed. The work is timely and will be of interest to broad readership, since it describes quantitatively and in detail the 3D arrangement and spatial proximities of an adult animal's internal organs, which might explain organ communication in space.

Please see below specific points that the authors should address before publication.

1. *The tomographic work is elegant and nicely shows the adult fly spatial internal organ arrangement. The authors use wild-type OregonR flies to perform their analyses. To exclude the possibility that organ shape is affected by the genetic background, did they also observe the same spatial organ relationships in other wild-type genotypes?*

We have extracted gut centrelines and conducted geometric morphometrics analysis of gut shape of two additional genetic backgrounds: Canton S (CS) and flies harbouring a *white* (*w¹¹¹⁸*) mutation. In both cases, we observe a comparable gut looping pattern and a sexual dimorphism in gut shape. This is now shown in Extended Data Fig. 1i-n.

2. Figure 1a-b: although the tomography images are clear, for the non-expert it would be useful to add the V (ventral) and D (dorsal) annotation in the figure or legend.

Yes, this is true. We have now inserted these.

3. Figure 1c: the authors should correct the regional annotations in the drawing. What they annotate as “anterior midgut” corresponds to R1 and R1 is never found in the gut loops according to published work on regionalization.

Yes, apologies for this oversight. This has now been rectified.

4. Page 3: “...most inter-organ distances are in the 5-500 μm range (Fig. 1a,b, Extended Data Table 1).” There is no information in Fig. 1a, b on interorgan distances. The authors should refer to Fig. 1k-m here and relevant information in Ext. Data Fig. 1g-h, l-o.

This has now been corrected too.

5. Page 3: The last sentence of the second paragraph is unclear. “... they are meant to elicit? We propose that the same signal can be used to convey a different message between organs A and B and between organs C and D because D is never within reach of A.” Maybe the authors mean: We propose that the same signal can be used to convey a different message between organs A and B and between organs A and C because C is never within reach of A?

Either option seems correct, but the reviewer’s suggestion does make it less confusing, so we have amended the wording accordingly.

6. Page 3 and Ext. Data Fig. 2: the ovoD1 females have shorter guts/gut loops. Is their overall body size smaller than controls? Also, is the intestinal tracheation of ovoD1 females of the same extent as in the controls? It would be interesting to show the internal organ-organ proximities of the ovoD1 fly in comparison to the control.

We have quantified the abdominal volume of *ovoD1* females in our microCT scan and it is comparable to that of control flies. This is now shown in Figure for Reviewer R4.

We have analysed their intestinal tracheation; interestingly, we have observed reduced numbers of intestinal trachea in *ovoD1* mutant flies (Figure for Reviewer R5). At this point, we would prefer not include these data in the manuscript as we do not know the reason for this effect: ecdysone, other ovary-derived signals, differences in feeding and/or nutritional effects could all play a role.

We have also segmented the organs of *ovoD1* females and quantified the distance between gut and ovary; the distance between gut and (the now reduced) ovaries is increased.

Interestingly, organ shrinkage does not invariably result in increased inter-organ distances; in starved flies, ovary volume is also reduced, but the distance between the gut and ovary is preserved. These data indicate that inter-organ distances can be spared or modulated depending on context, pointing to active regulation of this process (potentially to modulate haemolymph dynamics). Both experiments are now shown in Extended Data Fig. 2j-o.

7. The authors use a *bnl*-LexA reporter to show the *bnl* expression pattern in the midgut. They also generated a *bnl*-Gal4 reporter from the original LexA line, which does not lack any exons (does the latter express full length *bnl*? I did not see the data). They show that this reporter is expressed in the gut enterocytes (ECs) in larvae, and then in adults, as well as in the gut VM transiently in young adults (5 hrs APE); thus, they focus their silencing experiments in the ECs and the VM. Since the Bnl ligand is known to be very dynamic during development, it is possible that the reporters do not recapitulate the full expression pattern of *bnl*. In addition, it has been shown (Perochon et al, 2021; Tamamouna et al, 2021) that tracheal *bnl* is involved in visceral tracheal branching in adults in response to damage and tumors. Have the authors tested other potential *bnl* sources, e.g. the trachea, that may affect gut shape apart from the ECs?

Our trojan *bnl*-Gal4 reporter does not express full length *bnl*: it is a T2A fusion²⁵, which is truncated after the 1st *bnl* exon during translation (see schematic in Extended Data Fig. 4a). Consistent with lack of expression of a functional Bnl, these flies are homozygous lethal, just as the *bnl*-lexA flies are. To clarify, the difference with *bnl*-lexA is that our construct does not eliminate any endogenous genomic regions and the inserted T2A-Gal4 is under the control of the endogenous *bnl* promoters/enhancers. By contrast, the *bnl*-lexA reporter represents a deletion²⁶ (which could conceivably have eliminated cis-regulatory elements of *bnl*, and LexA expression is under the control of its own ectopic promoter. This is now more clearly explained in the Methods.

The reviewer is correct in stating that the above studies showed tracheal *bnl* expression in regeneration, not in homeostatic conditions.

To further validate that the relevant source of *bnl* for tracheal survival and branching in normal homeostatic conditions is the gut muscle, we have conducted the following experiments:

We have downregulated *bnl* from gut muscles using an independent visceral muscle driver: *vm-Gal4* (*GMR13B09-Gal4*, a putative hairy enhancer *Gal4* previously used as a gut visceral muscle driver²⁷). Like the *Hand-Gal4*-driven downregulation, this manipulation reduces tracheal numbers and affects gut shape in both males and females (the effect is less strong than that of *Hand-Gal4*, likely because of weaker expression, see our response to the next point). The expression pattern of both drivers is now shown in Extended Data Fig. 4e, f and the effects of *vm>bnl-RNAi* on tracheal cell numbers and gut shape are now shown in Extended Data Fig. 3g-k.

We have depleted Bnl expression specifically in intestinal enterocytes (the only other intestinal cell type which we find to express Bnl in normal homeostatic conditions; downregulation was achieved by means of *mex>bnl-RNAi*). This does not affect tracheal cell numbers, branching or gut shape. This is now shown in Extended Data Fig. 5a,b,e,f.

Finally, we have downregulated *bnl* from our *bnl-Gal4* line, which is not expressed in tracheal cells in normal, homeostatic conditions (Extended Data Fig. 4a-d). Like the muscle-specific manipulations, this results in loss gut trachea, further validating the *bnl-Gal4* driver as a tool (Figure for Reviewer R6).

Therefore, given that all (and only those) manipulations that include the muscle Bnl pool lead to a complete loss of gut trachea, we believe that the muscle is the key source of Bnl for tracheal survival and branching during pupation. This of course does not exclude roles for other tissues in other developmental/regenerative contexts such as those reported by^{24,28}. We have made a clearer distinction between tracheal roles in homeostasis vs regeneration in the revised manuscript, both in the relevant Results sections and the revised Discussion.

8. Figure 2: The authors use Hand-Gal4 to silence bnl (and later tra) specifically in the VM, but they do not actually show the specificity of expression of this driver. Maybe I have missed it, but I could not find any information about it either in the manuscript or in Flybase. If this is true, the authors should show the detailed characterization of the Hand-Gal4 driver (or provide a reference for its expression pattern).

Apologies for this oversight. We now provide a relevant reference. Although *Hand* expression in visceral muscles has been previously reported²⁹⁻³¹, we do describe expression of this *Hand-Gal4* trojan line in the midgut for the first time. We now provide detailed characterisation of the expression of both this *Hand-Gal4* and *vm-Gal4* (Extended Data Fig. 4e, f). Although *bnl* downregulation from both lines leads to reduced tracheation and changes in gut shape, we favoured the use of *Hand-Gal4* over *vm-Gal4* because, as the figure shows, *Hand-Gal4* is more strongly and specifically expressed in gut muscles.

9. Does VM-specific bnl silencing affect gut length?

Neither *Hand>bnl-RNAi* nor *vm>bnl-RNAi* affect gut length in males. In females, there is no effect of *vm>bnl-RNAi* on gut length, but *Hand>bnl-RNAi* does have a modest effect on gut length. Importantly, the effect on gut shape is apparent in both males and females, and the effect on shape in females is apparent even with *vm>bnl-RNAi* which does not affect gut length. The length data is shown in Extended Data Fig. 3e,k.

Also, expression of the bnlRNAi under Hand-Gal4 is induced early during development (without Gal80ts), so the lack of tracheal cells can be interpreted as a developmental phenotype. What happens to the tracheal cells? Are they not specified or do they die after they are born?

A clarification is needed here. Although *Hand-Gal4* is expressed throughout development, no *bnl* expression is detected in visceral muscles until the pupal stage (see Extended Data Fig. 4a-d); hence, *Hand>bnl-RNAi* targets this pupal peak of muscle *bnl* expression, which subsides in early adulthood.

We have nonetheless confined *bnl* downregulation in gut muscle to the pupal stage by means of *Gal80ts*-regulated, *Hand-Gal4*-driven *bnl-RNAi*. This results in loss of trachea comparable

to that seen upon constitutive depletion of *bnl* from the gut muscle. This is now shown in Extended Data Fig. 3f.

In addition, since VM-specific bnl silencing is a very important experiment for their study, the authors should use alternate VM-specific drivers (e.g., how-Gal4, which is also expressed in the trachea in larvae Aghajanian et al, Dev. Biol. 2016) to knock down bnl expression and assess gut shape etc, and silencing should be ideally restricted to adult stages. Alternatively, have the authors tried to silence bnl using the bnl-Gal4 they generated in this study? This will be informative too.

We believe the *bnl>bnl-RNAi*, *vm>bnl-RNAi* and the temporally controlled *Hand^{TS}>bnl-RNAi* experiments described above have addressed this point.

Finally, is VM-specific overexpression of bnl sufficient to change tracheation and the shape/looping of the gut?

We had, in fact, planned to do this experiment. The difficulty we encountered is that ectopic expression of *bnl* leads to enlarged or malformed guts, which would confound gut shape analysis. Indeed, strong *bnl* over-expression using *UAS-bnl^l* leads to dramatic increase in the number of terminal tracheal cells (as assessed with DSRF expression), far above the numbers seen in wild-type females (see Figure for Reviewer 1). We also attempted milder *bnl* over-expression using *UAS-bnISAM* (CRISPR-based overexpression from the endogenous *bnl* promoter²) but this also increased gut diameter. Although these manipulations do affect gut shape in both males and females (see Figure for Reviewer 1), we are reluctant to use these *bnl* expression tools to conduct rescue experiments because of their ectopic effects on the gut itself, which we have not observed in other manipulations.

We have, however, conducted a more physiological gain-of-function experiment by feminising the male gut muscle by means of *Hand-Gal4*-driven *UAS-Sxl* expression. This leads to increased Bnl levels in the gut muscles of males, which become comparable to those of females, without the effects on gut diameter observed upon *bnl* over-expression using *UAS-bnISAM*. As anticipated, this genetic manipulation does not affect Bnl expression in females. Pleasingly, this manipulation increases gut tracheation in males to female-like levels and affects gut shape in males, but not females. This is now shown in Fig. 4g-i and Extended Data Fig. 8j,k.

10. Figure 2h: it is not clear from the legend or the methods whether the number of DSRF-positive cells and tracheal length are normalized to the gut surface; they should be.

11. Figure 3c-d; Figure 4c: please see comment 10.

We do not believe it is the right normalisation to consider in the context of the biology that we are considering; we are investigating how the sex of one organ (gut) impacts another (trachea) and vice versa. It is therefore important to have an absolute quantification of the degree of branching in each case rather than a relative “coverage” value, which does not distinguish between effects on tracheal branching vs effects on the gut itself. As a hypothetical example, if we were changing the sex of the trachea and that reduced tracheal

branching, but this, in turn, reduced gut size, normalisation might mask both effects and give the same “coverage” value.

Nevertheless, to reassure the reviewer and any readers that may share their concerns, we have conducted this analysis for the wild-type guts in Figure 3 and have confirmed that tracheal cell branching remains higher in females even when normalised by gut area. This is now shown in Figure 3d.

*12. The authors observe sex differences in tracheal coverage and *bnl* abundance and since they cannot masculinize the female tracheal cells (Ext. Data Fig. 7a-h), they decide to masculinize ligand expression by silencing *tra* in the female VM (Fig. 4 and Ext. Data Fig. 7).*

*The *tra*RNAi is a key experiment and should be validated with additional *tra*RNAi lines, as well as with other genes involved in the sex determination pathway. For example, does silencing of *Sxl* or *tra2* affect gut *bnl* expression?*

Did the authors check whether the genes involved in the sex determination pathway are expressed in the tracheal cells and the VM (as the same lab had done in Hurdy et al, 2016)?

Firstly, it is important to clarify that both tracheal and gut muscle cells have an intrinsic sexual identity which can be genetically altered. Indeed, we have generated an endogenously tagged TraF (there are currently no working TraF antibodies) and have confirmed they both express the sex determinant TraF only in females but not in males – see Figure for Reviewer R2. We would rather not include these data in this manuscript because our former postdoc Bruno Hudry, who generated this line, is hoping to publish it in a manuscript he is currently putting together.

The “failed” masculinisation of trachea was not failed in the sense that we are able to downregulate TraF expression in females using *tra-RNAi*; effectiveness of the transgene is, in fact, demonstrated in this study in muscle cells. Indeed, we masculinise muscle features (e.g. *bnl* expression) by expressing the same *tra-RNAi* transgene from a muscle rather than tracheal driver (see Extended Data Fig. 8a-g). This transgene also masculinises other intestinal cell types as we have done and published several times before^{9,10}). The finding (which is in our opinion an important one in the sex determination field) is that, in this particular case, the sexual dimorphism in tracheal cell branching and number is not intrinsically controlled from the tracheal cells themselves by the intrinsic sex determination pathway (namely, TraF and its upstream regulator *Sxl* in females, see¹¹ for a review) but, rather, extrinsically by the levels of the gut muscle-derived FGF ligand *Bnl*.

But we acknowledge that the *tra-RNAi* experiment should be independently validated and have conducted the following experiments:

1. We have repeated the *tra* downregulation in tracheal cells experiment using an independent *tra-RNAi* line, with comparable results (namely, lack of effects on terminal tracheal cell number and branching). This is now shown in Extended Data Fig. 7.

2. We have conducted an additional and independent masculinisation experiment of gut muscles by downregulating the upstream intrinsic female sex determinant *Sxl* from gut muscles (by means of *Hand-Gal4*-driven *UAS-Sxl-RNAi* expression). Consistent with the previous *tra-RNAi* masculinisation experiment, we have observed masculinised *Bnl* expression in the gut muscles of females, reduced tracheal cell number, branching and gut curvature in females only. These data are now shown in Fig. 4a-f and Extended Data Fig. 8h,i.

3. As explained in response to a previous point, we have complemented the gut muscle *bnl* RNAi masculinisation experiment by conducting a feminisation experiment. To this end, we have expressed *UAS-Sxl* in the gut muscles (males do not normally express this female sex determinant) and have observed that this feminises (increases) *Bnl* expression levels in the gut muscles of males, and increases the number and branching of terminal tracheal cells only in males, to levels comparable to those in females. As expected, this genetic manipulation has no effects in females. Importantly, and opposite to the previous masculinisation experiment, this feminisation affects gut shape in males, but not in females. This is now shown in Fig. 4g-l and Extended Data Fig. 8j,k.

It seems that although Hand-Gal4>traRNAi-mediated female bnl masculinization reduces gut tracheal coverage and gut length (note that the tracheal features in Fig. 7c should be normalized to gut surface), the gut shape and curvature pattern does not change much as shown in Fig. 4d-f.

Regarding gut length. The sex differences in shape are apparent even when controlling for gut length (as revealed by our co-variation analysis of how gut shape co-varies with gut length or the volume of other organs, Supplementary Table 17-19). Some of the genetic manipulations that affect gut shape affect gut length. This is, however, not the case for all of them (see for example, *vm(13BO9)>bnl-RNAi*). We have conducted an additional experiment in which we starve flies for 48h, leading to gut shortening. Sex differences remain in this condition. This is now shown in Extended Data Fig. 1s-v, Supplementary Tables 12,13. Hence, the guts of males and females differ in shape irrespective of length, and trachea control gut shape even in situations that do not affect gut length.

The effect of tracheal masculinisation is indeed more subtle than that of, for example, complete tracheal ablation. While it is indeed difficult to visualise in the *Hand>traRNAi* masculinisation in Extended Data Fig. 8e (it is only possible to see a narrowing of the central loop inflexion), there is a shift and statistically significant difference in the MDS analysis shown in Extended Data Fig. 8g. The main effect is not on overall gut shape, but rather on localised effects in the central midgut loop region (as seen from changes in curvature occurring mostly in the central midgut region of females, rather than throughout the midgut as seen in Extended Data Fig. 8f). It is possible that Extended Data Fig. 8d does not reflect this well given that large changes in a small number of landmarks can be spread around the other landmarks during Procrustes superimposition; this is known as the 'Pinocchio effect'³².

Both our additional masculinisations (using *Sxl-RNAi* and an additional *tra-RNAi*) and the *UAS-Sxl* masculinisation have hopefully reassured the reviewer that the effect is real. Importantly, significant effects of masculinisations are observed in females only and *vice versa* for feminisations, further confirming the specificity of the phenotypes.

Finally, can overexpression of traF in males feminize bnl expression, tracheation and gut shape?

We attempted *traF*-mediated feminisations but we found the *UAS-traF* transgene to be a bit toxic when expressed in gut muscle, in line with unexpected effects previously reported to result from excessive TraF levels (https://bdsc.indiana.edu/stocks/misc/cline_notes.html). Instead, we have conducted the feminisation experiments using *UAS-Sxl* we have described in response to previous points, which have successfully feminised Bnl expression levels in muscles, tracheal number and branching and gut shape. This is now shown in Fig. 4g-l and Extended Data Fig. 8j,k.

13. Ext. Data Fig. 4b: in the single channel bnl-Gal4>Stinger-GFP images, the same guts should be shown in the top and bottom panels to indicate ECs and VM in the same gut region.

The purpose of Extended Data Fig. 4b (now Extended Data 4c) was to show that, at the time when *bnl* expression in gut visceral muscles is female-biased, epithelial expression is not; different gut regions were picked based on where expression is apparent at that stage, and a lower magnification was chosen so that differences in expression levels could be better appreciated. The enterocyte vs visceral muscle comparison that the reviewer requests is provided in Extended Data Fig. 4d, which does show expression in the same gut region.

14. Ext. Data Fig. 5, Fig.6 and Fig. 7: all tracheal features measured should be normalized to the gut surface.

Please see our response to the previous point concerning this.

15. All figures (main and supplementary) need titles.

Titles are now included, thanks.

16. The introduction is missing. Although there might be a length limit, the authors should at least introduce their study in one paragraph.

This has now been rectified.

17. Page 4, top paragraph: "intriguingly" is used in 2 consecutive sentences.

This has now been rectified.

18. Figure 3 is described in the text (Page 4, second paragraph) before the second half of Figure 2. All figures should be described in the order they are presented.

In the revised version, we have generally followed the order in which they are presented, but we also felt it was important to group figure panels thematically which may occasionally result in one sentence referring to a panel in a previous figure.

References

- 1 Sutherland, D., Samakovlis, C. & Krasnow, M. A. branchless encodes a Drosophila FGF homolog that controls tracheal cell migration and the pattern of branching. *Cell* **87**, 1091-1101 (1996).
[https://doi.org:10.1016/s0092-8674\(00\)81803-6](https://doi.org:10.1016/s0092-8674(00)81803-6)
- 2 Jia, Y. *et al.* Next-generation CRISPR/Cas9 transcriptional activation in Drosophila using flySAM. *Proc Natl Acad Sci U S A* **115**, 4719-4724 (2018). <https://doi.org:10.1073/pnas.1800677115>
- 3 Biteau, B., Hochmuth, C. E. & Jasper, H. JNK activity in somatic stem cells causes loss of tissue homeostasis in the aging Drosophila gut. *Cell Stem Cell* **3**, 442-455 (2008).
<https://doi.org:10.1016/j.stem.2008.07.024>
- 4 Choi, N. H., Kim, J. G., Yang, D. J., Kim, Y. S. & Yoo, M. A. Age-related changes in Drosophila midgut are associated with PVF2, a PDGF/VEGF-like growth factor. *Aging Cell* **7**, 318-334 (2008).
<https://doi.org:10.1111/j.1474-9726.2008.00380.x>
- 5 Rera, M., Clark, R. I. & Walker, D. W. Intestinal barrier dysfunction links metabolic and inflammatory markers of aging to death in Drosophila. *Proc Natl Acad Sci U S A* **109**, 21528-21533 (2012).
<https://doi.org:10.1073/pnas.1215849110>
- 6 Amcheslavsky, A., Jiang, J. & Ip, Y. T. Tissue damage-induced intestinal stem cell division in Drosophila. *Cell Stem Cell* **4**, 49-61 (2009). <https://doi.org:10.1016/j.stem.2008.10.016>
- 7 Ahmed, S. M. H. *et al.* Fitness trade-offs incurred by ovary-to-gut steroid signalling in Drosophila. *Nature* **584**, 415-419 (2020). <https://doi.org:10.1038/s41586-020-2462-y>
- 8 Zipper, L., Jassmann, D., Burgmer, S., Gorlich, B. & Reiff, T. Ecdysone steroid hormone remote controls intestinal stem cell fate decisions via the PPARgamma-homolog Eip75B in Drosophila. *Elife* **9** (2020).
<https://doi.org:10.7554/eLife.55795>
- 9 Hudry, B. *et al.* Sex Differences in Intestinal Carbohydrate Metabolism Promote Food Intake and Sperm Maturation. *Cell* **178**, 901-918 e916 (2019). <https://doi.org:10.1016/j.cell.2019.07.029>
- 10 Hudry, B., Khadayate, S. & Miguel-Aliaga, I. The sexual identity of adult intestinal stem cells controls organ size and plasticity. *Nature* **530**, 344-348 (2016). <https://doi.org:10.1038/nature16953>
- 11 Saccone, G. A history of the genetic and molecular identification of genes and their functions controlling insect sex determination. *Insect Biochem Mol Biol* **151**, 103873 (2022).
<https://doi.org:10.1016/j.ibmb.2022.103873>
- 12 Gaspar, P. *et al.* Characterization of the Genetic Architecture Underlying Eye Size Variation Within Drosophila melanogaster and Drosophila simulans. *G3 (Bethesda)* **10**, 1005-1018 (2020).
<https://doi.org:10.1534/g3.119.400877>
- 13 Klingenberg, C. P. Visualizations in geometric morphometrics: how to read and how to make graphs showing shape changes. *Hystrix It. J. Mamm.* **24**, 15-24 (2013).
- 14 Lobell, A. S., Kaspari, R. R., Serrano Negron, Y. L. & Harbison, S. T. The Genetic Architecture of Ovariole Number in Drosophila melanogaster: Genes with Major, Quantitative, and Pleiotropic Effects. *G3 (Bethesda)* **7**, 2391-2403 (2017). <https://doi.org:10.1534/g3.117.042390>
- 15 Nader, E. *et al.* Blood Rheology: Key Parameters, Impact on Blood Flow, Role in Sick Cell Disease and Effects of Exercise. *Front Physiol* **10**, 1329 (2019). <https://doi.org:10.3389/fphys.2019.01329>
- 16 Zabihisari, A., Parand, S. & Rezai, P. PDMS-based microfluidic capillary pressure-driven viscometry and application to hemolymph. *Microfluid Nanofluid* **27** (2023). <https://doi.org:ARTN810.1007/s10404-022-02617-0>
- 17 Altman, D. *et al.* Effect of hysterectomy on bowel function. *Dis Colon Rectum* **47**, 502-508; discussion 508-509 (2004). <https://doi.org:10.1007/s10350-003-0087-5>
- 18 Beaulieu, D. B. & Kane, S. Inflammatory Bowel Disease in Pregnancy. *Gastroenterol Clin N* **40**, 399-+ (2011). <https://doi.org:10.1016/j.gtc.2011.03.006>
- 19 Gau, C. R. *Historic Medical Perspectives of Corseting and Two Physiologic Studies with Reenactors.* (UMI, 2009).
- 20 Lane, W. A. Civilisation in relation to the abdominal viscera, with remarks on the corset. *The Lancet* **174**, 1416-1418 (1909). [https://doi.org:https://doi.org/10.1016/S0140-6736\(01\)11699-5](https://doi.org:https://doi.org/10.1016/S0140-6736(01)11699-5)
- 21 Sasaki, A., Nishimura, T., Takano, T., Naito, S. & Yoo, S. K. white regulates proliferative homeostasis of intestinal stem cells during ageing in Drosophila. *Nat Metab* **3**, 546-557 (2021).
<https://doi.org:10.1038/s42255-021-00375-x>

- 22 Arnold, A. P. Integrating Sex Chromosome and Endocrine Theories to Improve Teaching of Sexual Differentiation. *Cold Spring Harb Perspect Biol* **14** (2022). <https://doi.org:10.1101/cshperspect.a039057>
- 23 Affolter, M. *et al.* The Drosophila Srf Homolog Is Expressed in a Subset of Tracheal Cells and Maps within a Genomic Region Required for Tracheal Development. *Development* **120**, 743-753 (1994).
- 24 Perochon, J. *et al.* Dynamic adult tracheal plasticity drives stem cell adaptation to changes in intestinal homeostasis in Drosophila. *Nat Cell Biol* **23**, 485-496 (2021). <https://doi.org:10.1038/s41556-021-00676-z>
- 25 Diao, F. *et al.* Plug-and-play genetic access to drosophila cell types using exchangeable exon cassettes. *Cell Rep* **10**, 1410-1421 (2015). <https://doi.org:10.1016/j.celrep.2015.01.059>
- 26 Du, L. *et al.* Unique patterns of organization and migration of FGF-expressing cells during Drosophila morphogenesis. *Dev Biol* **427**, 35-48 (2017). <https://doi.org:10.1016/j.ydbio.2017.05.009>
- 27 Guo, Z., Driver, I. & Ohlstein, B. Injury-induced BMP signaling negatively regulates Drosophila midgut homeostasis. *J Cell Biol* **201**, 945-961 (2013). <https://doi.org:10.1083/jcb.201302049>
- 28 Tamamouna, V. *et al.* Remodelling of oxygen-transporting tracheoles drives intestinal regeneration and tumorigenesis in Drosophila. *Nat Cell Biol* **23**, 497-510 (2021). <https://doi.org:10.1038/s41556-021-00674-1>
- 29 Han, Z., Yi, P., Li, X. & Olson, E. N. Hand, an evolutionarily conserved bHLH transcription factor required for Drosophila cardiogenesis and hematopoiesis. *Development* **133**, 1175-1182 (2006). <https://doi.org:10.1242/dev.02285>
- 30 Kolsch, V. & Paululat, A. The highly conserved cardiogenic bHLH factor Hand is specifically expressed in circular visceral muscle progenitor cells and in all cell types of the dorsal vessel during Drosophila embryogenesis. *Dev Genes Evol* **212**, 473-485 (2002). <https://doi.org:10.1007/s00427-002-0268-6>
- 31 Lo, P. C., Zaffran, S., Senatore, S. & Frasch, M. The Drosophila Hand gene is required for remodeling of the developing adult heart and midgut during metamorphosis. *Dev Biol* **311**, 287-296 (2007). <https://doi.org:10.1016/j.ydbio.2007.08.024>
- 32 Klingenberg, C. P. How Exactly Did the Nose Get That Long? A Critical Rethinking of the Pinocchio Effect and How Shape Changes Relate to Landmarks. *Evol Biol* **48**, 115-127 (2021).

Figure for Reviewers

Figure R1

Figure R2

Figure R3

Red: abdomen, green: head, blue: thorax (all samples from virgin males)

Figure R4

Figure R5

Figure R6

Reviewer Reports on the First Revision:

Referees' comments:

Referee #1 (Remarks to the Author):

I congratulate the authors for the extensive revision work they have undertaken. I believe that the additional work and discussions presented in the revised manuscript significantly strengthen the study. Therefore, I wholeheartedly stand by my very positive initial assessment of this study, which is a remarkable achievement of the highest quality. Its innovative methods, insightful findings, and groundbreaking conclusions elevate it to a milestone in the scientific community. The implications of this research are profound, and it will undoubtedly inspire and drive further investigations in this exciting area of study. I recommend the publication of the manuscript in its current form without hesitation.

Referee #2 (Remarks to the Author):

Review

I have now read both the new version of the manuscript and the reply to reviewers. The authors have performed an extensive amount of work to address reviewer comments, and the manuscript is much improved. In my previous review, I had two main criticisms:

1) "this study shows that the gut is sexually dimorphic, the tracheal is sexually dimorphic, and their shapes are related to one another regardless of sex. They have not shown that the codependency of the trachea and gut is sexually dimorphic."

- Here the authors have clarified their previous experiments and have added new experiments.

2) "While the authors present possible roles for the organ dimorphism, there is no test for a possible role."

- Here the authors have performed several interesting experiments, many presented in Extended Figure 10.

For both of these points, the authors have provided some evidence to support their conclusions. Although, I still maintain my previous concerns that there is little evidence that the co-dependance is functionally sexually dimorphic. The best support for this is found in extended figure 10b and c, which I highly suggest adding to the main figures as these data would cap off the paper nicely. In fact, the authors reference these data to address 4 of the 5 reviewers, yet they are buried in the supplement.

Given the massive amounts of detailed data, I do not think it is reasonable to ask the authors to perform any additional experiments and I strongly recommend publishing the work in its current form.

Referee #3 (Remarks to the Author):

The revised manuscript is much improved. The authors have made a clear separation between the data and the conclusions directly supported by the data from their more speculative inferences. I think this manuscript would be of general interest and appeal to the readers of Nature,

While the fact that there is sexual dimorphism in the intestine is interesting, I feel that the most significant finding of this study is the back-and-forth signaling between the intestine and the tracheal system which seems to provide the structural tether for the intestine. The pattern of signaling from the intestine to the tracheal system differs between the sexes which results in differences between the tracheal system of males and females. These differences in the "tether" allow for conformational differences in the intestine. As the authors point out, tethering systems (e.g. the vasculature, the mesentery) could play a role in directing the conformation and growth of soft organs and these could account not just for sexual dimorphism but also differences between related species. I feel that authors could emphasize this aspect more than the sexually dimorphic nature of the intestine in the title and in the abstract.

I found one sentence a little confusing: "Indeed, it differentially impacts two hyperproliferative responses: it abrogates the normal difference between sucrose- and damage (DSS)-induced proliferation, whereas it increases age-induced hyperproliferation in the intestinal epithelium(Extended Data Fig. 10b,c)."

This sentence seems to imply that both sucrose and DSS induce hyperproliferation and that this difference is abrogated by the absence of gut trachea. I suspect that the authors are using sucrose as a baseline for proliferation and comparing that to the hyperproliferation caused by DSS. Is that correct? Maybe this can be explained better by breaking it up into two sentences - one about DSS vs sucrose and the other about age-induced hyperproliferation.

Referee #4 (Remarks to the Author):

In this revised version of the manuscript, the authors have made a herculean effort to address the comments from 5 peer-reviewers systematically. In doing so, they have not only closed the key gaps but in my view have added two further intriguing and important findings, to their already stunning work. First, they show that the trachea's primary role is not to deliver oxygen to the gut but, rather, to hold gut loops together. Second, they show that instead of impacting food transit or ingestion, the sexually dimorphic role of the trachea in gut geometry supports the organism's fecundity and starvation resilience. In addition, the authors have made available their code for analyzing the morphological data in GitHub and providing details on how to run their code. Overall, this work adds a new dimension to how organs maintain sexually dimorphic shape. I congratulate the authors on the beautiful study and highly recommend the manuscript for publication in Nature.

Akhila Rajan

Referee #5 (Remarks to the Author):

The authors have revised the manuscript substantially and have answered to all my concerns.
Beautiful work!

Author Rebuttals to First Revision:

Referee #1

I congratulate the authors for the extensive revision work they have undertaken. I believe that the additional work and discussions presented in the revised manuscript significantly strengthen the study. Therefore, I wholeheartedly stand by my very positive initial assessment of this study, which is a remarkable achievement of the highest quality. Its innovative methods, insightful findings, and groundbreaking conclusions elevate it to a milestone in the scientific community. The implications of this research are profound, and it will undoubtedly inspire and drive further investigations in this exciting area of study. I recommend the publication of the manuscript in its current form without hesitation.

No response needed.

Referee #2

Review

I have now read both the new version of the manuscript and the reply to reviewers. The authors have performed an extensive amount of work to address reviewer comments, and the manuscript is much improved. In my previous review, I had two main criticisms:

1) *“this study shows that the gut is sexually dimorphic, the tracheal is sexually dimorphic, and their shapes are related to one another regardless of sex. They have not shown that the codependency of the trachea and gut is sexually dimorphic.”*

- Here the authors have clarified their previous experiments and have added new experiments.

2) *“While the authors present possible roles for the organ dimorphism, there is no test for a possible role.”*

- Here the authors have performed several interesting experiments, many presented in Extended Figure 10.

For both of these points, the authors have provided some evidence to support their conclusions. Although, I still maintain my previous concerns that there is little evidence that the co-dependence is functionally sexually dimorphic. The best support for this is found in extended figure 10b and c, which I highly suggest adding to the main figures as these data would cap off the paper nicely. In fact, the authors reference these data to address 4 of the 5 reviewers, yet they are buried in the supplement.

As explained in our previous response, we agree that the anatomical/signalling co-dependence does not differ between the sexes in the sense that, in both males and females, a muscle-derived ligand (Bnl) promotes tracheal branching; tracheal branches also “stitch” gut loops together in both sexes. What differs between the sexes is the levels of the muscle-derived ligand and, consequently, the degree of tracheal branching and gut stitching – less Bnl made by gut muscles in males means less tracheal branching which, in turn, results in a male-like gut shape.

This sex-shared co-dependence but sex-biased signalling does result in a sex-biased functional co-dependence; as the reviewer states, tracheal ablation and gut muscle masculinisation both result in effects that differ somewhat between the sexes.

As suggested by the reviewer, we have created an additional main figure (Fig. 5) with the relevant functional data including former panels 10b and c.

Perhaps part of the misunderstanding regarding this co-dependence stems from our (mis)use of “sexually dimorphic”: the signal is not sexually dimorphic - just sex-biased in its levels. For this reason, we have also modified some of our wording around sex differences.

Given the massive amounts of detailed data, I do not think it is reasonable to ask the authors to perform any additional experiments and I strongly recommend publishing the work in its current form.

Referee #3

The revised manuscript is much improved. The authors have made a clear separation between the data and the conclusions directly supported by the data from their more speculative inferences. I think this manuscript would be of general interest and appeal to the readers of Nature,

While the fact that there is sexual dimorphism in the intestine is interesting, I feel that the most significant finding of this study is the back-and-forth signaling between the intestine and the tracheal system which seems to provide the structural tether for the intestine. The pattern of signaling from the intestine to the tracheal system differs between the sexes which results in differences between the tracheal system of males and females. These differences in the "tether" allow for conformational differences in the intestine. As the authors point out, tethering systems (e.g. the vasculature, the mesentery) could play a role in directing the conformation and growth of soft organs and these could account not just for sexual dimorphism but also differences between related species. I feel that authors could emphasize this aspect more than the sexually dimorphic nature of the intestine in the title and in the abstract.

This is a good point. We have mentioned these points in the discussion and have revised the abstract accordingly.

I found one sentence a little confusing: "Indeed, it differentially impacts two hyperproliferative responses: it abrogates the normal difference between sucrose- and damage (DSS)-induced proliferation, whereas it increases age-induced hyperproliferation in the intestinal epithelium(Extended Data Fig. 10b,c)."

This sentence seems to imply that both sucrose and DSS induce hyperproliferation and that this difference is abrogated by the absence of gut trachea. I suspect that the authors are using sucrose as a baseline for proliferation and comparing that to the hyperproliferation caused by DSS. Is that correct? Maybe this can be explained better by breaking it up into two sentences - one about DSS vs sucrose and the other about age-induced hyperproliferation.

This paragraph has now been amended as suggested. The paragraph now reads as follows: Absence of gut trachea does, however, impact whole-body physiology, particularly in females and in response to challenges. Indeed, it differentially impacts two hyperproliferative responses: on the one hand, it increases age-induced hyperproliferation in the intestinal epithelium. By contrast, it reduces damage-induced intestinal proliferation in flies fed a detergent (DSS)/sucrose mixture relative to the baseline intestinal proliferation observed in sucrose only-fed flies.

Referee #4

In this revised version of the manuscript, the authors have made a herculean effort to address the comments from 5 peer-reviewers systematically. In doing so, they have not only closed the key gaps but in my view have added two further intriguing and important findings, to their already stunning work. First, they show that the trachea's primary role is not to deliver oxygen to the gut but, rather, to hold gut loops together. Second, they show that instead of impacting food transit or ingestion, the sexually dimorphic role of the trachea in gut geometry supports the organism's fecundity and starvation resilience. In addition, the authors have made available their code for analyzing the morphological data in GitHub and providing details on how to run their code. Overall, this work adds a new dimension to how organs maintain sexually dimorphic shape. I congratulate the authors on the beautiful study and highly recommend the manuscript for publication in Nature.

Akhila Rajan

No response needed.

Referee #5

The authors have revised the manuscript substantially and have answered to all my concerns. Beautiful work!

No response needed.